# FastIsostasy v1.0 – A regional, accelerated 2D GIA model accounting for the lateral variability of the solid Earth

**Jan Swierczek-Jereczek**[1,2]**, Marisa Montoya**[1,2]**, Konstantin Latychev**[3]**, Alexander Robinson**[4]**, Jorge Alvarez-Solas**[1,2]**, and Jerry Mitrovica**[5]

[1]Department of Earth Physics and Astrophysics, Complutense University of Madrid, Madrid, Spain.
[2]Geosciences Institute, CSIC-UCM, Madrid, Spain.
[3]Seakon, Toronto, Canada.
[4]Alfred Wegener Institute, Helmholtz Centre for Polar and Marine Research, Potsdam, Germany.
[5]Department of Earth and Planetary Sciences, Harvard University, Massachusetts, USA.

**Correspondence:** Jan Swierczek-Jereczek (janswier@ucm.es)

**Abstract.** The vast majority of ice-sheet modelling studies rely on simplified representations of the Glacial Isostatic Adjustment (GIA), which, among other limitations, do not account for lateral variations of the lithospheric thickness and upper-mantle viscosity. In studies of the last glacial cycle using 3D GIA models, this has however been shown to have major impacts on the dynamics of marine-based sectors of Antarctica, which are likely to be the greatest contributors to sea-level rise in the coming centuries. This gap in comprehensiveness is explained by the fact that 3D GIA models are computationally expensive, seldomly open-source and require a complex coupling scheme. To close this gap between "best" and "tractable" GIA models, we here propose FastIsostasy, a regional GIA model capturing lateral variations of the lithospheric thickness and mantle viscosity. By means of Fast-Fourier Transforms and a hybrid collocation scheme to solve its underlying partial differential equation, FastIsostasy can simulate 100,000 years of high-resolution bedrock displacement in only minutes of single-CPU computation, including the changes in sea-surface height due to mass redistribution. Despite its 2D grid, FastIsostasy parametrises the depth-dependent viscosity and therefore represents the depth dimension to a certain extent. FastIsostasy is here benchmarked against analytical, as well as 1D and 3D numerical solutions and shows good agreement with them. For a simulation of the last glacial cycle, its mean and maximal error over time and space respectively yield less than 5 and 16% compared to a 3D GIA model over the regional solution domain. FastIsostasy is open-source, documented with many examples and provides a straight-forward interface for coupling to an ice-sheet model. The model is benchmarked here based on its implementation in Julia, while a Fortran version is also provided to allow for compatibility with most existing ice-sheet models. The Julia version provides additional features, including a vast library of adaptive time-stepping methods and GPU support.

## 1 Introduction

### 1.1 GIA is an important feedback on ice-sheet dynamics

Glacial isostatic adjustment (GIA) denotes the crustal displacement that results from changes in the ice, liquid water and sediment columns, as well as the associated changes in Earth's gravity and rotation axis (Mitrovica et al., 2001), ultimately altering the sea level (Farrell and Clark, 1976). In the present work, we focus on the deformation and gravitational effects. For ice sheets, the former is a net negative feedback on mass balance through the lapse rate of the troposphere and both imply additional negative feedbacks on the dynamics of marine-based regions, where enhanced melting leads to a grounding line retreat but also to a regional bedrock uplift and a decrease of the sea-surface height (SSH), due to the reduced load applied upon the solid Earth and the lesser gravitational pull of the ice sheet on the oceans (Gomez et al., 2010, 2012, 2015, 2018; Whitehouse et al., 2019). As depicted in Fig. 1, these effects combine in a decrease in relative sea level (RSL), which is defined as the difference be-

tween the SSH and the bedrock elevation. Compared to the retreated state (panel (b) of Fig. 1), this decrease ultimately leads to a readvance of the grounding line (panels (c) and (d) of Fig. 1), therefore conditioning the marine ice-sheet instability along with the buttressing effect from ice shelves (Gudmundsson et al., 2012). It was shown that the representation of the deformation and gravitational feedbacks can stabilise grounding lines on retrograde slopes (Gomez et al., 2010, 2012) and that a rapid bedrock uplift can prevent the collapse of marine-terminating glaciers that are transiently forced (Konrad et al., 2015, 2016). Furthermore, an uplifting bedrock might lead isolated bathymetric peaks to connect with the ice sheet, creating so-called pinning points (Adhikari et al., 2014) that further contribute to the stability of a marine ice sheet. Although the negative feedbacks are illustrated here for ice-sheet retreat, they conversely apply to ice-sheet growth.

In addition to these effects, recent work has shown that the ice-sheet evolution might be significantly affected by the forebulge dynamic on longer time scales, for which viscous effects become important. This process denotes the region of comparatively small bedrock uplift (subsidence) surrounding a region of pronounced bedrock subsidence (uplift), which results from a positive (negative) surface load anomaly. Albrecht et al. (in revision) suggest that the forebulge formation represents a positive feedback on ice sheet growth through a decrease of the RSL close to the ice-sheet margin and Kreuzer et al. (in revision) show that a subsiding forebulge can increase sub-shelf melting through the formation of oceanic gateways that ease the intrusion of warm circumpolar deep water.

### 1.2 Laterally-variable structure of the solid Earth modulates GIA

For a given load applied to the solid Earth, the time scale of bedrock deformation is determined by the horizontal extent of the load and the mantle viscosity. The amplitude and the pattern of deformation are in turn determined by the magnitude of the load, the mantle density and the (elastic) lithospheric thickness. These properties are close to being laterally homogeneous in many regions of the solid Earth, which motivated the development of 1D GIA models, where properties are assumed to depend only on the depth coordinate (Dziewonski and Anderson, 1981). However, some regions are an exception to this and present significant lateral variability of solid-Earth properties (further simply referred to as LV), even on relatively short spatial scales. Since Antarctica displays a strong dichotomy between a moderately rifting system in the west and an old craton in the east (Behrendt, 1999), it represents a prototypical example of LV. As depicted in Fig. 2, the lithospheric thickness and upper-mantle viscosity are respectively as little as $50\,\mathrm{km}$ and $10^{18}\,\mathrm{Pa\,s}$ in the west and as large as $250\,\mathrm{km}$ and $10^{23}\,\mathrm{Pa\,s}$ in the east (Barletta et al., 2018; Heeszel et al., 2016; Ivins et al., 2022; Lloyd

et al., 2015, 2020; Morelli and Danesi, 2004; Nield et al., 2014; Whitehouse et al., 2019; Wiens et al., 2022).

For simulations of the Antarctic Ice Sheet (AIS) on the time scale of glacial cycles, accounting for LV by using 3D GIA models has shown great differences compared to 1D GIA models (Albrecht et al., in revision; Gomez et al., 2018; Van Calcar et al., 2023), leading to discrepancies reaching up to $700\,\mathrm{km}$ for the grounding-line position, more than $1\,\mathrm{km}$ for the ice thickness and several metres for the sea-level equivalent volume of the AIS. Although these impacts are large, they are to be expected, given that the AIS is characterised by large marine-based regions. The East-Antarctic basins and the West-Antarctic Ice-Sheet (WAIS) respectively represent sea-level contributions from ice grounded below sea level of about $19.2\,\mathrm{m}$ and $3.4\,\mathrm{m}$ at present day (Fretwell et al., 2013) and their evolution strongly depends on the representation of the GIA feedbacks depicted in Figure 1. While both regions are likely to present abrupt transitions to ice-free conditions under warming scenarios, the WAIS is thought to have particularly low resilience, displaying a bifurcation at a mean global warming of as little as $2\,^\circ\mathrm{C}$ with respect to the pre-industrial era (Garbe et al., 2020). In the context of anthropogenic warming, this is very likely to result in an unprecedented rate of sea-level rise, challenging the adaptation of coastal livelihoods that represent a large portion of human societies (Kulp and Strauss, 2019).

For these reasons, comprehensive projections of sea-level rise require the representation of the Antarctic LV in coupled ice-sheet/GIA settings. Furthermore, the mantle viscosity is uncertain and involves discrepancies of up to two orders of magnitude at some locations in the upper mantle (i.e. above $\sim 670\,\mathrm{km}$ depth), depending on how viscosities are inferred from sparse seismic data (Ivins et al., 2022). Such uncertainties thus need to be propagated to the solution - typically by an ensemble of simulations. This was addressed by Coulon et al. (2021) by relying on a computationally efficient 2D GIA model that allows for laterally-variable relaxation times of the deformational response. However, this is not a standard practice since only a few ice-sheet models are coupled to a laterally-variable solid Earth, typically represented by using a 3D GIA model (Albrecht et al., in revision; Gomez et al., 2018; Van Calcar et al., 2023). This is largely due to the fact that 3D GIA models are computationally too expensive for large ensemble simulations and represent a level of complexity that may be much higher than what is required to answer most of the on-going research questions related to the evolution of ice sheets. With the exception of CitComSVE (Zhong et al., 2022), 3D GIA models do not present open-source codes to this date[1] and some of them even require a commercial license (e.g. Huang et al., 2023). These obstacles have prevented most ice-sheet modelling studies from using 3D GIA models.

---

[1] Some of them can, however, be obtained upon request, such as Seakon (Latychev et al., 2005)

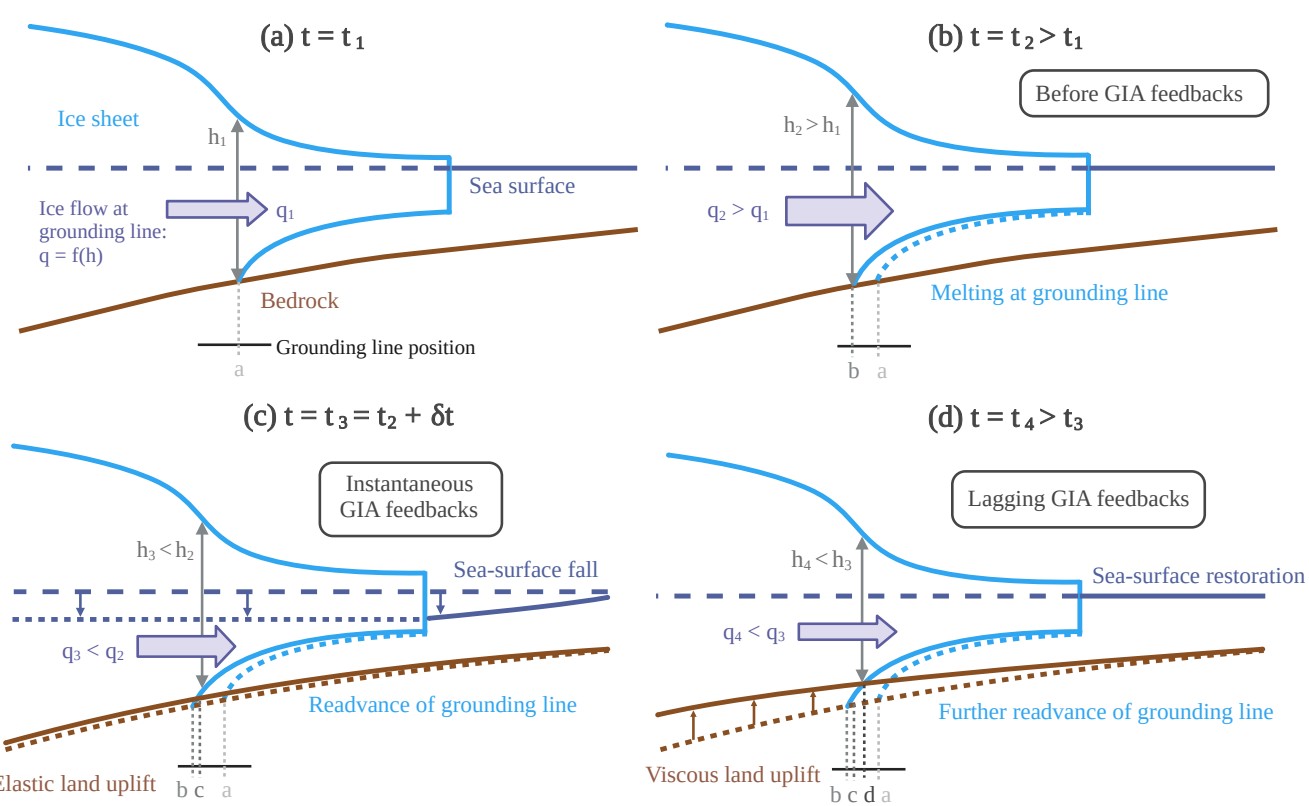

**Figure 1.** Idealised representation, adapted from Whitehouse et al. (2019), of the negative GIA feedbacks on marine-terminating glaciers with retrograde bedrock (e.g. Gomez et al., 2010). We perturb (a) the initial configuration of the ice-sheet by (b) enhanced sub-shelf melting, leading to grounding-line retreat, and therefore larger thickness and increased outflow at the grounding line. (c) The loss of ice leads to an instantaneous ($\delta t \ll 1\,\text{yr}$) decrease of the RSL, which can be decomposed into an elastic uplift of the bedrock and a decrease of the SSH due to the reduction of the gravitational pull on the ocean, leading to a readvance of the grounding line. (d) The elastic uplift is followed by a larger, viscous uplift which further readvances the grounding line and compensates the mass anomaly generated by the ice loss, therefore restoring the SSH close to its original value. The dashed lines used for the ice and the bedrock contour represent their original position (a).

### 1.3 FastIsostasy: reducing the misrepresentation of LV at low computational cost

The vast majority of ice-sheet simulations rely on greatly simplified GIA models without accounting for the parametric uncertainties of the solid Earth, thus potentially introducing biases in sea-level projections (Gomez et al., 2015). This also holds for the Ice-Sheet Model Intercomparison Project (ISMIP) (Seroussi et al., 2020), used as the physical basis for the reports of the Intergovernmental Panel on Climate Change (IPCC). In summary, the ice-sheet modelling community faces the somewhat paradoxical situation of being increasingly aware of how important 3D GIA is, without being able to represent it at a reasonable computational cost. The work of Coulon et al. (2021) partly addresses this with a computationally efficient 2D GIA model but is also characterised by an important limitation: the viscous response is parametrised by a field of relaxation times. However, the re-sponse time scale of the solid Earth depends not only on the viscosity, but also on the wavelength of the load as mentioned above. For these reasons, deriving spatially coherent maps of the relaxation time and constraining them within realistic ranges is not straightforward, as pointed out by Coulon et al. (2021).

To tackle these issues, we here propose FastIsostasy, a regional 2D GIA model inspired by first principles and specially tailored for the needs of ice-sheet modellers. FastIsostasy (1) accounts for LV, (2) parametrises the depth-dependence of the mantle viscosity, (3) captures the dependence of the response time scale on the mantle viscosity and the spatial scale of the load, (4) approximates the regional gravitation response and sea-level evolution, (5) is computationally inexpensive, (6) is extensively tested and (7) offers a simple, open-source and extensively documented interface for a simplified coupling to an ice-sheet model. To illustrate its capabilities, Antarctica is used as leitmotif of the present

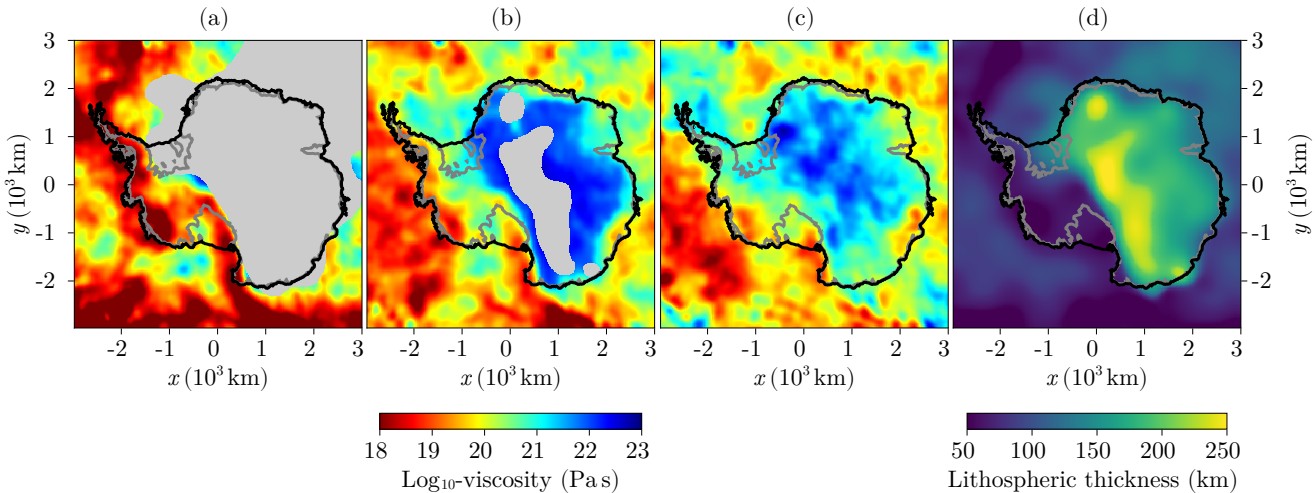

**Figure 2.** (a-c) Upper-mantle viscosity from Whitehouse et al. (2019) and Ivins et al. (2022) at 100, 200 and 300 km depth, respectively. If the lithospheric thickness (Pan et al., 2022), depicted in (d), is larger than the layer depth, a grey shading is applied. The black and dark grey contour lines respectively indicate the present-day ice-sheet and grounded ice margins (Morlighem et al., 2020).

work since it displays (1) a high LV and depth-dependence of solid-Earth properties as depicted in Fig. 2, (2) a high sensitivity to GIA due to the vast marine sectors of the AIS, (3) a large impact on the future of human societies due to possible rapid sea-level rise and (4) large uncertainties in the solid-Earth parameters due, in part, to limited regional data sets. Antarctica might therefore be "the toughest test" when it comes to GIA modelling. Accordingly, the tools provided here are equally well applicable to any other region covered by a past, present or future ice sheet.

### 1.4 FastIsostasy in the model hierarchy

The hierarchy of GIA models displays an important complexity gap between the computationally cheap models, which are largely used by the ice-sheet modelling community, and the computationally expensive models, developed by the GIA community. To give an impression of this, we herein present a brief overview, summarised in Tab. 1 and Tab. 2, of the GIA model classes that are available to date, and focus on the computation of the deformational response, with references to open-source implementations that we are aware of. The governing equations of the three first models can be found in Appendix A.

#### 1.4.1 ELRA

The Elastic-Lithosphere/Relaxed-Asthenosphere (ELRA, Le Meur and Huybrechts, 1996) model conceptualises the structure of the solid Earth as two layers stacked along the depth dimension of a Cartesian coordinate system, obtained by a regional projection of the spherical Earth. The elastic lithosphere is parametrised by its constant thickness

$T(x, y) = T$ and undergoes instantaneous compression under the effect of a load. It is underlain by the asthenosphere[2], idealised as a viscous half-space parametrised by a constant relaxation time $\tau(x, y) = \tau$. According to this, the solution exponentially converges to the equilibrium one, which is computed by convolving the load with a Green's function. Parametrising the transient behaviour with a relaxation time is however simplistic, since, in reality, the response time scale of the solid Earth does not only depend on the viscosity, but also on the wavelength of the load, as mentioned previously. Furthermore, ELRA does not represent the depth-dependence of the mantle viscosity or any LV. Due to its 2D regional domain, it ignores the gravitational and rotational feedbacks on the sea level, as well as the pre-stress and self-gravitation (Purcell, 1998) - the latter being partly cancelled by the lack of sphericity (Amelung and Wolf, 1994). Konrad et al. (2014) demonstrated that ELRA displays important transient differences to a 1D GIA model as well as discrepancies in the representation of the peripheral forebulge. Despite these numerous flaws, ELRA - or even simplified versions of it - remains a widespread choice among ice-sheet modellers (DeConto and Pollard, 2016; Lipscomb et al., 2019; Pattyn, 2017; Quiquet et al., 2018; Robinson et al., 2020; Rückamp et al., 2019), as it mimics the viscoelastic behaviour of the solid Earth with little implementation effort and at low computational cost. Its simplicity has led to a large number of implementations within open-source ice-sheet models (e.g. Lipscomb et al., 2019; Robinson et al., 2020) but, to our

---
[2]In cratonic regions with a thick lithosphere (e.g. East-Antarctica) there might be no asthenosphere. This does not prevent the use of ELRA, which is suited, more generally, to approximate the sublithospheric mantle.

knowledge, no modular implementation is available to date. Ice-sheet modellers have therefore repeatedly spent time implementing ELRA, possibly with suboptimal computational performance as it does not represent the primary focus of their work.

### 1.4.2 ELVA

This modelling approach was proposed by Cathles (1975), applied to ice-sheet modelling for the first time by Lingle and Clark (1985) and efficiently implemented by Bueler et al. (2007) through a Fourier collocation method (FCM). Although this model is sometimes named after the authors of the aforementioned work, we here try to provide a unifying terminology and therefore call it Elastic-Lithosphere/Viscous-mAntle (ELVA). ELVA resembles ELRA in its structure but is parametrised by the spatially homogeneous upper-mantle viscosity $\eta(x,y) = \eta$. It thus avoids any conversion from viscosity to relaxation time and allows the mechanical response to depend on the wavelength of the load (Bueler et al., 2007). Furthermore, it permits embedding more of the radial structure of the mantle viscosity by introducing a viscous channel between the elastic plate and the viscous half-space[3]. However, it does not address any other limitation of ELRA. It is worth mentioning that PISM (Winkelmann et al., 2011) provides an open-source implementation of ELVA, which is however embedded within a larger code base. This lack of modularity is addressed by `giapy` (Kachuck, 2017), a Python implementation of ELVA that might be more accessible than codes traditionally written in Fortran or C++. ELVA was used, for example, by Kachuck et al. (2020) and Book et al. (2022) to study the stabilising potential of rapid bedrock uplift on the WAIS grounding line retreat.

### 1.4.3 LV-ELRA

The laterally-variable ELRA (LV-ELRA) proposed by Coulon et al. (2021) is a generalisation of ELRA to include laterally-variable upper-mantle relaxation time $\tau(x,y)$ and lithospheric thickness $T(x,y)$. The equilibrium displacement is obtained by solving equations derived from thin plate theory (Ventsel and Krauthammer, 2001), which requires the use of Finite Difference Methods (FDMs) and is computationally more expensive than ELRA, since a large system of linear equations needs to be solved. To obtain $\tau(x,y)$, Coulon et al. (2021) apply a Gaussian smoothing on a binary field, with $\tau(x,y) = \tau_1$ in East-Antarctica and $\tau(x,y) = \tau_2$ in the rest of the domain. Since LV-ELRA does not include a lateral coupling between the transient behaviour of neighbouring cells, this smoothing ensures a certain spatial coherence when relaxing the displacement field to the equilibrium solution, i.e. neighbouring cells have similar time scales. In addition to

the limitations mentioned in Sect. 1.3, this coupling prevents the representation of very localised features, depicted in Fig. 2 and inferred in many studies (e.g. Barletta et al., 2018; Heeszel et al., 2016; Nield et al., 2014). In the rest of the manuscript, we will refer to these limitations as the ones resulting from a relaxed rheology.

Although computing the changes in SSH resulting from changes of the Earth's gravity field requires, a-priori, a global domain, it can also be approximated on a regional one. This was done by Coulon et al. (2021) and combined with LV-ELRA, resulting in the so-called *elementary GIA model*. This approach represents one of the most comprehensive regional GIA models developed to date and a valuable improvement for regional modelling, as it bypasses the computational expense of more complex models. The open-source ice-sheet model Kori includes the effort of Coulon et al. (2021), however not in a modular way that is directly usable to other ice-sheet modellers.

### 1.4.4 Global 1D GIA models

Global 1D GIA models capture the radial structure of the solid Earth (down to the core-mantle boundary) but none of its lateral variability. They compute the gravitational field as well as the vertical and horizontal deformation by solving the underlying PDEs after spherical harmonic expansion of the dependent variables. 1D GIA models typically represent the spatial heterogeneity of the SSH, the migration of shorelines and the rotational feedback (Kendall et al., 2005; Mitrovica and Milne, 2003; Spada and Melini, 2019). Most of them were cross-validated by Spada et al. (2011) and Martinec et al. (2018), showing great agreement while presenting intermediate computational cost. However, they are incapable of rendering any LV (e.g. Klemann et al., 2008). Spada and Melini (2019) proposed an open-source implementation of 1D GIA, which remains an exception in the field.

### 1.4.5 Regional 3D GIA models

Based on Finite Element Methods (FEMs), Nield et al. (2018) and Weerdesteijn et al. (2023) have proposed regional models to compute the solid-Earth deformation in the presence of LV. Unlike ELRA, ELVA and LV-ELRA, these regional GIA models resolve the depth dimension (down to the core-mantle boundary) and allow for grid refinement in regions where a higher resolution is needed. In particular, the work of Weerdesteijn et al. (2023) provides an open-source implementation that is compatible with heavily parallel hardware by extending ASPECT, a model originally developed to solve mantle convection problems. Despite this, ASPECT requires about an hour to compute a few hundred years of high-resolution bedrock deformation on 256 CPUs. This represents a computational cost that is too high for most on-going ice-sheet modelling studies, while ignoring the gravitational and rotational effects of GIA.

---

[3]A viscous channel is of finite thickness, unlike a viscous half-space, which is of infinite thickness.

| | ELRA | ELVA | Elementary GIA model (LV-ELRA) | FastIsostasy (LV-ELVA) |
|---|---|---|---|---|
| **Grid** | 2D | 2D | 2D | 2D |
| **Rheology** | Relaxed | Maxwell | Relaxed | Maxwell |
| **LV** | × | × | $\simeq$ | ✓ |
| **Radial structure** | 2 layers (lumped) | 3 layers (lumped) | 2 layers (lumped) | $L$ layers (lumped) |
| **Domain** | regional | regional | regional | regional |
| **Distortion accounted for** | × | × | × | ✓ |
| **Sea level treatment** | × | × | $\simeq$ | $\simeq$ |
| **Variable ocean surface** | × | × | × | $\simeq$ |
| **Rotational feedback** | × | × | × | × |
| **Numerical scheme** | Green's function | FCM | FDM | FDM/FCM |
| **Computational cost** | low | low | low/intermediate | low |
| **Exemplary publications** | Konrad et al. (2014); Le Meur and Huybrechts (1996) | Bueler et al. (2007); Kachuck et al. (2020); Book et al. (2022) | Coulon et al. (2021) | This study |

**Table 1.** Comparison of the GIA models with 2D grids and low computational cost. $L \in \mathbb{N}$ here denotes an arbitrary number of layers. Well-represented phenomena are symbolised by "✓" and neglected ones by "×". Phenomena that are represented with a large amount of simplification are denoted by "$\simeq$". For instance, LV-ELRA is here considered to only partially represent LV, since it is subject to the limitations of a relaxed rheology. Another example of partially represented phenomenon is the change in sea level, which typically requires a global domain but can be reasonably approximated regionally as done in the elementary GIA model and FastIsostasy (c.f. Section 2.4).

### 1.4.6   Global 3D GIA models

3D GIA models account for all the processes represented in 1D GIA models and are, in addition, capable of fully capturing the heterogeneity of solid Earth properties. This also results in simulations that are more complicated to set up, since the user needs to provide fields of lithospheric thickness and mantle viscosity. Unlike 1D GIA models, 3D ones have not been systematically benchmarked but can be considered to be the best technology available for cases like Antarctica. In these models, the computation of the deformational response either relies on spherical harmonics (e.g. Bagge et al., 2021), FEM (e.g. A et al., 2013; Huang et al., 2023; Martinec, 2000; Sasgen et al., 2018; van der Wal et al., 2015; Wu, 2004; Zhong et al., 2022), Finite Volume Method (FVM, Latychev et al., 2005; Gomez et al., 2018), or perturbation theory (e.g. Wu and Wang, 2006). Simulations on glacial time scales typically require at least several hours and up to few days (Albrecht et al., in revision; Pan et al., 2022; Zhong et al., 2022) of computation, even with heavily parallelised codes. This is particularly problematic for propagating parametric uncertainties of the solid Earth on long simulations, since the limit of computational resources is typically reached with only one or few ensemble members. As mentioned above, Zhong et al. (2022) proposed the first open-source implementation of a 3D GIA code.

### 1.4.7   FastIsostasy

The summary above points out a gap between the elementary GIA model (Coulon et al., 2021) and regional 3D GIA models (Nield et al., 2018; Weerdesteijn et al., 2023): the former is computationally cheap but suffers the limitation of a relaxed rheology, whereas the latter accurately represent the viscoelastic response of the solid Earth but come with a computational cost that makes them impractical for most ice-sheet modelling studies. FastIsostasy fills this gap by relying on LV-ELVA, a laterally-variable generalisation of ELVA, coupled to a Regional Sea-Level Model (ReSeLeM), both of which are introduced in Sect. 2, along with a discussion of the underlying limitations. In Sect. 3, we discuss the practical features of its Julia implementation, such as the adaptive time stepping used for integration and GPU support. In Sect. 4, we subsequently benchmark FastIsostasy against analytical, as well as 1D and 3D numerical solutions. Finally, we discuss the results as well as possible future improvements.

**Remarks on open-source codes.** Thanks to recent work, the GIA model classes listed above now present at least one open-source code, which has typically become available much later than the first equivalent proprietary code. For instance, 1D and 3D GIA models already exist for 40 and 20 years, respectively, but their first open-source implementation were only published much later, respectively in Spada and Melini (2019) and Zhong et al. (2022). We here specifically refer to source codes that are licensed and can be downloaded and used without request. However, all open-source codes mentioned above lack tools from modern software development, including (1) dynamically built documentations with code examples, (2) automated test suites and (3) a transparent development, which can be eased, for instance, by the use of GitHub issues and pull requests. Although these as-

|  | Global 1D | Regional 3D | Global 3D |
|---|---|---|---|
| **Grid** | 3D | 3D | 3D |
| **Rheology** | Maxwell | Maxwell | Maxwell |
| **LV** | ✗ | ✓ | ✓ |
| **Radial structure** | $L$ layers (resolved) | $L$ layers (resolved) | $L$ layers (resolved) |
| **Domain** | global | regional | global |
| **Distortion accounted for** | ✓ | ✗ | ✓ |
| **Sea level** | ✓ | ✗ | ✓ |
| **Variable ocean surface** | ✓ | ✗ | ✓ |
| **Rotational feedback** | ✓ | ✗ | ✓ |
| **Numerical scheme** | spherical harmonics | FEM | diverse |
| **Computational cost** | intermediate | intermediate/high | high |
| **Exemplary publications** | Spada and Melini (2019) | Nield et al. (2018); Weerdesteijn et al. (2023) | Latychev et al. (2005); Zhong et al. (2022) |

**Table 2.** Comparison of the GIA models with 3D grid and higher computational cost. $L \in \mathbb{N}$ here denotes an arbitrary number of layers. We here focus on Maxwell rheology, since it is the most commonly used in literature. However, these models can also be adapted to represent other rheologies.

pects are of technical nature, we believe that they can make GIA models more user-friendly and their development more transparent, participative and reliable. This, in turn, eases the progress of research and we have therefore applied these concepts to FastIsostasy.

Since LV-ELVA is a generalisation of ELVA, the latter can be used in FastIsostasy by simply providing it with homogeneous parameter fields. Additionally, we implemented ELRA in an efficient way, further described in Appendix A. Thus, both of these simpler models now present a modular, optimised and documented implementation, distributed under GNU General Public License v3.0.

## 2 Model description

### 2.1 Preliminary considerations

As depicted in Fig. 3, FastIsostasy assumes a rectangular domain $\Omega \subset \mathbb{R}^2$, obtained from a projection of the spherical Earth onto a Cartesian plane with dimensions $2 W_x$ and $2 W_y$, in the directions of the lateral coordinates $x$ and $y$ respectively. We introduce the uniform spatial discretization step $h_x = h_y = h$ such that the domain is subdivided into $N_x \times N_y$ cells, with $N_x, N_y \in \mathbb{N}$. We define all variables that are not specified as scalars (c.f. Tab. 3) to be smooth fields, as, for instance, the vertical load $\sigma^{zz}(x, y, t) : \mathbb{R}^3 \to \mathbb{R}$. For convenience, we will omit the space and time dependence from now on. The discretized equivalent of smooth fields are denoted by bold symbols, e.g. $\boldsymbol{\sigma^{zz}} \in \mathbb{R}^{N_x \times N_y}$ and their entries by the index notation $\sigma^{zz}_{i,j}$, with $i \in \{1, 2, \ldots, N_x\}$, $j \in \{1, 2, \ldots, N_y\}$. The vertical load field is expressed as:

$$\sigma^{zz} = -g \left( \rho^{\mathrm{ice}} \Delta H^{\mathrm{ice}} + \rho^{\mathrm{sw}} \Delta H^{\mathrm{sw}} + \rho^{\mathrm{sed}} \Delta H^{\mathrm{sed}} \right), \quad (1)$$

with $g$ the mean gravitational acceleration at the Earth surface, and $\rho^{\mathrm{ice}}$, $\rho^{\mathrm{sw}}$, $\rho^{\mathrm{sed}}$ the mean densities of ice, seawater and sediment, respectively.[4] The height anomalies $\Delta H^{\mathrm{ice}}$, $\Delta H^{\mathrm{sw}}$ and $\Delta H^{\mathrm{sed}}$ of the corresponding columns are defined with respect to a reference state. On this domain, the first and second spatial derivatives of an arbitrary field $M$ can be computed with central differences:

$$
\begin{aligned}
\mathcal{D}_x M_{i,j} &= \frac{M_{i+1,j} - M_{i-1,j}}{2\, h\, K_{i,j}}, \\
\mathcal{D}_y M_{i,j} &= \frac{M_{i,j+1} - M_{i,j-1}}{2\, h\, K_{i,j}}, \\
\mathcal{D}_{xx} M_{i,j} &= \frac{M_{i+1,j} - 2 M_{i,j} + M_{i-1,j}}{h^2 \cdot K^2_{i,j}}, \\
\mathcal{D}_{yy} M_{i,j} &= \frac{M_{i,j+1} - 2 M_{i,j} + M_{i,j-1}}{h^2 \cdot K^2_{i,j}}, \\
\mathcal{D}_{xy} M_{i,j} &= \mathcal{D}_y \left( \mathcal{D}_x M_{i,j} \right),
\end{aligned}
\quad (2)
$$

with $K$ the distortion factor of the chosen projection.[5] Furthermore, the pseudo-differential operator $|\nabla|$ of an arbitrary matrix $\boldsymbol{M}$ is adapted from Bueler et al. (2007) to suit distorted grids:

$$|\nabla| \boldsymbol{M} = \mathcal{F}^{-1} \left( \boldsymbol{\kappa} \odot \mathcal{F}(\boldsymbol{M}) \right) \oslash \boldsymbol{K}, \quad (3)$$

with $\odot$ the element-wise product, $\oslash$ the element-wise division, $\mathcal{F}$ the Fourier transform, $\mathcal{F}^{-1}$ its inverse and $\boldsymbol{\kappa}$ the coefficient matrix derived in Bueler et al. (2007). Models that do not account for distortion underestimate the length and area of cells away from the reference latitude and therefore

---

[4]FastIsostasy's interface already accepts external forcing from sediments but they will be ignored for the present work.

[5]The distortion $K$ does not appear in $\sigma^{zz}$ since it cancels out when computing the volume-to-area ratio.

require a domain with restricted spatial extent, a limitation that is here overcome.

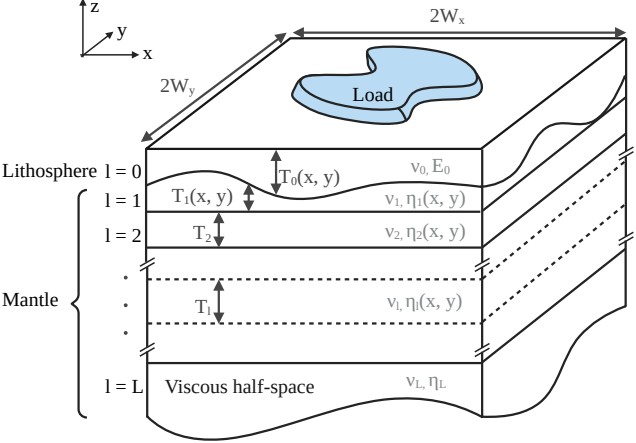

**Figure 3.** Schematic representation of a typical FastIsostasy domain.

## 2.2 Lumping the depth dimension

As depicted in Fig. 3, the vertical structure of the solid Earth is modelled by a stack of layers along the vertical dimension $z$. With the layer index $l \in \{0, 1, \ldots, L-1, L\}$ going from top to bottom, the layers are:

- $l = 0$: an elastic plate with laterally-constant Young modulus $E_0(x,y) = E_0$ and Poisson ratio $\nu_0(x,y) = \nu_0$, and laterally-variable thickness $T_0(x,y)$.

- $l \in \{1, 2, \ldots, L-2, L-1\}$: an arbitrary number of viscous channels, each with laterally-constant Young modulus $E_l(x,y) = E_l$, Poisson ratio $\nu_l(x,y) = \nu_l$ and laterally-variable viscosity $\eta_l(x,y)$. As depicted in Fig. 3, the first of these layers has a laterally-variable thickness $T_1(x,y)$ that is complementary to $T_0(x,y)$ and allows all further layers to have a homogeneous one $T_l(x,y) = T_l$ for $l \geq 2$.

- $l = L$: a viscous half-space with laterally-constant Young modulus $E_L(x,y) = E_L$, Poisson ratio $\nu_L(x,y) = \nu_L$ and viscosity $\eta_L(x,y) = \eta_L$.

Whereas $l = 0$ represents the lithosphere, all further layers represent the remaining mantle. FastIsostasy lumps the latter layers into a single layer by computing a so-called *effective viscosity* for the whole mantle. The key to do so is provided by Cathles (1975), where a three-layer model including an elastic plate, a viscous channel and a viscous half-space is converted into a two-layer model where the viscous channel and the viscous half-space have been lumped into a single half-space by introducing the following scaling factor:

$$R(\kappa, \tilde{\eta}, T) =$$
$$\frac{2\tilde{\eta}\, C\, S + \left(1 - \tilde{\eta}^2\right) T^2 \kappa^2 + \tilde{\eta}^2\, S^2 + C^2}{\left(\tilde{\eta} + \tilde{\eta}^{-1}\right) C\, S + \left(\tilde{\eta} - \tilde{\eta}^{-1}\right) T\kappa + S^2 + C^2}, \quad (4)$$

with $T$ the channel thickness, $\tilde{\eta}$ the channel viscosity divided by the half-space viscosity, $C = \cosh(T\kappa)$ and $S = \sinh(T\kappa)$. The characteristic wavenumber $\kappa = \pi\lambda^{-1}$ is defined by choosing a characteristic wavelength $\lambda$ for the load. Hence, solving the three-layer case can be formulated as solving the two-layer case with the half-space viscosity scaled by $R$. We propose to generalise this idea by performing an induction from the bottom to the top layers, i.e., with decreasing $l$:

**Initialisation:** layer $l = L$ is a viscous half-space with $\eta_l^{\text{eff}} = \eta_L$.

**Induction step:** layer $l + 1$ can be represented as viscous half-space with $\eta_{l+1}^{\text{eff}}$ and is overlain by a viscous channel $l$. These can be converted in an equivalent half-space with effective viscosity:

$$\eta_l^{\text{eff}} = R\left(\kappa, \frac{\eta_l}{\eta_{l+1}^{\text{eff}}}, T_l\right) \cdot \eta_{l+1}^{\text{eff}}. \quad (5)$$

Thus, $\eta_1^{\text{eff}}$ is the effective viscosity of the half-space representing the compound of layers $l \in \{1, 2, \ldots, L-1, L\}$. In essence, this represents a nonlinear mean of the viscosity over an arbitrary number of layers, which is only computed at initialisation and can improve the parametrisation of the depth dimension compared, for instance, to Bueler et al. (2007). However, this approach presents important limitations:

- Since the depth dimension is not resolved, the multimodal response of the Earth to surface loading is not captured as accurately as in 1D and 3D GIA models. For instance, the larger the load, the deeper the deformation into the mantle and the more relevant the radial layering of viscosity and density, which is not captured in FastIsostasy.

- The deformational response is likely to be less accurate for loads with dimensions that substantially differ from the characteristic wavelength $\lambda$, used in Equation 4 to lump the depth dimension. We however emphasise that, for a given ice sheet, $\lambda$ can be chosen such that the near field of deformation is well represented, which is typically what is required in coupled simulations to ice-sheet models. For all the computations presented here, we choose $\lambda$ to be the mean of $\{W_x, W_y\}$.

- Equation 4, as derived in Cathles (1975), applies to laterally-constant viscosities. In the present case, we

| $M_E$ (kg) | $r_E$ (km) | $g\,(\mathrm{m\,s^{-2}})$ | $\rho^{\mathrm{ice}}\,(\mathrm{kg\,m^{-3}})$ | $\rho^{\mathrm{sw}}$ | $\rho^{\mathrm{l}}$ | $\rho^{\mathrm{m}}$ | $E_0$ (Pa) | $\nu_0$ (1) | $A_{\mathrm{pd}}\,(\mathrm{m^2})$ |
|---|---|---|---|---|---|---|---|---|---|
| $5.972\cdot10^{24}$ | 6371 | 9.8 | 910 | 1028 | 3200 | 3400 | $6.6\cdot10^{10}$ | 0.28 | $3.625\cdot10^{14}$ |

**Table 3.** Numerical values of constants in FastIsostasy, from left to right: Earth mass, Earth radius, mean gravitational acceleration at Earth's surface, density of ice, sea-water, lithosphere and mantle, elastic modulus and Poisson ratio of the lithosphere (compressible), present-day ocean surface area. Values for the solid Earth are largely derived from the Preliminary Reference Earth Model (PREM, Dziewonski and Anderson, 1981) and the ocean surface from Cogley (2012).

however allow the viscosity to be laterally variable and apply this scaling for each column.

Finally, to account for compressibility and lateral variations of the shear modulus, a scaling $\alpha$ of the viscosity is introduced and described in Appendix C. This yields the corrected effective viscosity $\eta$, which brings the Maxwell time of FastIsostasy close to that of a 3D GIA model;

$$\eta = \alpha\,\eta_1^{\mathrm{eff}}. \tag{6}$$

Although converting the 3D problem into a 2D one introduces the limitations mentioned above, it also greatly reduces the computational cost. In addition, the partial differential equation (PDE) governing an elastic plate on a viscous half-space can be transformed into an ordinary differential equation (ODE), as we describe in the next section.

## 2.3 LV-ELVA

We now assume that the aforementioned lumping of the layers has been performed and that the lithosphere and underlying mantle are represented by an elastic plate overlaying a viscous half-space. Since the vertical extent of the plate is typically two orders of magnitude smaller than its horizontal one, it is considered to be thin. By assuming a Maxwell rheology, the total vertical displacement $u^{\mathrm{el}} + u$ of the bedrock resulting from a stress $\sigma^{zz}$ can be decomposed in an elastic and a viscous response, respectively denoted by $u^{\mathrm{el}}$ and $u$. As in Bueler et al. (2007), the elastic response of the lithosphere is computed by a convolution of the load $\sigma^{zz}$ with an appropriate Green's function $\Gamma^{\mathrm{el}}$:

$$u^{\mathrm{el}} = \Gamma^{\mathrm{el}} \otimes \frac{K^2\sigma^{zz}}{g}. \tag{7}$$

This represents the instantaneous compression of the lithosphere and accounts for the distortion resulting from the projection. In reality, this process takes place on the time scale of days, but it can be considered to be instantaneous compared to the long time scales of the viscous response and the ice-sheet dynamics. To construct the elastic Green's function, a "Guttenberg-Bullen A" spherical Earth model is assumed, and values from Farrell (1972, table A3) are used, as done by Bueler et al. (2007). This treatment of the elastic response

shows great agreement with a 3D GIA model, as shown in Section 4.

When material from the solid Earth is displaced, a hydrostatic force counteracting the load arises. We define the pressure field $p$ as the sum of all these effects:

$$p = \sigma^{zz} - g\left(\rho^{\mathrm{l}}\,u^{\mathrm{el}} + \rho^{\mathrm{m}}\,u\right), \tag{8}$$

with $\rho^{\mathrm{l}}$ and $\rho^{\mathrm{m}}$ mean densities of the lithosphere and the upper mantle. Since the displacement occurs in Earth's outermost layers, we here assume $g$ to be constant over these shallow depths. 3D GIA models usually represent the elastic lithosphere as a viscous layer with very high viscosity and the elastic displacement therefore also implies a hydrostatic force. We argue that this is closer to reality and adapt this point of view to the present context by including the elastic displacement in Eq. 8, unlike Bueler et al. (2007) and Coulon et al. (2021). The evolution of the viscous displacement is therefore coupled to the elastic one and is governed by:

$$2\eta(x,y)\,|\nabla|\left(\frac{\partial u}{\partial t}\right) = F$$

$$F = p + \frac{\partial^2 M_{xx}}{\partial x^2} + 2\frac{\partial^2 M_{xy}}{\partial x\partial y} + \frac{\partial^2 M_{yy}}{\partial y^2} \tag{9}$$

with $M_{xx}$, $M_{yy}$, $M_{xy}$ the flexural moments for a thin plate (Coulon et al., 2021; Ventsel and Krauthammer, 2001):

$$M_{xx} = \int_{-T_0/2}^{T_0/2} \sigma^{xx} z\,dz = -D\left(\frac{\partial^2 u}{\partial x^2} + \nu\frac{\partial^2 u}{\partial y^2}\right), \tag{10}$$

$$M_{yy} = \int_{-T_0/2}^{T_0/2} \sigma^{yy} z\,dz = -D\left(\frac{\partial^2 u}{\partial y^2} + \nu\frac{\partial^2 u}{\partial x^2}\right), \tag{11}$$

$$M_{xy} = \int_{-T_0/2}^{T_0/2} \sigma^{xy} z\,dz = -D(1-\nu)\frac{\partial^2 u}{\partial x\partial y}. \tag{12}$$

In these equations, $D = E_0\,T_0^3(x,y)\,(12(1-\nu_0^2))^{-1}$ is the laterally-variable lithospheric rigidity field. The PDE can be understood as an ad-hoc generalisation of ELVA (Bueler et al., 2007; Cathles, 1975; Lingle and Clark, 1985) that is

inspired by LV-ELRA (Coulon et al., 2021) and further described in Appendix A. Though we did not manage to formally derive it by generalising the work of Cathles (1975) to heterogeneous viscosities, Eq. 9 yields results that are very close to those of a 3D GIA model, as shown in Section 4. The right-hand side $F$ of the PDE can be evaluated by finite differences, as defined in Equation 2:

$$\boldsymbol{F} = \boldsymbol{p} + \mathcal{D}_{xx}\boldsymbol{M}_{xx} + 2\mathcal{D}_{xy}\boldsymbol{M}_{xy} + \mathcal{D}_{yy}\boldsymbol{M}_{yy}. \tag{13}$$

Subsequently, a Fourier collocation of this equation can be achieved by making use of Eq. 3 and rearranging terms:

$$\frac{\partial \boldsymbol{u}}{\partial t} = \mathcal{F}^{-1}\left(\mathcal{F}(\boldsymbol{F} \odot \boldsymbol{K} \oslash 2\boldsymbol{\eta}) \oslash (\boldsymbol{\kappa} + \boldsymbol{\varepsilon})\right), \tag{14}$$

with $\varepsilon \ll 1$ a regularisation term to avoid division by 0. Thus, by using a hybrid FDM/FCM approach the PDE expressed in Eq. 9 is transformed into an ODE, which can be solved with explicit integration methods. In particular, we thus avoid solving a large system of linear equations, as normally done in FDM, FVM or FEM codes. Note that in Bueler et al. (2007) the closed form of a Crank-Nicolson (implicit) scheme is derived, thus providing unconditional stability. Due to the complexity of the right-hand side, finding such a closed form for LV-ELVA is more challenging and goes beyond the scope of this work. We emphasise that the smaller time steps resulting from explicit schemes might nevertheless be needed for (1) coupling purposes, as a dense output in time can be provided to the ice-sheet model, and (2) accurately capturing the fast dynamics that can occur in regions of low viscosity.

Far away from the ice sheet, the changes of the load are comparatively small and we therefore require the displacement to be zero at the domain boundaries[6]. However, FCM does not allow explicit treatment of Dirichlet boundary conditions (BCs). To enforce its approximate representation, we subtract the mean displacement of the corner vertices from the solution at each time step, which is here expressed with the common choice of notation from programming:

$$u_{i,j} := u_{i,j} - \frac{1}{4}(u_{1,1} + u_{1,N_y} + u_{N_x,1} + u_{N_x,N_y}). \tag{15}$$

Note that this differs from Bueler et al. (2007), where the whole domain boundary is used for this purpose. We argue that our approach is a better representation of the required BC because corner points are (1) further away from the load and (2) equidistant from the centre of a rectangular domain.

---

[6]This is not completely correct since the ocean load changes at the domain margin, which is however impossible to represent accurately in a regional model.

## 2.4 Regional sea-level model (ReSeLeM)

In a coupled setting, a GIA model is typically expected to take the ice thickness field as an input and to return the RSL field, $S$, as an output, which can be expressed as:

$$
\begin{aligned}
S(x,y,t) &= z^{\mathrm{ss}}(x,y,t) - z^{\mathrm{b}}(x,y,t), &(16)\\
z^{\mathrm{ss}}(x,y,t) &= s(t) + N(x,y,t) + c(t), &(17)\\
z^{\mathrm{b}}(x,y,t) &= z^{\mathrm{b}}_{\mathrm{ref}}(x,y) + u(x,y,t) + u^{\mathrm{el}}(x,y,t), &(18)
\end{aligned}
$$

with $z^{\mathrm{ss}}$ the SSH, $z^{\mathrm{b}}$ the bedrock elevation, $s$ the barystatic sea level (BSL), $N$ the SSH perturbation due to changes in the gravitational field and $c$ a time-dependent scalar. In a global model, $c$ ensures mass conservation of water, which is however impossible to ensure in a regional model with open boundaries. In contrast, we here use $c$ to impose a mean zero SSH perturbation at the domain boundary, similar to Equation 15. The sea-level terminology used here follows the definitions of Gregory et al. (2019), which we refer to for any further detail. Whereas the displacements $u$ and $u^{\mathrm{el}}$ are computed as described in Sect. 2.3, the SSH perturbation $N$ can be regionally approximated by the convolution of the mass anomaly with an appropriate Green's function $\Gamma^N$, as proposed in Coulon et al. (2021):

$$
\begin{aligned}
N &= \Gamma^N \otimes \frac{K^2 p}{g}, &(19)\\
\Gamma^N(\theta) &= \frac{R_{\mathrm{e}}}{M_{\mathrm{e}}}\left(\frac{1}{2\sin(\theta/2)}\right), &(20)
\end{aligned}
$$

with $R_{\mathrm{e}}$ the Earth radius at the equator, $M_{\mathrm{e}}$ the Earth mass, and $\theta$ the colatitude. To avoid dividing by $\theta = 0°$, we impose a minimal colatitude of the order of the resolution. By neglecting the changes in surface load from sediments and expressing $\Delta H^{\mathrm{sw}}$ with the RSL, Eq. 1 becomes:

$$
\begin{aligned}
\sigma^{zz} &= -g\mathcal{A}(\rho^{sw}\Delta S\mathcal{O} + \rho^{\mathrm{ice}}\Delta H^{\mathrm{ice}}\mathcal{C}), &(21)\\
\Delta S &= S - S_{\mathrm{ref}}, &(22)\\
\Delta H^{\mathrm{ice}} &= H^{\mathrm{ice}} - H^{\mathrm{ice}}_{\mathrm{ref}}, &(23)
\end{aligned}
$$

with $\mathcal{O}$ the ocean function, $\mathcal{C}$ the continent function, $\mathcal{G}$ the grounded-ice function, which can be expressed by introducing the indicator function $\mathbb{1}$ and the ice thickness above flotation $H^{\mathrm{af}}$:

$$
\begin{aligned}
\mathcal{O} &= 1 - \max(\mathcal{C},\mathcal{G}), &(24)\\
\mathcal{C} &= \mathbb{1}(S < 0), &(25)\\
\mathcal{G} &= \mathbb{1}(H^{\mathrm{af}} > 0), &(26)\\
H^{\mathrm{af}} &= H^{\mathrm{ice}} - S\frac{\rho^{\mathrm{ice}}}{\rho^{\mathrm{sw}}}. &(27)
\end{aligned}
$$

In Eq. 21, the activation mask $\mathcal{A}$ defines what we further consider the near field of displacement: it yields 1 close to the ice sheet and 0 otherwise. This approach is similar to what is done in Coulon et al. (2021) and is, by definition, somewhat arbitrary. The choice made here is illustrated in Sect. 4 for Antarctica. Most importantly, the activation mask enforces a zero change in load close to the boundary of the domain, which is necessary to fulfil the BCs expressed in Equation 15. This is a limitation compared to a global GIA model but allows to account for water column changes in the near field of the ice sheet, unlike most regional models (Book et al., 2022; Bueler et al., 2007; Kachuck et al., 2020; Lingle and Clark, 1985; Le Meur and Huybrechts, 1996; Weerdesteijn et al., 2023).

In Goelzer et al. (2020), the evolution of the BSL is described as[7]:

$$s(t) = \frac{V(t)}{A_{\mathrm{pd}}}, \tag{28}$$

$$V(t) = V^{\mathrm{af}}(t)\frac{\rho^{\mathrm{ice}}}{\rho^{\mathrm{sw}}} + V^{\mathrm{pov}}(t) + V^{\mathrm{den}}(t), \tag{29}$$

with $A_{\mathrm{pd}}$ the present-day ocean surface and $V$ the volume contribution of ice sheets to the ocean. The latter is decomposed into $V^{\mathrm{af}}$, the contribution from ice above flotation; $V^{\mathrm{pov}}$, the contribution from changes in the bedrock height; and $V^{\mathrm{den}}$, the contribution from density differences between meltwater and seawater. We refer the reader to Goelzer et al. (2020) for the detailed computation of these quantities, which are defined with respect to a reference state, typically the present-day one. Assuming a fixed ocean surface to compute the evolution of the sea level can, however, lead to a bias of tens of metres over glacial cycles. To tackle this, we propose an extension of Goelzer et al. (2020) that accounts for the time dependence of the ocean surface $A(t)$ when computing the BSL. To this end, we first introduce a time discretisation $t = k\,\Delta t$, with $\mathcal{O}(\Delta t) = 10\,\mathrm{yr}$. We further introduce the ocean surface as a function $A(s)$, which is nonlinear with respect to the BSL $s$ when a realistic topography is used, as shown in Fig. 4 and described with more detail in Appendix B. The volume change $\Delta V_k = V_k - V_{k-1}$ over a time step leads to a change $\Delta s_k = s_k - s_{k-1}$ of the BSL. Since $\Delta t$ is much smaller than glaciological time scales, $\Delta s_k$ is a small number and the nonlinear relationship between the volume contribution, the ocean surface and the BSL can be approximated by a trapezoidal rule, pictured in Fig. 4 and described by the following equation:

$$\Delta V_k = (s_k - s_{k-1})\frac{A(s_k) + A(s_{k-1})}{2}. \tag{30}$$

---

[7]We refer the reader to Adhikari et al. (2020) for an alternative treatment.

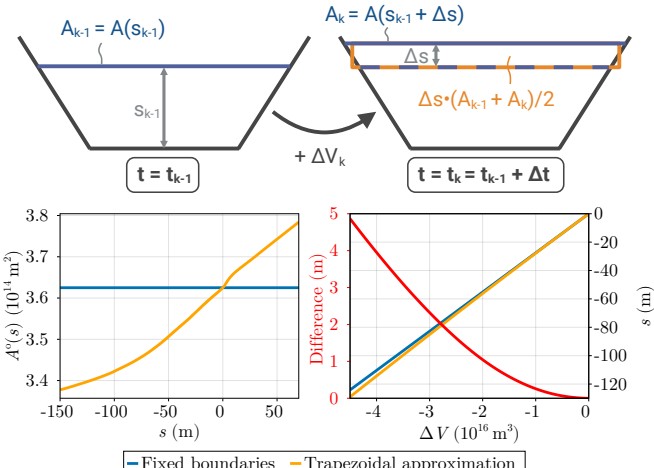

**Figure 4.** (Top) Schematic representation of the trapezoidal approximation used to solve the nonlinear dependence between BSL, $s$, and ocean surface, $A$. We hereby use $A_k$ as shorthand for $A(s_k)$. (Bottom left) Present-day ocean surface and ocean-surface function, $A(s)$, as computed by the trapezoidal approximation of the basin evolution. (Bottom right) BSL computed for a change in ice volume equivalent to the LGM, for fixed boundaries compared to the trapezoidal approximation.

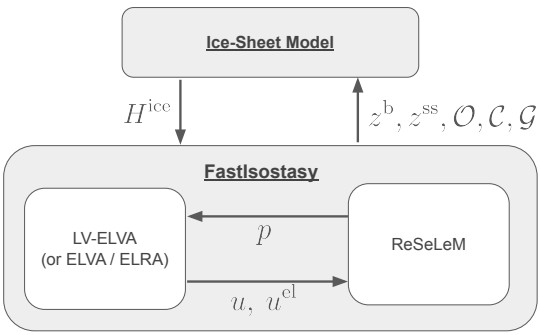

**Figure 5.** Interface between FastIsostasy and an ice-sheet model, adapted from De Boer et al. (2017) and Coulon et al. (2021).

Equation 30 is solved by using $s_{k-1}$ as an initial guess and the updated BSL, $s_k$, is typically obtained after a few iterations of the nonlinear solver. This is of course an important simplification compared to global GIA models, which typically resolve the migration of shorelines (Kendall et al., 2005; Mitrovica and Milne, 2003). Nonetheless, this is an improvement compared to fixing $A(t) = A_{\mathrm{pd}}$. In particular, the bottom-right panel of Fig. 4 shows that the sea level of the Last Glacial Maximum (LGM) is overestimated by about $5\,\mathrm{m}$ for fixed ocean boundaries compared to our trapezoidal approximation. This can lead to differences of several kilometres in the grounding-line position, depending on the local bedrock slope. We emphasise that a more sophisticated ap-

proach than ours is likely to require a global domain, which we here want to avoid.

In summary, allowing for a time variable ocean surface is the main adaptation of the sea-level treatment described in previous work (Coulon et al., 2021; Goelzer et al., 2020) and it constitutes ReSeLeM. FastIsostasy involves, as depicted in Fig. 5, a coupling between LV-ELVA and ReSeLeM.

## 2.5 Limitations

LV-ELVA presents limitations, since it relies on a linear PDE describing the macroscopic behaviour of the solid Earth as a Maxwell body. Therefore it does not account for transient rheologies (Caron et al., 2017; Ivins et al., 2021), nonlinear rheologies (Gasperini et al., 2004; Kang et al., 2021), composite rheologies (van der Wal et al., 2010, 2015), anisotropy (Accardo et al., 2014; Beghein et al., 2006), or microscale properties of the material (Van Calcar et al., 2023). LV-ELVA only computes the vertical displacement of GIA and neglects the horizontal one, which has a negligible impact on ice-sheet dynamics. Nonetheless, the horizontal displacement might be used to constrain GIA models through GNSS measurements and its implementation is left for future versions of the model.

The governing equation of LV-ELVA was postulated here in an ad-hoc way, as described in Appendix A, but lacks a formal derivation. Since the depth dimension is not resolved in LV-ELVA, Stokes flow of the mantle is not fully represented, similar to depth-integrated solvers of ice-sheet dynamics and shallow-water approximations used in general circulation models. In addition, the regional nature of the domain makes it inherently complicated to ensure BCs that are consistent with a global conservation of mass. Therefore, the sea-level is only solved in an approximate way and the feedback from perturbations in the Earth's rotation, which will be small near the poles, is not accounted for. In particular, we emphasise that the update of the BSL described in Eq. 30 fails to give good results if most of the contribution stems from an ice sheet that is not included in the domain. For instance, the BSL computed in FastIsostasy over a glacial cycle cannot be correct if the domain only covers Antarctica, since most of the contribution stems from the Northern Hemisphere ice sheets. However, this can however be by setting up a domain for the Northern Hemisphere and coupling it to the Antarctic one, or simply by taking the BSL obtained from proxies or global GIA models.

Future releases of FastIsostasy will focus on addressing some of these problems. We emphasise that, despite these limitations, the results presented in Sect. 4 show that FastIsostasy is a significant improvement compared to ELRA and ELVA, since it represents the near-field GIA response of laterally-variable Earth structures with improved accuracy and a negligible increase in computational cost. We believe that this adequately covers the needs of most ice-sheet modellers.

## 3   Implementation, performance and further remarks

FastIsostasy has been implemented in Julia (FastIsostasy.jl) and in Fortran. Julia (Bezanson et al., 2017) is a high-performance language with a vast ecosystem, on which FastIsostasy.jl relies to offer convenient features and efficient computation:

1. To evaluate the right-hand side of the ODE obtained in Eq. 14 and perform the convolutions used to compute the elastic and the gravitational response, FastIsostasy.jl uses forward and inverse Fast-Fourier Transforms (FFTs), which are implemented in an optimised way in FFTW.jl (Frigo and Johnson, 2005). Evaluating the right-hand side therefore scales with a computational complexity of $\mathcal{O}(N \log_2 N)$, for a matrix of size $N = N_x \times N_y$. To achieve an even better speed increase, (1) $N_x, N_y$ are generally chosen as powers of 2, (2) FFTs are precomputed as far as possible and (3) the transforms are computed in-place to reduce the memory allocation. FastIsostasy owes its name to `fast-earth`, an early implementation of ELVA (referred in the acknowledgements) and to its reliance on FFTs to perform all the expensive computations.

2. To subsequently integrate the right-hand side in time, FastIsostasy.jl uses OrdinaryDiffEq.jl (Rackauckas and Nie, 2017), a package that offers a wide range of optimised routines. We here restrict ourselves to explicit methods, which range from the simplest explicit Euler scheme up to schemes of order 14. For all the results presented here, we used the Runge-Kutta method proposed in Bogacki and Shampine (1996). Explicit integration schemes typically require decreasing the time step with increasing spatial resolution, which is handled by the adaptive time-stepping methods of OrdinaryDiffEq.jl to prevent instabilities. This requires more evaluations of the right-hand side and leads the scaling of computational complexity for the full problem to be higher than $\mathcal{O}(N \log_2 N)$ associated with FFTs. By providing keyword arguments, the user is able to influence any option related to the integration in time, such as the scheme, the error tolerance, the minimal time step, etc.

3. FastIsostasy.jl uses CUDA.jl (Besard et al., 2019) and ParallelStencil.jl to optionally run performance-relevant computations on a GPU (so far restricted to NVIDIA hardware). Due to their heavily parallelised architecture, GPUs are able to scale better than CPUs for some computations. The speed increase thus obtained will be illustrated in Test 1 of the model validation. Offering a GPU-parallelised GIA code is unprecedented to our knowledge and only requires the user to set the keyword argument `use_cuda=true`.

4. In FastIsostasy.jl, the nonlinearity introduced by the time-dependent ocean surface is solved by using NL-

solve.jl, in combination with an interpolation of $A(s)$, which is constructed at initialisation using Interpolations.jl. Since $A(s)$ is monotonic and initial guesses are close to the solution, the computation time associated with this step is negligible. Furthermore, whereas the adaptive time-stepping is convenient to enforce stability of the viscous displacement, updating the diagnostics - such as the elastic displacement, the ocean surface and sea level - can be done less frequently. For instance $\Delta t = 10\,\text{yr}$ is used in the present work and can be determined by the user through a keyword argument.

As illustrated above, FastIsostasy.jl relies on numerous Julia packages. Since it is a registered package, it can however be easily installed, along with all its dependencies, by simply running `add FastIsostasy` in Julia's package manager. Furthermore, it is thoroughly documented at https://janjereczek.github.io/FastIsostasy.jl/dev/, including a description of the application programming interface, a tutorial and practical examples, which are a simplified version of the code used for the results shown in Section 4. Additionally, FastIsostasy.jl is designed in a modular way that facilitates its coupling to an ice-sheet model and we therefore believe that the implementation burden associated with its use is very low.

Since Julia does not yet support compilation to binaries, FastIsostasy is additionally programmed in Fortran to allow for compatibility with most existing ice-sheet models and has already been coupled to Yelmo (Robinson et al., 2020). Since Fortran does not provide packages that allow convenience at the level of the Julia ecosystem, the Fortran version: (1) does not allow computation on GPU, (2) only provides explicit Euler for integration in time, and (3) does not allow for time-evolving ocean boundaries.

## 4 Model validation and benchmarks

We now validate FastIsostasy with series of tests:

– Test 1: a comparison to an analytical solution for an idealised load on a homogeneous, flat Earth. This aims to check that the numerics are well-implemented for the simplest case and that our results are comparable to Bueler et al. (2007).

– Test 2: a comparison to benchmark solutions of three different 1D GIA models, presented in Spada et al. (2011). This aims to understand the discrepancies which can arise from the lumping of depth-dependent viscosity profiles and the regional approximations of the SSH perturbation.

– Test 3: a comparison to the 3D GIA model Seakon (Gomez et al., 2018; Latychev et al., 2005) on idealised cases of LV. This aims to check whether Eq. 9 and its

discretization, Eq. 14, are valid approximations of the deformational response in the presence of LV. Here we will also compare the elastic displacement of Seakon and FastIsostasy.

– Test 4: a comparison to Seakon with realistic LV and forced by the ice loading of a full glacial cycle. This aims to check whether loads and Earth structures of typical applications can be reasonably well represented.

These tests are summarised in Tab. 4 and aim to quantify, as independently as possible, each source of error between FastIsostasy and the baseline solutions listed above. This is measured by an absolute and a relative value, respectively defined as:

$$e^{\text{abs}}(x,y,t) = |u_{\text{FI}}(x,y,t) - u_{\text{BL}}(x,y,t)|, \tag{31}$$

$$e(x,y,t) = \frac{e^{\text{abs}}(x,y,t)}{\max\limits_{x,y,t} |u_{\text{BL}}(x,y,t)|}, \tag{32}$$

with the indices "FI" and "BL" respectively indicating the FastIsostasy and baseline solutions. We refer to the mean and maximal errors over space as $\bar{e}(t)$ and $\hat{e}(t)$. In the forthcoming analysis, we will often mention a tight upper bound for these timeseries to quantify them in a scalar way, and will emphasise the maximal error, since the spatial mean can hide important local discrepancies.

### 4.1 Test 1 – Analytical solution for idealised load on homogeneous Earth

We first reproduce the test proposed in Bueler et al. (2007) by using a 2-layer model with $W_x = W_y = 3000\,\text{km}$, $N = N_x = N_y = 256$ and $h \simeq 23\,\text{km}$. The first layer is parametrised by the lithospheric thickness $T(x,y) = 88\,\text{km}$ and the underlying half space by the mantle viscosity $\eta(x,y) = 10^{21}\,\text{Pa\,s}$. The load is a Heaviside function in time that represents a flat cylinder of ice, with radius $R = 1000\,\text{km}$ and thickness $H = 1\,\text{km}$, placed at the centre of the computation domain. For this idealised case, an analytical solution of the viscous solution is provided in Bueler et al. (2007), yielding:

$$u(r,t) = \rho^{\text{ice}} g H R \int\limits_0^\infty \psi(\kappa)\text{d}\kappa, \tag{33}$$

$$\psi(\kappa) = \beta^{-1}\left[\exp\left(-\frac{\beta t}{2\eta\kappa}\right) - 1\right] J_1(\kappa R_0) J_0(\kappa r) \tag{34}$$

with $J_0$ and $J_1$ the Bessel functions of first kind and respectively of order 0 and 1, and $\beta = \beta(\kappa) = \rho^{\text{m}} g + D\kappa^4$. To make the solution of FastIsostasy comparable to this, we set $K_{i,j} = 1$ and $\rho^{\text{l}} = 0$, which neglects distortion and prevents the elastic displacement from contributing to the pressure term. Panel (a) of Fig. 6 shows cross-sections of the domain along the $x$ dimension, demonstrating that the numerical solution closely follows the analytical one. In complement,

| Test | Compared to... | Load | $T, \eta$ | $L$ | $\bar{e}(t) < \dots$ | $\hat{e}(t) < \dots$ |
|------|----------------|------|-----------|-----|----------------------|----------------------|
| 1 | Analytical solution | Ice cylinder | homogeneous | 1 | 0.019 | 0.021 |
| 2 | 1D GIA models | Ice cylinder & ice cap | homogeneous | 2 | 0.11 | 0.19 |
| 3 | 3D GIA model | Ice cylinder | Gaussian (c.f. Appendix D) | 2 | 0.05 | 0.15 |
| 4 | 3D GIA model | Glacial cycle (ICE6G_D) | Antarctic LV | 3 | 0.05 | 0.16 |

**Table 4.** Summary of the tests performed using FastIsostasy. The two last columns provide tight upper bounds on the mean and maximal errors over space. The Antarctic LV used in Test 4 is the same as in Pan et al. (2022)

panel (b) shows the corresponding maximal and mean error over time. For $t \geq 5000\,\mathrm{yr}$, the viscous displacement is captured with $\bar{e}^{\mathrm{abs}} < \hat{e}^{\mathrm{abs}} < 1\,\mathrm{m}$. For $t \leq 2000\,\mathrm{yr}$, the displacement surface is well captured in terms of shape but appears to be slightly shifted along the $z$-dimension due to the approximate treatment of the BCs as written in Eq. 15, leading to a larger upper bound on the error $\hat{e}^{\mathrm{abs}}(t) < 5.8\,\mathrm{m}$, which corresponds to $\hat{e} < 0.021$. Whereas in Bueler et al. (2007) a correction of this effect is applied based on the knowledge of the analytical solution, we here decide not to do so. First, because such correction only applies to this specific case and second, because users should be informed about the potential numerical error that will arise in their experiments. When imposing Eq. 15, the unrealistically high forcing rate resulting from the Heaviside load leads to errors that are higher than what is expected from a load that is coherent in time. The discretisation error presented here should be therefore understood as an upper bound for a flat Earth.

Panel (c) shows that the maximal and mean equilibrium error respectively decrease with a slope of $-0.4$ and $-0.33$ in $\log_2 - \log_{10}$ space, showing that convergence to the analytical solution of equilibrium can be achieved relatively quickly with increasing resolution. The run-time on CPU (Intel i7-10750H 2.60GHz) versus GPU (NVIDIA GeForce RTX 2070) is depicted in panel (d) and shows that using a GPU is advantageous for $N \geq 64$, which corresponds to the typical problem size for ice-sheet modelling. More specifically, the CPU and GPU computation time respectively increase with a slope of 0.73 and 0.17 in $\log_2 - \log_{10}$ space, thus giving a clear advantage to GPU computation for large problems. Thanks to the hybrid FCM/FDM scheme used to evaluate Eq. 14, the scaling of computation time on both CPU and GPU is better than is usually obtained from FDM, FVM and FEM since all of them rely on solving a large system of linear equations.

### 4.2 Test 2 – 1D GIA solutions of idealised loads on layered Earth

In Spada et al. (2011), a range of 1D GIA models are benchmarked against each other and show excellent agreement on various experiments. Here, we reproduce the benchmark tests called "1/2" (geodetic quantities) and "2/2" (geodetic rates),

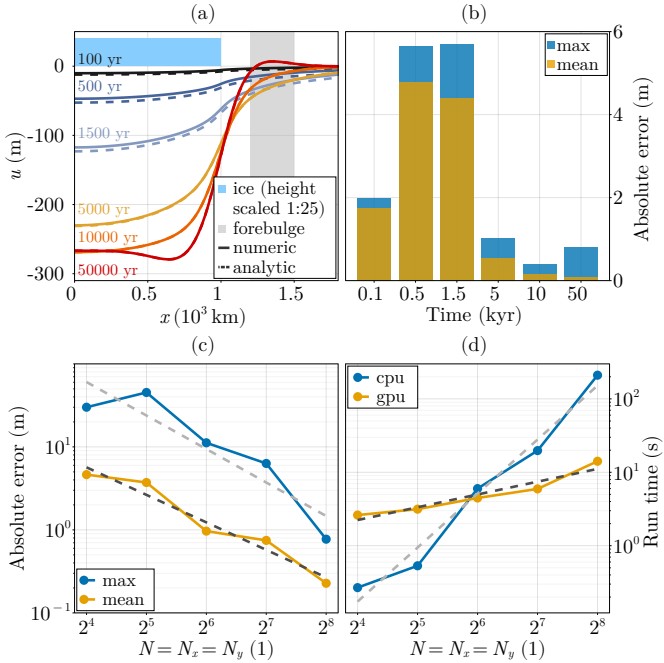

**Figure 6.** (a) Transient cross-sections of bedrock displacement along $x$-axis, from the center of the domain until $x = 1800\,\mathrm{km}$. The resolution used here is $N_x = N_y = 2^8$, $h \simeq 23\,\mathrm{km}$. (b) Corresponding mean and maximal errors over time. (c) Resolution-dependence of maximum and mean error at equilibrium with respect to the analytical solution, with the light and dark grey dashed lines representing the corresponding linear regressions, respectively with slopes of $-0.4$ and $-0.33$ in $\log_2 - \log_{10}$ space. (d) Resolution dependence of the computation time on CPU versus GPU, with the light and dark grey dashed lines representing the corresponding linear regressions, respectively with slopes of 0.73 and 0.17 in $\log_2 - \log_{10}$ space.

which are similar to Test 1 but presents the following differences:

– The computation domain is a stereographic projection of the spherical Earth, centred at colatitude $\theta = 0°$, and the effect of distortion is therefore included, unlike for Test 1. We apply ice loads with $\rho^{\mathrm{ice}} = 931\,\mathrm{kg\,m^{-3}}$, cho-

sen in agreement with Spada et al. (2011), and the following geometries:

(A) an ice cap with maximal height $H_{\max} = 1.5\,\mathrm{km}$, radius $\theta = 10°$ and its shape defined by a cosine function.

(B) a cylindrical ice load of thickness $H = 1\,\mathrm{km}$ and radius $\theta = 10°$,

- The Earth structure has three layers, namely (1) a lithosphere of thickness $T_0 = 70\,\mathrm{km}$ and shear modulus $G_0 = 5 \cdot 10^{11}\,\mathrm{Pa}$, (2) an upper mantle of thickness $T_1 = 600\,\mathrm{km}$ and viscosity $\eta_1 = 10^{21}\,\mathrm{Pa\,s}$ and (3) a lower mantle reaching down to the core-mantle boundary with a viscosity $\eta_2 = 2 \cdot 10^{21}\,\mathrm{Pa\,s}$. For any further detail, we refer the reader to the M3-L70-V01 profile shown in (Spada et al., 2011). In FastIsostasy, these layers are translated into an elastic plate, a viscous channel and a viscous half-space.

- The results of SSH perturbation provided in Spada et al. (2011) allow us to check the validity of Eq. 19, used in FastIsostasy.

Figure 7 demonstrates that the viscous displacement, its rate and the SSH computed in FastIsostasy qualitatively follow the results of Spada et al. (2011). The latter corresponds to the outputs of PMTF, VILMA (Martinec, 2000) and VEENT, which show such good agreement that they are gathered into a single output. Quantitatively, the mean displacement error between FastIsostasy and Spada et al. (2011) is small, with $\bar{e}^{\mathrm{abs}}(t) < 27.0\,\mathrm{m}$ for both loading cases. In addition, the equilibrium displacement is well represented with a maximal error of $\hat{e}^{\mathrm{abs}}(t = 100\,\mathrm{kyr}) < 7\,\mathrm{m}$ and $\hat{e}^{\mathrm{abs}}(t = 100\,\mathrm{kyr}) < 12\,\mathrm{m}$ and a mean error of $\bar{e}^{\mathrm{abs}}(t = 100\,\mathrm{kyr}) < 5\,\mathrm{m}$ and $\bar{e}^{\mathrm{abs}}(t = 100\,\mathrm{kyr}) < 7\,\mathrm{m}$, for the cap and the disc load respectively. The maximal difference arises at $t = 10\,\mathrm{kyr}$ for the cylindrical load and yields $\hat{e}^{\mathrm{abs}} < 47\,\mathrm{m}$, i.e. $\hat{e} < 0.19$. In both cases, the difference in vertical displacement is propagated to the computation of the SSH perturbation according to Equation 19. As shown in the last row of Fig. 7, this leads to a maximal difference between FastIsostasy and the 1D GIA models that reaches at most $6\,\mathrm{m}$ but less than $2\,\mathrm{m}$ on average, for maximal SSH perturbations around $40\,\mathrm{m}$ in both models.

Since the experimental setup is as similar as possible for the 1D GIA models and FastIsostasy, these differences can be largely attributed to (1) the lumping of the depth dimension as performed in Eq. 4, which leads the two approaches to solve different equations and (2) to the regional domain used here, which only allows an approximate treatment of the BCs as described in Equation 15.

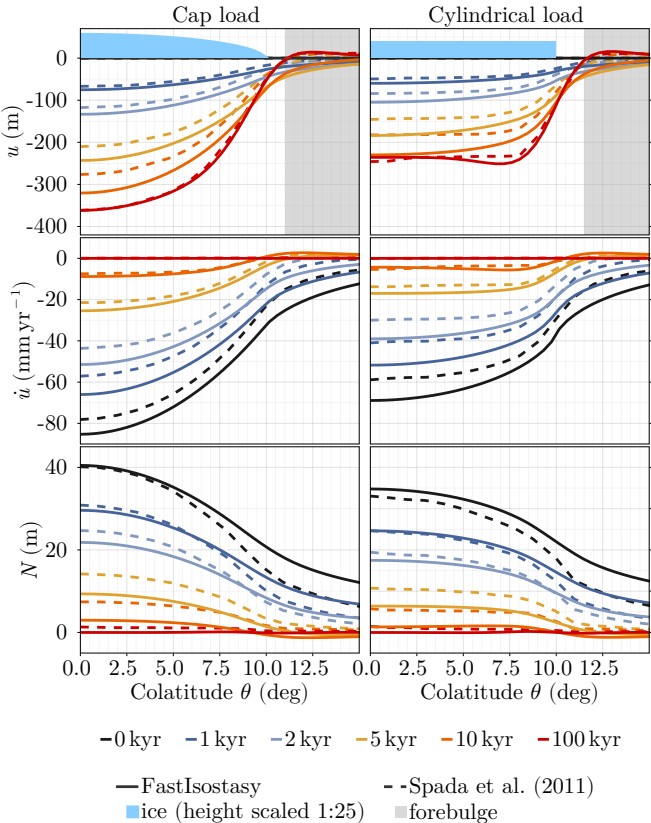

**Figure 7.** Comparison of FastIsostasy and Spada et al. (2011) on tests "1/2" and "2/2". From top to bottom: viscous displacement $u$, viscous displacement rate $\dot{u}$ and resulting SSH perturbation $N$. From left to right: cylinder and cap of ice applied as load.

### 4.3 Test 3 – 3D GIA solution of idealised load on idealised LV-Earth

Seakon is a global 3D GIA model that includes all the processes mentioned in Section 1.4.6. It solves the deformational response of the solid Earth with FVM on an unstructured grid, which is typically finer at the poles, where the bedrock displacement is largest. It has been extensively used in GIA studies (e.g. Austermann et al., 2021; Mitrovica et al., 2009; Pan et al., 2021, 2022) and coupled to an ice-sheet model in Gomez et al. (2018). For further details about the model and the adaptive grid refinement, we refer the reader to Latychev et al. (2005) and Gomez et al. (2018). We here benchmark FastIsostasy against Seakon for idealised cases with LV similar to that estimated across Antarctica. Here again, a cylindrical ice load with $H = 1\,\mathrm{km}$ and $R = 1000\,\mathrm{km}$ is applied on a domain with $W_x = W_y = 3000\,\mathrm{km}$ and $N_x = N_y = 128$. To isolate the error from the lumping of the depth dimension, the vertical structure of the solid Earth is kept as simple as possible, with a mantle viscosity that is constant along $z$.

We distinguish four cases (a-d), which are all parametrised by a Gaussian-shaped anomaly that is almost zero on the boundary and yields its largest value at the interior of the domain. For case (a) (case b), this anomaly represents a decrease (increase) from $T = 150\,\mathrm{km}$ down to $T = 50\,\mathrm{km}$ (up to $T = 250\,\mathrm{km}$) of the lithospheric thickness towards the interior of the domain. For case (c) (case d), this anomaly represents an exponential decrease (increase) from $\eta = 10^{21}\,\mathrm{Pa\,s}$ down to $\eta = 10^{20}\,\mathrm{Pa\,s}$ (up to $\eta = 10^{22}\,\mathrm{Pa\,s}$) of the mantle viscosity towards the interior of the domain. The heterogeneities (a-d) are shown in Appendix D and are used to generate results that are refered to by "LV-ELVA". To quantify the improvement resulting from the use of LV-ELVA instead of ELVA (or ELRA), we also generate results with the nominal, homogeneous parameters $T(x,y) = 150\,\mathrm{km}$ and $\eta(x,y) = 10^{21}\,\mathrm{Pa\,s}$ (or $\tau = 3{,}000\,\mathrm{yr}$) and index them with "ELVA" (or "ELRA").

As can be seen in the top and middle row of Fig. 8, LV-ELVA closely follows Seakon on cases (a-b) by showing similar time scales, amplitudes and shapes of the bedrock displacement. In the bottom row of Fig. 8, the maximal and mean relative differences respectively remain at $\hat{e}(t) < 0.07$ and $\bar{e}(t) < 0.03$ over time. In comparison, ELVA yields similar errors for case (a) and slightly higher in case (b), with values of $\hat{e}(t) < 0.12$ and $\bar{e}(t) < 0.05$. For ELRA, the maximal error over time is slightly higher, although not significantly. We recall that the lithospheric thickness is an important control on the shape of the bedrock displacement but only an indirect one on its magnitude and time scale. This can be seen in Eq. 9, where the lithospheric rigidity $D$ is only multiplied with spatial derivatives of the displacement. Misrepresenting the LV of lithospheric thickness therefore only has a marginal effect for cases (a-b), but we emphasise that its impact on the displacement magnitude can however become important when the load presents localised features, as later shown in Test 4. Furthermore, accounting for a heterogeneous lithospheric thickness can impact the bedrock slopes significantly, which are an important control on ice-sheet grounding-line stability and therefore on the evolution of ice sheets. Finally, as shown in an additional experiment presented in Appendix D, (LV-)ELVA yields equally low error values in the absence of a lithosphere. This is the extreme case of a thin lithosphere, where the absence of flexural moments effectively decouples the displacement of neighbouring cells - a behaviour that is present in both Seakon and (LV-)ELVA.

The advantage of using LV-ELVA over ELVA and ELRA becomes significant when studying (c-d). For these cases, ELVA yields large transient differences compared to Seakon, with $\hat{e}(t) < 0.37$ and $\bar{e}(t) < 0.11$. Here again, ELRA shows marginally higher error values. This clearly shows that neither ELRA nor ELVA are suited to represent the typical variations of viscosity over Antarctica. In comparison, LV-ELVA yields errors of only $\hat{e} < 0.11$ and $\bar{e} < 0.04$, similar to those obtained on tests (a-b). Since these values are systematically

lower than what was obtained in Test 2, it appears that the error of LV-ELVA mainly stems from the layered Earth structure that can only be partially accounted for on a 2D grid, and not from the LV generalisation presented in Equation 9. This is further supported by an additional test, presented in Appendix D, where Seakon and LV-ELVA adopt a 1D Earth structure following PREM. This leads to errors of $\hat{e} < 0.16$ and $\bar{e} < 0.06$, which is very close to what was observed in Test 2.

For cases (a-b), it should be noted that ELRA, ELVA and LV-ELVA present the same equilibrium state because of the constant lithospheric thickness, as can be formally deduced by setting the time derivative to $0$ in the equations presented in Appendix A. We stress that the higher transient error of ELVA and ELRA can therefore be easily missed when considering equilibrium states. This is the case in Le Meur and Huybrechts (1996), where the only spatial comparison across models is made for a quasi-equilibrium state. We further draw attention to the fact that the transient error metrics used in the present study are are stricter than plotting the mean spatial displacement of each model over time, which is done in Le Meur and Huybrechts (1996) and which potentially hides large localised differences between ELRA and the 1D GIA model used for comparison.

Throughout Test 3, LV-ELVA underestimates the peripheral forebulge by about $[10, 15]\,\mathrm{m}$, which is a systematic error. When considering a layered Earth, as shown in Appendix D, this value becomes as big as $40\,\mathrm{m}$ for early time steps but evolves to less than $20\,\mathrm{m}$ at equilibrium. Konrad et al. (2014) observed a similar behaviour when comparing ELRA to a 1D GIA. This comparatively large transient is the source of the aforementioned error of $\bar{e}(t) < 0.16$. This most likely arises because the mantle flow contributing to the amplitude of the forebulge is not resolved in LV-ELVA. Since the forebulge forms in the vicinity of the ice margin, this might be an important error source to keep in mind when comparing FastIsostasy to a 3D GIA model in a coupled ice-sheet context, especially when studying the possibility of a forebulge feedback as proposed in Albrecht et al. (in revision).

When performing Test 3, we noticed that a large lithospheric thickness, a large gradient of the lithospheric thickness, or a low viscosity all lead to a higher computational cost. This is consistent with theoretical insights, since all of these cases lead to a larger value of the right-hand side in Eq. 14, thus making the ODE stiffer and requiring smaller time steps to resolve this with sufficient accuracy in time. For quantitative values, we refer the reader to the comparative table of runtimes provided in Appendix D.

## 4.4 Test 4 – 3D GIA solution of the last glacial cycle on a realistic Earth

So far, FastIsostasy has been tested with idealised loads and parameter fields. We now consider the more realistic case of

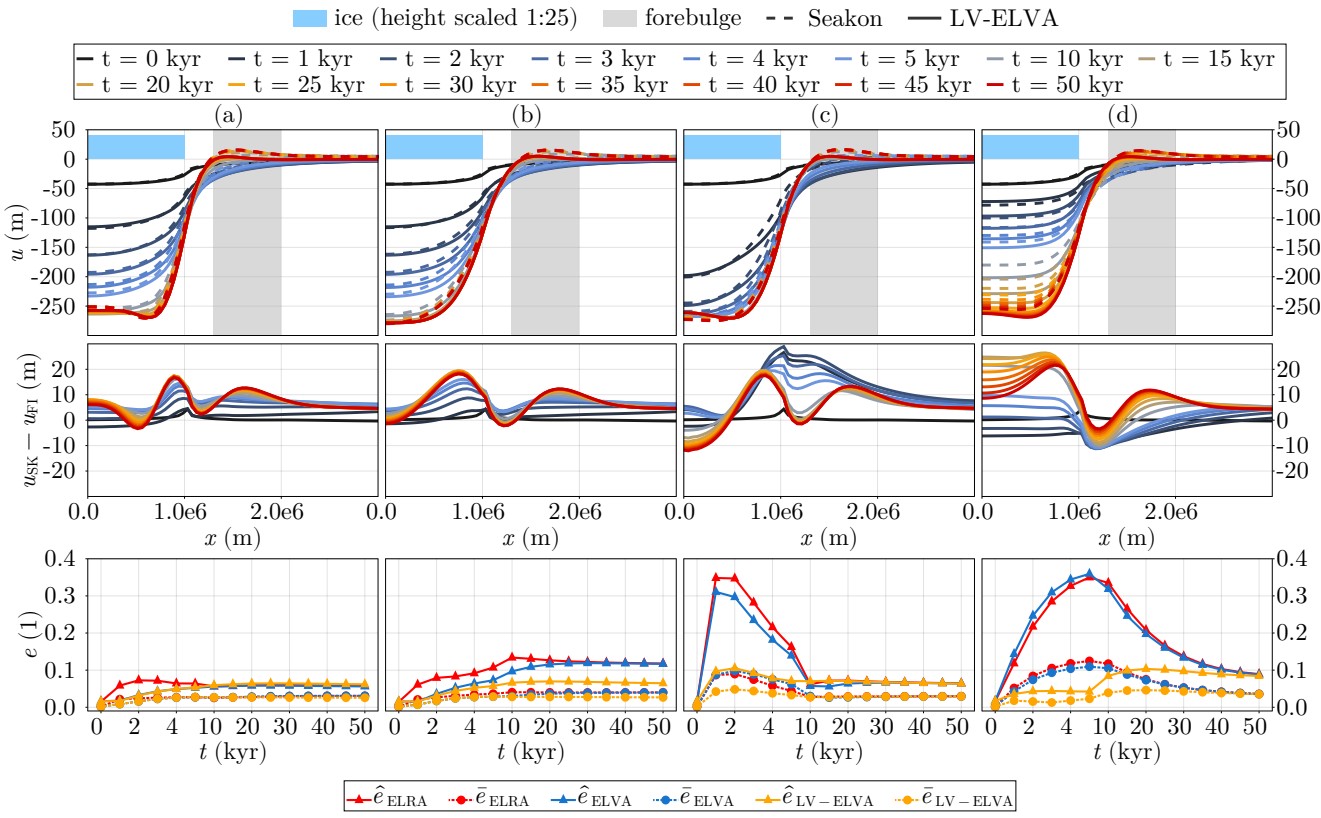

**Figure 8.** Comparison of FastIsostasy and Seakon for (a-b) heterogeneous lithospheric thickness and (c-d) upper-mantle viscosity. (Top row) cross section of the domain along the $x$ dimension, displaying the displacement of both models and (middle row) the corresponding difference. (Bottom row) Transient evolution of the mean and maximal relative errors of ELRA, ELVA and LV-ELVA compared to Seakon.

simulating the GIA response of two different Earth structures to the last glacial cycle, as reconstructed in ICE6G_D (Peltier et al., 2018), an updated version of ICE6G_C (Argus et al., 2014; Peltier et al., 2015) after a mismatch with the present-day uplift was pointed out in Purcell et al. (2016). The first structure is a 1D Earth that does not present any LV. The second structure is a 3D Earth with the lithospheric thickness and the mantle viscosity fields from Pan et al. (2022), which are similar to those depicted in Figure 2. We now compare the results of 5 different models: ELRA, ELVA, LV-ELVA, Seakon 1D (SK1D) and Seakon 3D (SK3D). It should be noted that the present comparison omits LV-ELRA, since its implementation goes beyond the scope of the present work.

For the regional models to be comparable with each other, ELRA, ELVA and LV-ELVA are coupled to ReSeLeM, which uses a BSL forcing derived from SK3D instead of Equation 30, in accordance with the comment made in Section 2.5. To perform the lumping of the depth dimension for LV-ELVA, we define the viscous half-space to begin at 300 km. We observed this to yield lower error metrics than $\{400, 500\}$ km, which appears coherent since, according to Eq. 4, deeper models might overestimate the contribution of

the deeper layers of the mantle to the effective viscosity. We recall that the effective viscosity should primarily capture the response to loads with characteristic lengths of continental ice sheets and that deeper layers are mostly excited by loads with larger wavelengths.

The error plot shown in panel (a) of Fig. 9 depicts, for all time steps, the displacement of SK3D, considered to be closest to reality, against ELRA, ELVA, SK1D and LV-ELVA. We hereby only represent the points within the activation mask $\mathcal{A}$, represented by the black contours of panels (c-h) and corresponding to the typical region of interest for ice-sheet modellers. The position around the identity shows that ELVA leads to displacements that are biased towards lower values, especially for $u_{SK3D} \leq -300\,\mathrm{m}$ where the error comes close to $e^{\mathrm{abs}} \simeq 130\,\mathrm{m}$. Although this bias is somewhat smaller for SK1D, it still reaches similar maximal values. In comparison, LV-ELVA is centred around the identity and presents no such bias. This can be explained by the fact that a thinner lithosphere and a less viscous mantle in West-Antarctica allows for larger transient displacements around LGM. Furthermore, the spread around the identity, especially around the lower, unbiased displacement values, is an additional

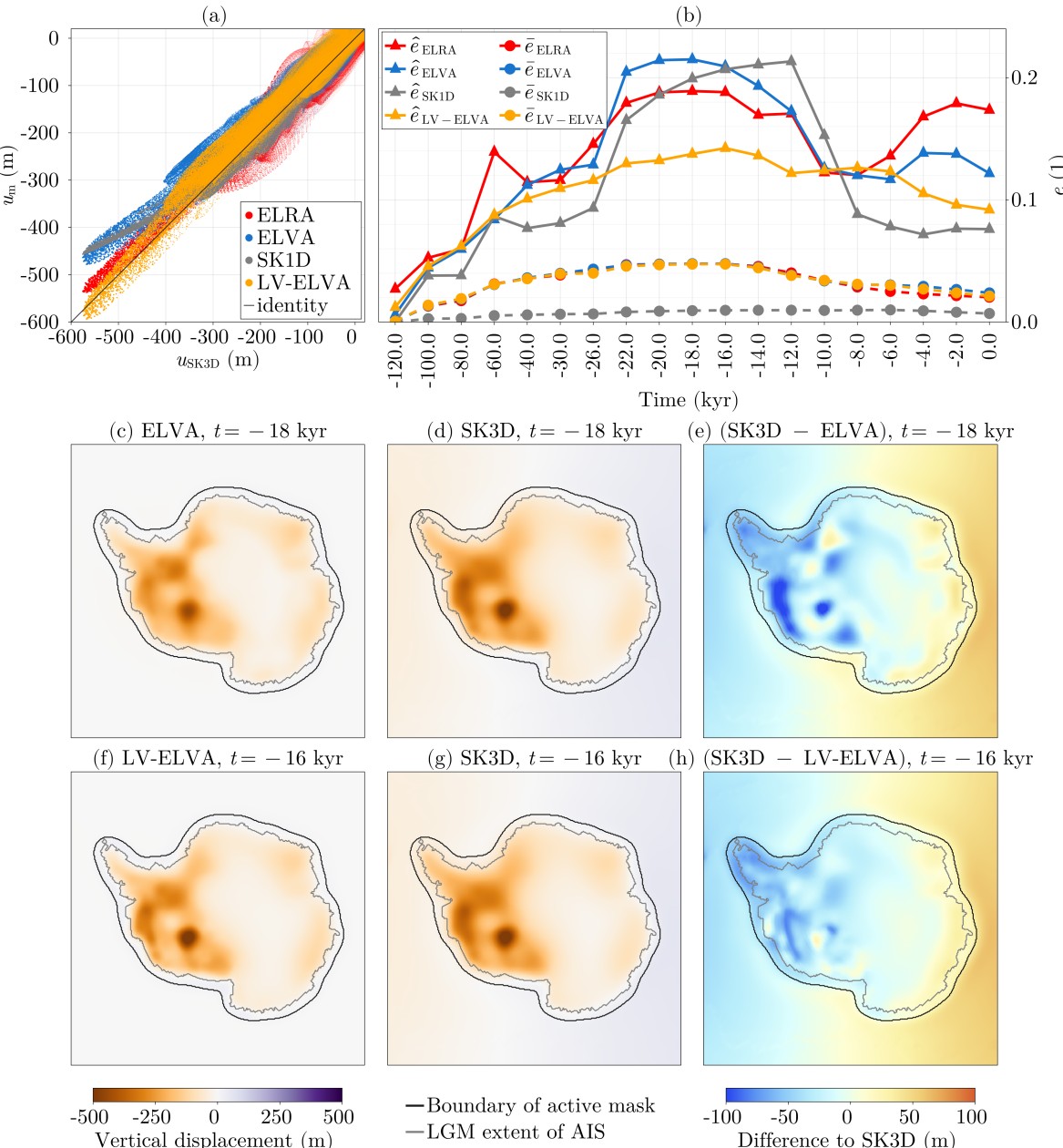

**Figure 9.** Comparison of Seakon and FastIsostasy, forced by the ice loading from ICE6G_D (Peltier et al., 2018). (a) Displacements at all time steps of ELRA, ELVA, SK1D and LV-ELVA against SK3D for cells within the active mask. (b) Transient mean and maximal displacement errors, respectively denoted by $\bar{e}(t)$ and $\hat{e}(t)$, of ELRA, ELVA, SK1D and LV-ELVA with respect to SK3D, for all domain cells. (c-e) Displacement of ELVA, SK3D and their differences for the time step of maximal error. (f-h) Same as (c-e) for LV-ELVA.

metric to take into account. For $u_{\mathrm{SK3D}} \geq -300\,\mathrm{m}$, SK1D, LV-ELVA and ELVA respectively present the smallest, the intermediate and the largest spread. As expected, ELRA shows a larger bias than LV-ELVA but, surprisingly, a smaller one than SK1D and ELVA. However, the spread of ELRA is larger than for any other model.

Panel (b) depicts the mean and maximal relative difference for ELRA, ELVA, LV-ELVA and SK1D with respect to SK3D. The mean and maximal value respectively relate to the spread around the identity and the bias observed in panel (a). In panel (b), the error metrics are computed for the full domain and therefore include the far field, where the rotational feedback dominates the displacement. Since none of the regional models is capable of capturing this, SK1D shows the smallest mean error over time, with $\bar{e}_{\mathrm{SK1D}}(t) < 0.01$. In comparison, ELRA, ELVA and LV-ELVA show larger mean error metrics that are however similar among each other, with $\bar{e}(t) < 0.04$. When computing the mean error only over the active mask, this difference between SK1D and the regional models vanishes, with $\bar{e} \leq 0.01$ for all models. We believe that this latter definition of the mean error is closer than what is relevant to most ice-sheet modellers but highlighting the larger error of regional models in the far field ensures that the users of FastIsostasy are aware of this limitation. The low values of the mean error that are observed regardless of the model can be explained by the fact that most of the regions, especially the bulk of East Antarctica, can be well represented by an intermediate mantle viscosity and lithospheric thickness.

Interestingly, the peak values of $\hat{e}_{\mathrm{SK1D}}$ and $\hat{e}_{\mathrm{ELVA}}$ are very close to each other and yield about $0.22$, corresponding to $\hat{e}^{\mathrm{abs}} \simeq 130\,\mathrm{m}$. In both cases, these values are observed over $t \in [-22, -12]\,\mathrm{kyr}$, which correspond to the $10\,\mathrm{kyr}$ of rapid deglaciation following LGM. In comparison, ELRA presents a peak maximal error with a similar timing and a marginally smaller amplitude $\hat{e}_{\mathrm{ELRA}} < 0.19$. However, it presents large errors over the last $6\,\mathrm{kyr}$ of simulation. This points out that high errors of ELRA, ELVA and SK1D are to be expected when rapid changes of ice thickness occur - a situation that could be triggered by sustained anthropogenic climate warming. For ELVA, the peak difference to SK3D is observed at $t = -18\,\mathrm{kyr}$. The corresponding displacement fields and their difference are plotted in (c-e), corroborating that ELVA does not allow enough displacement in West-Antarctica after the LGM. The location of the peak error points out that the use of ELVA, coupled to an ice sheet model, may lead to significant errors since the WAIS is based on a retrograde bedrock below sea level and is therefore particularly sensitive to the GIA response, as pointed out in the introduction.

Compared to ELRA, ELVA and SK1D, LV-ELVA reduces the maximal error down to about $\hat{e}_{\mathrm{LV-ELVA}} \leq 0.14$, which corresponds to $\hat{e}^{\mathrm{abs}} \leq 80\,\mathrm{m}$. The displacement fields for the time $t = -14\,\mathrm{kyr}$ are plotted in panels (f-h) and show that the near-field displacement is reasonably well matched, even for the time step showing the largest error metrics. In particular,

the displacement is slightly underestimated in most of the active mask. Since this appears to be a systematic offset, it could easily be corrected by tuning the density and/or the viscosity chosen for LV-ELVA[8]. We however decide not to do so to highlight the differences to SK3D without additional tuning.

Throughout Test 4, SK3D has been used as the baseline model. In Appendix D, we show a similar analysis that compares ELRA and ELVA by using SK1D as baseline. There it appears (1) that ELVA presents smaller errors than ELRA and (2) that LV-ELVA presents smaller errors compared to SK3D than ELVA compared to SK1D.

**Run-time analysis.** Despite the errors compared to Seakon, and potentially to any 3D GIA model, we believe that FastIsostasy can be a particularly appealing tool, since the $122\,\mathrm{kyr}$ run takes only about 9 minutes to compute for a horizontal resolution of $h = 20\,\mathrm{km}$, resulting in $350 \times 350$ grid points. For ELVA, the absence of LV leads to a reduced computation time of only about 4 minutes and outperforms ELRA, which requires about 6 minutes. These computations were performed on an NVIDIA GeForce RTX 2070, a low-cost GPU with moderate performance by the standards of 2024. Although the time stepping is adaptive, no values beyond $dt = 10\,\mathrm{yr}$ are used. This potentially provides the ice-sheet model with an input that is very dense in time, for instance as opposed to Gomez et al. (2018).

In comparison, the Seakon simulation takes about 4.5 days on 150 CPUs with a time step of $dt \in [125, 1000]\,\mathrm{yr}$. Assuming an ideal parallelisation scaling of 100%, this corresponds to about a million minutes of CPU-runtime, which roughly corresponds to five orders of magnitude more than what FastIsostasy requires. The models used in Albrecht et al. (in revision) and Zhong et al. (2022) present smaller computation times, reducing this order of magnitude down to 4.

Furthermore, the power consumption of the GPU used in the present study is of $185\,\mathrm{W}$[9], compared to a typical value of more than $100\,\mathrm{W}$ for a single, modern CPU. As the energy consumption is expressed as the product of power and computation time, FastIsostasy appears to be less energy-consuming than Seakon by several orders of magnitude. Finally, we draw attention to the fact that the acquisition price of the GPU used here is a few hundred euros, which is far less than that of a large CPU cluster.

Of course, the run-time of Seakon and FastIsostasy are not directly comparable: Seakon solves the global GIA problem, which requires a grid with many more degrees of freedom that reaches down to the core-mantle boundary. The output of Seakon is much richer, since it includes, among others, the rotational feedback, the position of migrating shorelines,

---

[8]This could simply be done by hand or, for instance, with an unscented Kalman inversion, as shown in the code documentation.

[9]Tabulated value found at https://www.nvidia.com/es-es/geforce/graphics-cards/compare/?section=compare-20

the horizontal displacement of the bedrock and the relative sea level at any point on Earth. Nonetheless, these quantities tend to be less relevant for stand-alone ice-sheet modelling, and FastIsostasy therefore offers an opportunity to regionally mimic the behaviour of a 3D GIA model at very low computational, energy and financial cost.

## 5    Conclusions

Throughout all the tests, FastIsostasy displays a maximal and mean error over space of $\hat{e}(t) < 0.2$ and $\bar{e}(t) < 0.05$, both being typically much smaller for most time steps. In particular, Test 1 has shown that the discretisation error of FastIsostasy is very small and that, with increasing problem size, its computational expense scales better than what is typically obtained with FDM, FVM and FEM solvers. Test 2 has shown that FastIsostasy represents the deformational response and the SSH perturbation with a relatively low error compared to 1D GIA models, despite solving the problem on a 2D regional grid. Test 3 and Test 4 have shown that LV-ELVA produces greatly reduced errors with respect to SK3D, compared to ELRA and ELVA, and even to SK1D for the near-field GIA response. This means that the model uncertainty between FastIsostasy and Seakon is smaller than the upper bound on parametric uncertainty, given by the difference between a 1D and a 3D Earth structure (Albrecht et al., in revision).

In conclusion, FastIsostasy can greatly reduce the transient error of bedrock displacement compared to ELRA, ELVA and even to 1D GIA models for regions of significant LV. This was achieved by introducing LV-ELVA, which generalises the work of Bueler et al. (2007) and Coulon et al. (2021) and was coupled to ReSeLeM, which regionally approximates the transient changes in ocean load. Whereas the differences between FastIsostasy and global GIA models are within the range of parametric uncertainties, the computation time is typically reduced by 3 to 5 orders of magnitude. For most ice-sheet models, FastIsostasy can thus represent a leap in GIA comprehensiveness at very low computational cost, even for high-resolution runs on the time scale of glacial cycles. This has straight-forward applications, since the GIA response is particularly relevant for the many marine-based regions of the AIS, where significant LVs of the solid Earth can be present. This is the case for the Amundsen sector, which could become the largest source of sea-level rise in the coming century and therefore requires projections that account, as accurately as possible, for the stabilising effect of GIA.

Since fields of the lithospheric thickness and the upper-mantle viscosity can be easily found in literature, FastIsostasy reduces the difficulty of creating meaningful ensembles compared to relaxed rheologies (Coulon et al., 2021). The very short runtime of FastIsostasy offers an efficient method of propagating the uncertainties of the solid-Earth parameters to past and future climatic scenarios.

We believe that GIA modellers, as well as the few ice-sheet models that are coupled to a 3D GIA model, can benefit from FastIsostasy, since it can be used as a fast-prototyping tool. In particular, a scheme to tune the parameters of FastIsostasy can turn it into an emulator of a 3D GIA model with better interpretability than, for instance, machine learning techniques. Nonetheless, it should be emphasised that some scientific questions can only be answered with a global 3D GIA model and that FastIsostasy is a complementary tool that does not aim to replace it. Finally, we believe that the relatively abbreviated code of FastIsostasy and its few equations compared to 1D or 3D GIA models are particularly well suited for educational purposes.

## Appendix A:  From ELRA to LV-ELVA

Following Le Meur and Huybrechts (1996), the governing equations of ELRA yield:

$$\rho^{\mathrm{m}} g u^{\mathrm{eq}} + D \nabla^4 u^{\mathrm{eq}} = \sigma^{zz}, \qquad (A1)$$

$$\frac{\partial u}{\partial t} = \frac{1}{\tau}(u - u^{\mathrm{eq}}) \qquad (A2)$$

with $u^{\mathrm{eq}}$ the equilibrium displacement and all further quantities already introduced in this paper. Equation A1 can be solved by convolving the load with a Green's function derived from a Kelvin function of order 0, as described in Le Meur and Huybrechts (1996). In FastIsostasy, the convolution is performed via FFTs, which is much faster than a computation in time domain. Coulon et al. (2021) generalised the equations of ELRA to a laterally-variable lithospheric thickness and relaxation time (LV-ELRA):

$$\rho^{\mathrm{m}} g u^{\mathrm{eq}} + \frac{\partial^2 M_{xx}^{\mathrm{eq}}}{\partial x^2} + 2\frac{\partial^2 M_{xy}^{\mathrm{eq}}}{\partial x \partial y} + \frac{\partial^2 M_{yy}^{\mathrm{eq}}}{\partial y^2} = \sigma^{zz}, \quad (A3)$$

$$\frac{\partial u}{\partial t} = \frac{1}{\tau(x,y)}(u - u^{\mathrm{eq}}), \qquad (A4)$$

with $M_{xx}^{\mathrm{eq}}$, $M_{xy}^{\mathrm{eq}}$ and $M_{yy}^{\mathrm{eq}}$ the flexural moments at equilibrium, computed by introducing $u^{\mathrm{eq}}$ into Equations 10-12. For ELRA as well as LV-ELRA, a relaxed rheology is used, which presents the limitations explained in Section 1.3. ELVA, as proposed by Bueler et al. (2007); Cathles (1975); Lingle and Clark (1985), addresses this limitation because its governing equation is directly parametrised by viscosity:

$$2\eta |\nabla| \frac{\partial u}{\partial t} + \rho^{\mathrm{m}} g u^{\mathrm{eq}} + D\nabla^4 u = \sigma^{zz}. \qquad (A5)$$

However, this assumes a constant lithospheric thickness $T(x,y) = T$ and sub-lithospheric mantle viscosity $\eta(x,y) = \eta$ throughout the domain, which therefore prevents the representation of LV. We tried to generalise the derivation presented in Cathles (1975) to LV, however unsuccessfully. Instead, we combined Eq. A3 and Eq. A5, thus obtaining:

$$2\eta(x,y)|\nabla|\frac{\partial u}{\partial t} + \rho^{\mathrm{m}} g u^{\mathrm{eq}} + \frac{\partial^2 M_{xx}}{\partial x^2} + 2\frac{\partial^2 M_{xy}}{\partial x \partial y} +$$

$$\frac{\partial^2 M_{yy}}{\partial y^2} = \sigma^{zz}, \tag{A6}$$

with the flexural moments accounting for the laterally-variable lithospheric thickness. By introducing a pressure term, as done in Eq. 8, and accounting for the distortion $K$, we obtain the governing equation of LV-ELVA, as written in Equation 9. For constant parameters, LV-ELVA simplifies to the equation of ELVA and can be therefore understood as an ad-hoc generalisation that reduces the error made compared to a 3D GIA model. Using ELVA in FastIsostasy is simply achieved by running LV-ELVA with constant parameters. This seamless approach offers a code that is easier to maintain and an interface that is simpler to use, at the expense of a minor increase in computational expense compared to the use of a specific solver that takes advantage of the simplifications made in ELVA.

## Appendix B: Ocean surface function

The function $\tilde{A}(s) : \mathbb{R} \to \mathbb{R}$ is here computed by summing the surfaces of cells situated below the BSL, $s$, based on the 1 arc-minute global topography of ETOPO1 (Amante and Eakins, 2009). Note that this slightly overestimates the ocean surface, since all regions below sea-level are counted as part of the ocean, including, for instance, parts of the Netherlands. To tackle this, we introduce a bias correction scaling $\gamma$, which avoids any offset for the present-day value $A_{\mathrm{pd}}$ and depends on the uncorrected value $\tilde{A}(s)$:

$$A(s) = \gamma \tilde{A}(s), \qquad \text{with}: \quad \gamma = \frac{A_{\mathrm{pd}}}{\tilde{A}(s=0)}. \tag{B1}$$

To reduce the runtime, we precompute $A(s)$ as a piecewise-linear interpolator for $s \in [-150, 70]\,\mathrm{yr}$ with a discretization of $\Delta s = 0.1\,\mathrm{m}$. The resulting function is depicted in the bottom-left panel of Fig. 4 and shows that, for the range of realistic sea-level contributions over glacial cycles, the trapezoidal approximation leads to variations of the ocean surface between -7% and +4% around the present-day value.

## Appendix C: Scaling the effective viscosity

Two important characteristics of the mantle have to be accounted for, such that the Maxwell time $\tau = \eta E^{-1}$ of FastIsostasy is comparable to that of a 3D GIA model. This is done by introducing two correction factors. First, one of the underlying assumptions made by Cathles (1975) is that the mantle is incompressible, i.e. $\nu^{\mathrm{i}} = 0.5$ is assumed as opposed

to $\nu_0 = 0.28$ for the elastic lithosphere. In reality, the mantle is however a compressible medium with $\nu^{\mathrm{c}} \simeq 0.28$. We now look for $\eta^{\mathrm{i}}$, the viscosity that has to be used in the incompressible case in order to match the Maxwell time of the compressible case. By introducing the shear modulus $G = E\left(2\left(1+\nu\right)\right)^{-1}$, we obtain:

$$\frac{\eta^{\mathrm{i}}}{2\,G\left(1+\nu^{\mathrm{i}}\right)} \quad = \quad \frac{\eta^{\mathrm{c}}}{2\,G\left(1+\nu^{\mathrm{c}}\right)} \tag{C1}$$

$$\Leftrightarrow \eta^{\mathrm{i}} \quad = \quad \frac{1+\nu^{\mathrm{i}}}{1+\nu^{\mathrm{c}}}\,\eta^{\mathrm{c}} = \alpha^{\mathrm{c}}\,\eta^{\mathrm{c}}. \tag{C2}$$

In essence, this means that compressible media have a larger Maxwell time and that we need to slightly increase the viscosity values to render this, since Eq. A5 used to postulate Eq. 9 assumes an incompressible viscous flow. This is supported by Fig. C1, which shows that a 1D GIA model displays longer decay times for a compressible mantle, compared to an incompressible one.

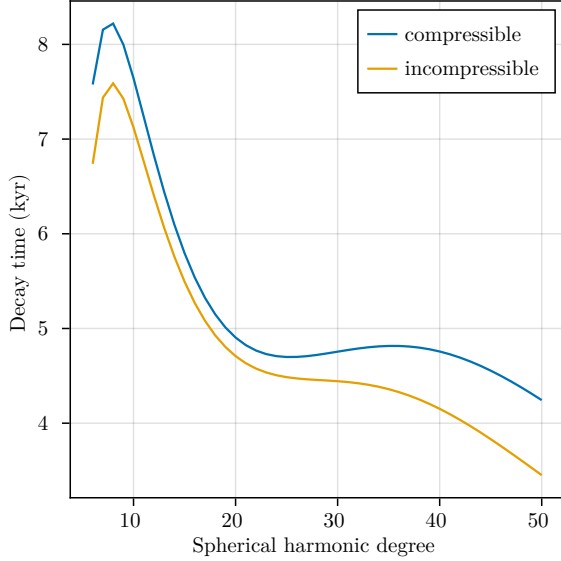

**Figure C1.** Decay times of a 1D GIA model with $\nu^{\mathrm{c}} = 0.28$ (compressible) and $\nu^{\mathrm{i}} = 0.5$ (incompressible).

Second, both the shear modulus and the viscosity depend on the temperature of the medium. For instance, a positive temperature anomaly in the mantle leads to a negative anomaly of both viscosity and shear modulus. This means that the decrease of the Maxwell time due to the decrease of viscosity is somewhat compensated by the decrease in shear modulus. We have chosen to compute this scaling by calibrating FastIsostasy to results of a 3D GIA model:

$$\eta = \exp\left(\log_{10}\left(\frac{\eta_0}{\eta^{\mathrm{i}}}\right)\right)\eta^{\mathrm{i}} = \alpha^{\mathrm{G}}(\eta^{\mathrm{i}}) \cdot \eta^{\mathrm{i}}, \tag{C3}$$

with $\eta_0 = 10^{21}\,\mathrm{Pa\,s}$ the calibration constant used throughout this work. We thus obtain a relation between the viscosity

$\eta^{\mathrm{c}}$, inferred from seismic measurements, and the corrected effective viscosity $\eta$, ultimately used in FastIsostasy:

$$\eta = \alpha^{\mathrm{G}} \alpha^{\mathrm{c}} \eta^{\mathrm{c}} = \alpha \eta^{\mathrm{c}}. \tag{C4}$$

If the depth dimension is lumped according to Eq. B1, then the viscosity field $\eta_1^{\mathrm{eff}}$, representing the compound of layers from $l = 1$ to $l = L$, is used for $\eta^{\mathrm{c}}$.

## Appendix D:  Complementary information on Test 3 & 4

The anomalies of lithospheric thickness and upper-mantle viscosity used in Test 3 are represented in Figure D1 and result from a scaled Gaussian distribution $\mathcal{N}(0, \sigma)$ with $\sigma = (W/4)^2 \boldsymbol{I}$, and $\boldsymbol{I} \in \mathbb{R}^{2 \times 2}$ the identity matrix.

In addition to Test 3, we perform two simulations with laterally-constant Earth structures in Seakon. The first one corresponds to PREM (Dziewonski and Anderson, 1981) and the second to a single-layer mantle without elastic lithosphere. The results are respectively depicted in columns (a) and (b) of Fig. D2. Unsurprisingly, the first case shows a very similar error pattern to what is obtained in Test 2 and highlights that the main error source of FastIsostasy comes from the lumped depth dimension rather than from the generalisation of ELVA to LV-ELVA. The second case shows that, in the absence of a lithosphere, the match between Seakon and FastIsostasy yields $\bar{e}(t) < 0.04$ and $\hat{e}(t) < 0.08$. In particular, this example shows that, in both models, the absence of a lithosphere effectively decouples neighbouring cells due to the absence of flexural moments.

Whereas Fig. 9 uses SK3D as baseline for the error metrics, we propose to use SK1D as a baseline to compare ELRA and ELVA in Figure D3. In Fig. D3, it appears that ELRA is more biased towards large displacements than ELVA. Furthermore, the mean error is similar for both models but the maximal error is overall higher for ELRA, with $\hat{e}(t) < 0.21$. In comparison, the maximum error of ELVA yields $\hat{e}(t) < 0.16$. The middle row of Fig. D3 highlights that the higher error of ELRA stems from an overestimated displacement in the Mary Byrd Land. In comparison, the bottom row shows that ELVA displays a much more homogeneously distributed error.

*Code and data availability.* FastIsostasy is available under GNU General Public License v3.0 at https://github.com/JanJereczek/ FastIsostasy.jl (Julia version) and https://github.com/palma-ice/ FastIsostasy (Fortran version). The data used in the present work can be found at https://github.com/JanJereczek/isostasy_data. The archived versions of the code and data used for this paper can be found at https://zenodo.org/doi/10.5281/zenodo.10419117 and https://zenodo.org/doi/10.5281/zenodo.10419334.

*Video supplement.* Animations of the results obtained by FastIsostasy on Test 4 can be found at https://github.com/JanJereczek/ FastIsostasy.jl.

*Author contributions.* J.S-J. conceptually developed FastIsostasy as well as its Julia version. The Fortran one was co-developed by M.M., A.R. and J.S-J. The Seakon simulations were performed by K.L., who, along with J.X.M. also provided insights on 3D GIA modelling for the purposes of this manuscript. This manuscript was prepared by J.S-J. with contributions from all co-authors.

*Competing interests.* The authors declare no competing interests.

*Disclaimer.*

*Acknowledgements.* We would like to thank Ed Bueler and Constantine Khroulev for the helpful email exchange. Their openly accessible implementations of ELVA, available at https://github.com/ pism and https://github.com/bueler/fast-earth, greatly eased the initial phase of FastIsostasy's development and inspired its name. We'd also like to thank Douglas Wiens, Ana Negredo and Javier Fullea for providing valuable comments and/or data. A particular thanks goes to the reviewers of this paper, who provided many helpful comments to improve this manuscript. Finally, J.S-J. wants to thank the whole PalMA team for providing their daily support and for creating a fantastic working environment.

*Financial support.* J.S-J. is funded by CriticalEarth, grant no. 956170, an H2020 Research Infrastructure of the European Commission. A.R. received funding from the European Union (ERC, FORCLIMA, 101044247). M.M. is funded by the Spanish Ministry of Science and Innovation (project MARINE, grant no. PID2020-117768RB-I00).

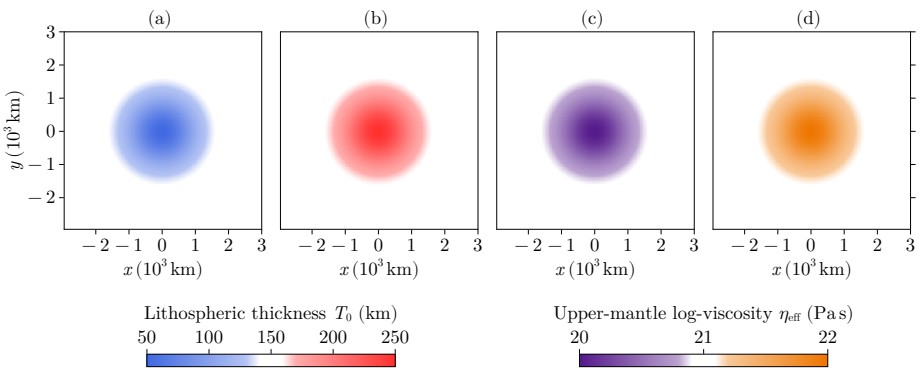

**Figure D1.** Gaussian-shaped LV used in Test 3 for (a) a lithospheric thinning, (b) a lithospheric thickening, (c) a viscosity decrease and (d) a viscosity increase towards the centre of the domain.

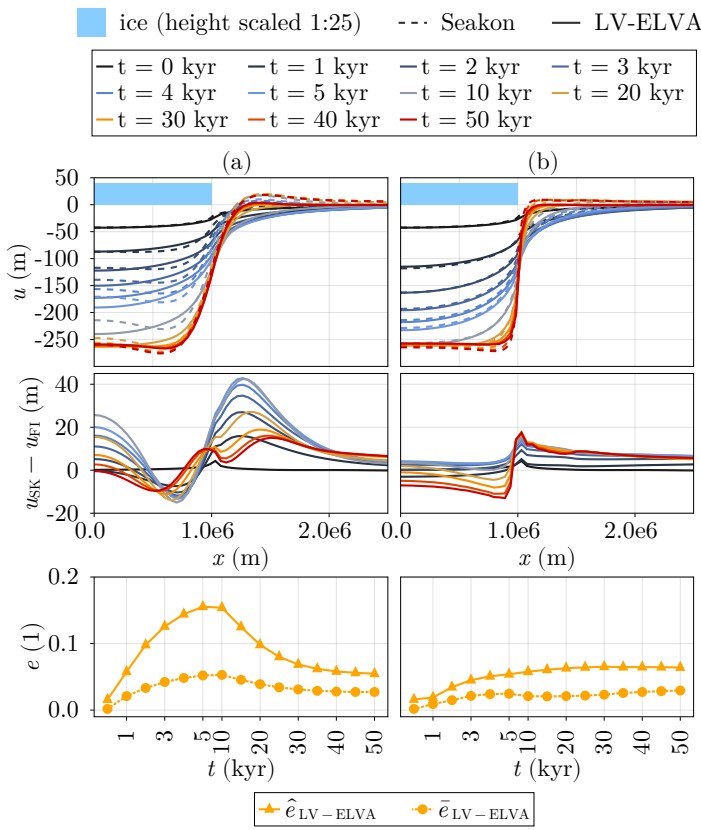

**Figure D2.** Comparison of FastIsostasy and Seakon for (a) a layered mantle following PREM and (b) a mantle with a single layer and no lithosphere.

| Case | Decreasing $T$ | Increasing $T$ | Decreasing $\eta$ | Increasing $\eta$ | PREM | $T = 0$ | ELRA |
|---|---|---|---|---|---|---|---|
| Runtime (s) | 35.6 | 40.7 | 33.8 | 25.0 | 24.1 | 24.9 | 14.2 |

**Table D1.** Runtime of FastIsostasy in Test 3 for a resolution of $N_x = N_y = 128$, on a single CPU (Intel i7-10750H 2.60GHz).

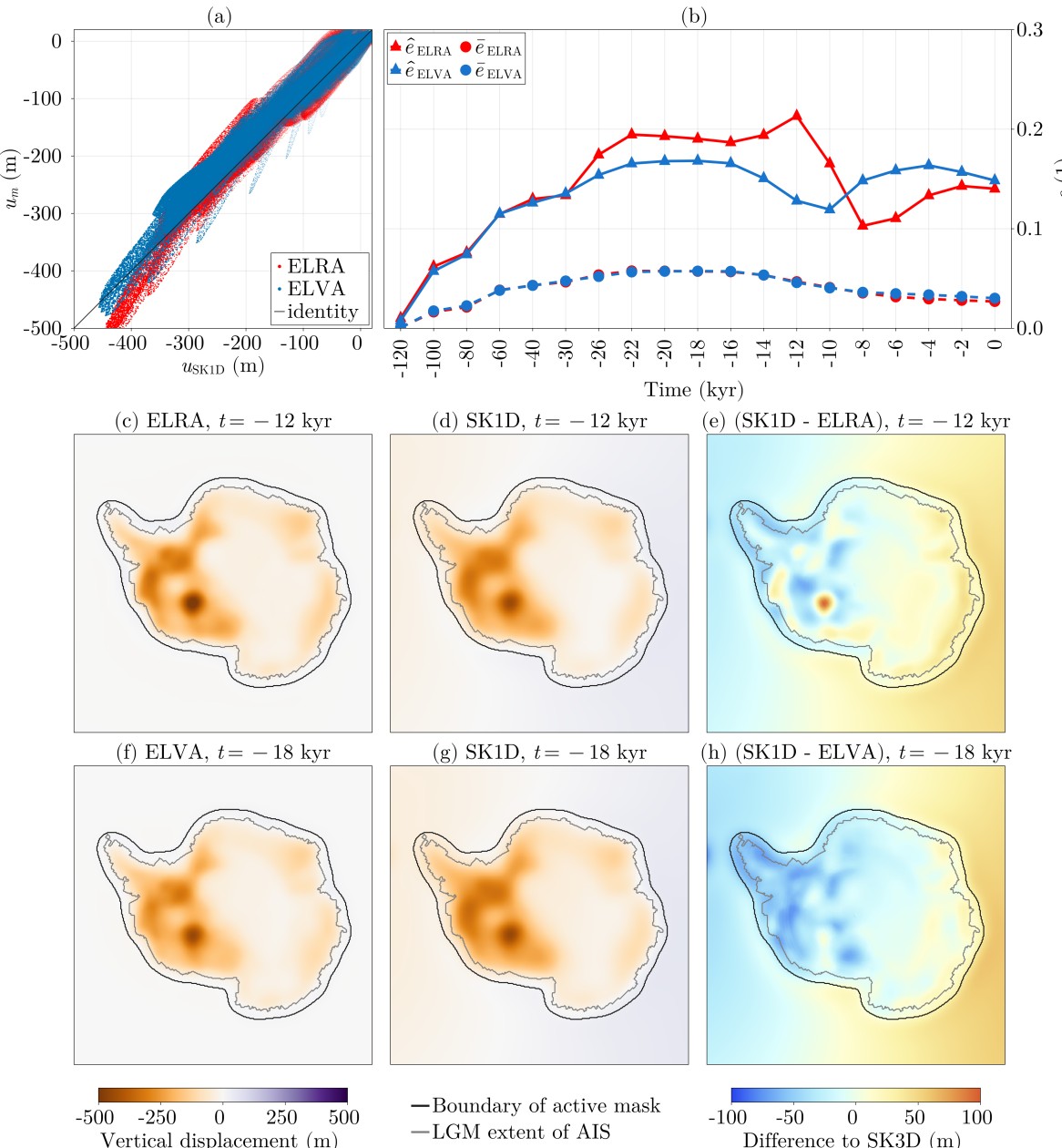

**Figure D3.** Comparison of ELRA, ELVA and SK1D forced by ICE6G_D (Peltier et al., 2018). (a) Displacements at all time steps of ELRA and ELVA against SK1D, for cells within the active mask. (b) Transient mean and maximal errors of ELRA and ELVA with respect to SK1D. (c-e) Displacement of ELRA, SK1D and their differences for the time step of maximal error. (f-h) Same as (c-e) for ELVA.

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
