# Peer review of "FastIsostasy v1.0 – A regional, accelerated 2D GIA model accounting for the lateral variability of the solid Earth"

_EGUsphere, 2023_

## Referee Comment (RC4)

**By Caroline van Calcar**

Swierczek-Jereczek et al. present a new GIA model, FastIsostacy, based on the Fourier collocation method that can include lateral viscosity and lithospheric thickness variations in the mantle. The new model is compared to the existing 3D self-gravitating visco-elastic Earth model Seakon. They show a maximum error of 0.2 over a glacial cycle when FastIsostacy is used compared to Seakon. Although the error is not negligible, FastIsostacy has the potential to be coupled to ice sheet models because the runtime is very short and the model is open source.

The manuscript is well written. The introduction provides a great overview of current existing methods used in GIA models. The second part of the introduction, "FastIsostasy in the model hierarchy", provides a clear overview of the different GIA models. Table 1 also includes which numerical scheme is used for each type of GIA model, but the numerical scheme are not explained in the text of the section "FastIsostasy in the model hierarchy". For this manuscript, the FDM/FCM and the numerical scheme of Seakon are particularly important and should be described in more detail.

The method section explains quite well the details of the new method developed but should elaborate more on the effect of simplifications in FastIsostasy in order to reduce computational cost compared to Seakon. Line 503 states that tuning can be done easily but doesn't explain what tuning would exactly be required. Also, line 537-539 states that the difficulty of creating meaningful ensembles is decreased but even though different viscosity fields are available, it is not shown how FastIsostasy performs using different viscosity fields. It is necessary to discuss the implication of the use of effective viscosity, the characteristic wavelength, and alfa. Equation (6) suggests that the alpha is selected to improve the fit to a 3D model, which means the results depend on a particular 3D GIA model. Are alpha and W fixed for all comparisons? That results in conclusions that are dependent on this choice, which should be emphasized. Tests for different alpha and W are required to have robust conclusions on the accuracy of FI3D.

FastIsostasy is presented in line 533-534 as a model that can greatly reduce the error of bedrock displacement compared to ELRA and ELVA and that is useful coupled to an ice sheet model due to the short computation time. However, there are no results presented in this manuscript that compare the performance of FastIsostasy with the most widely used GIA model in ice sheet modelling, the ELRA model and the laterally varying ELRA model so there could also be no conclusion about the reduction of the eror of FastIsostasy compared to ELRA. It is therefore not shown whether FastIsostasy is an improvement on what already exists. The focus of the paper should therefore be changed and the introduction rewritten with less focus on coupled ice sheet – GIA models, or a comparison with ELRA should be shown.

In line 361 the authors justify an error of 0.2. However, the error should ideally be smaller than the error introduced by different parametrizations. In some cases this error could be in the order of a hundred meters, which is significant and could have a large effect on ice dynamic models. In comparison, 1D GIA model benchmark study show much lower errors. Furthermore, from figure 8b it can be seen that the error between FastIsostasy 3D and Seakon 3D after 8 kyear is larger than the effect of 3D rheology itself. In that case, differences between FastIsostasy and global GIA models are outside of the range of parametric uncertainties, which contradicts the conclusion in line 534-535. The conclusions on the

performance of FastIsostasy should therefore be more considerate of the large error rather than accept them compared to an arbitrary standard, especially since the values hold for certain choices of the resolution.

**Specific comments**

Line 3-4: The impact of 3D GIA on ice sheet dynamics has only been shown for glacial cycles and not yet for projections. This sentence is suggesting that it has been shown. Please include in this sentence that the impact has only been shown over a glacial cycle.

Line 6: An iterative coupling scheme is required when simulating a glacial cycle but it hasn't been studied yet whether iterations are required to simulate projections. Projections have been performed without an iterative coupling scheme using a 1D GIA model and there are no published projections using a 3D GIA model. The need for an iterative coupling scheme is therefore not an argument why 3D GIA models are not used in ice sheet models. I would suggest to, instead, include that 3D GIA models are not used in ice sheet modelling because the effect of 3D GIA is not known, 3D GIA models are computationally expensive and the coupling scheme is complex to apply.

Line 12-13: Please include the value of error here instead of mentioning that the agreement is very good.

Line 15: The Fortran version is not provided yet, according to the data availability section. I suggest to include in this sentence that it will be provided.

Line 44-47: The impact of 3D GIA over a glacial cycle doesn't necessarily mean that the impact is also large over projections of a much shorter time scale. In multiple places in the introduction, the distinction between what has been studied over glacial cycles and what has been studied over projections is not clear. There are studies that show a significant effect of using 3D GIA compared to 1D GIA over a glacial cycle using a coupled ice sheet – GIA model (for example Gomez et al., 2018 & van Calcar et al., 2023) and from recent history till present day using a GIA model with a prescribed ice history (for example Blank et al., 2021). However, in this manuscript, results from Gomez et al. and van Calcar et al. are presented as if they show the impact of 3D GIA in projections as well, which is not the case. There are studies that show the importance of 3D GIA in projections using uncoupled models, such as Yousefi et al. (2022) but this study is not referenced in the manuscript. Currently, there are no publications on coupled ice sheet – 3D GIA models used for projections. This distinction should be made more clear throughout the introduction and the references to Yousefi et al. and Blank et al. should be added.

Line 47: Include that sea level contributions from the basins are 19.2 and 3.4 m at present day.

Line 58: Include references of the 3D GIA models that you are referring to.

Line 62-64: Whether or not the ice-sheet modelling community is well aware of the how important 3D GIA is, is subjective. There are only a few studies showing the importance of 3D GIA over a glacial cycle and in projections and there are no published studies simulating projections using coupled ice dynamic – 3D GIA models. It can therefore be argued how well informed the ice sheet modelling community is up to this point and how well aware they could be without so many studies. I suggest to only mention that 3D GIA models are computationally expensive and complex to couple to an ice sheet model.

Line 81: It is worth mentioning that there might be no asthenosphere at certain locations in Antarctica, and that ELRA includes that mantle, but that does not weigh against the confusion that it could cause to change a name that has been used in numerous papers since 1996. I suggest to leave the name as it is.

Line 86: Add the constant "flexural rigidity" and "lithospheric thickness" in the text, as these are other important parameters in the ELRA model.

Line 101-102: To improve the readability, provide a short explanation about the difference between a viscous channel and a viscous half space.

Line 109-112: Include that 1D GIA models also include the buoyancy effect of the core on the mantle and the mantle on the lithosphere.

Line 118: Include reference A et al. (2012), and Huang et al. (2023) for the finite element method.

Line 119-121: The referenced models in this sentence (Gomez et al. and van Calcar et al.) are coupled ice sheet – GIA models, which require a much longer simulation time than 3D GIA models by itself. Since this section is solely about 3D GIA models, a simulation time of weeks is not applicable.

Line 123-124: The 3D GIA model in van Calcar et al. uses timesteps varying from approximately 1 to 1000 years, depending on the ice loading and the deformation rate, so the lower limit of the timestep of 3D GIA models is not accurate. Furthermore, it is not clear why it is relevant in this context that GIA models sometimes have a larger timestep than ice sheet models.

Line 126-139: The manuscript mentions two regional models specifically (Coulon et al., 2021 & Weerdesteijn et al., 2023). However, it is not clear why these two are picked out, since there are multiple other 1D and 3D GIA regional models (Nield et al., 2018; Book et al., 2022). I suggest to move line 126-129 to the section about LV-ELRA, and line 129-135 to the section about 1D GIA models. Also explain why Coulon et al. and Weerdesteijn et al. are mentioned specifically, and not other regional models. Some other important references are missing, such as Book et al. (2022), who used a similar method as this manuscript for a regional model focused on Thwaites glacier, and Kachuck et al. (2020).

Line 136-137: The available GIA models have a runtime acceptable for modelling ice sheets over glacial cycles, the runtime is only not acceptable to perform ensemble studies with a wide parameter space. Please include this nuance in the manuscript.

Line 137-138: Could you define what is meant by "complexity gap" since there are regional 3D GIA models.

Line 155-159: To improve readability, include a sentence to explain why a placeholder field is used and what the pseudo-differential operator represents.

Line 168-169: Add whether the viscous half space have a variable or constant thickness.

Line 180: Clarify in the text whether R is computed at each time step.

Line 248: Please include why it is required that the far-field displacement should be zero.

Line 307: It is not clear what "in-place" means.

Line 340: To improve readability, include reference to Spada et al. (2011).

Line 415: Given the negligible maximal difference in displacement between the 1D GIA models of Spada et al. (2011), a maximal difference of 0.16 between FastIsostasy and the 1D GIA models of Spada et al. is relatively large. Also, purely based on this idealized test, it cannot be stated that FastIsostasy can replace 1D GIA models. This is also shown by figure 8, showing a maximal error of about 0.8 around -4000 years between the SK1D and FI1D, which is relatively large for a benchmark test.

Line 421-422: Please include the reference to the chosen viscosity.

Line 438-439: The error of FastIsostasy compared to Seakon can be different when a realistic ice load with a realistic Earth rheology is used. Whether FastIsostasy can be used in regional ice sheet models should therefore be concluded based on test 4 as well and should not be stated based on only test 3.

Line 460-461: Could you quantify the error tolerance and adaptive time stepping.

Line 472: Please include the results of this test in the manuscript and quantify what is meant with "better results".

Line 479: Quantify the offset in the forebulge and the implication of leaving the forebulge out of the presentation of the results.

Line 502: Define what is meant by "worst case".

Line 502: Define which region is meant by "near field".

Line 505: The pattern of the rotational feedback is described as "a subtle dipole separated by a great circle" but when one doesn't know what the pattern of rotation looks like, the description is not so clear. It could be described as a gradient from east to west outside of the grounding line.

Line 512: It would be useful to include what the runtime would be when a higher resolution is used.

Line 515: The timestep of the GIA model used in van Calcar et al. (2023) is dynamic and is about 1 year when the deformation rate is high. The convergence of the ice-sheet and GIA histories are there to reach a present day bedrock topography when simulation a glacial cycle. Those iterations would be needed by any ice sheet model coupled to FastIsostasy as well when a glacial cycle is simulated. The iterations are therefore not related to the time step of the GIA model. This should be corrected in the text.

Line 517: Include the resolution of Seakon.

Line 520: It is unclear what is meant by "much richer".

line 533-534: It should be stated more explicitly when FastIsostasy performs better than 1D GIA models, namely between -22 and -10 kyr of the glacial cycle.

Line 541-542: Clarify what is meant by "it minimizes the misrepresentation of the GIA feedback".

Line 560: A discussion should be included that compares the conclusion of this paragraph with literature that does compare incompressible and compressible models, such as Huang et al. (2023). Is this increase in viscosity consistent with literature? Describe the limitations of increasing viscosity instead of including compressibility.

**Technical corrections**

Line 20: impacting > altering

Line 169/Footnote 1: API is not defined.

Line 225: Define F.

Line 242: Define ODE

Line 268: A0 is not defined.

Line 276: SLC is not defined.

Line 365: Define parameters.

Line 501: t = -14 kyr > t = -16 kyr

**Figure and table comments**

- Table 1: includes a description of the rheology, but it is not clear what is meant by Maxwell-like.
- Table 2: Please include a short description of the parameters in the caption.
- Figure 4: To improve readability, include the definition of variables in the caption.

---

## Author Comment (AC2)

We thank the reviewer for acknowledging the ambition inherent to our article, namely developing a GIA model of reduced complexity, able to regionally mimic a 3D GIA model at much lower computational cost, for application to coupled GIA/ice-sheet modelling. We are disappointed that the reviewer felt that our goal was not achieved with FastIsostasy, since none of the concerns they raise argue against the code accuracy that we established through the benchmarks. We emphasise that eight experiments (Fig.7, Fig. 8, Fig. A2 and Fig A3) show differences relative to a 3D GIA model that are small (less than 5% on average, and maximum ca. 15% over space and time). In our view, this benchmarking effort establishes FastIsostasy as the most accurate approach to date for mimicking 3D GIA at greatly reduced computational cost. As such, we believe that it is a significant contribution to coupled ice-sheet/GIA simulations and that GMD is the ideal venue for communicating our results to both the ice-sheet and the GIA communities.

We believe that the reviewer's criticisms can be comprehensively addressed with a series of changes to the text, and that they stem in part from a misunderstanding of the model, which is not a 3D model, and from some confusing text in the original manuscript. As a first step in this regard, we start by proposing a modification of the title:

**FastIsostasy v1.0 – An accelerated, regional 2D GIA model accounting for the lateral variability of the solid Earth**

For the sake of clarity, we will now address each comment of the reviewer. In our opinion, some terms that appear in the review, including the words "comical", "haphazardly" and "author-touted", are not appropriate in a scholarly discussion . We have thus tried to answer the reviewer's concerns whilst maintaining a constructive tone.

1. *The introduction to the manuscript is poorly written. Perhaps, it's symptomatic of a larger issue. While Figure 1 might certainly be relevant to a follow-on paper, these frames are not relevant to this research paper. The heart of this GMD submission is to solve a generally time-dependent 3-D viscoelastic flow problem with complex 3-D distribution of material properties, and then show that the prediction of vertical time dependent deflection is reliable, albeit with some error. The pertinence of the problem to ice sheet stability and past and future sea-level change can be addressed in a single paragraph with appropriate referencing. What is needed in this manuscript are more meaningful descriptions of the approximations and the tests against the Seakon code. I elaborate more in item 2 below. The paper should give the details at the same level as might be seen in journals such as the Journal of Computational                                                        Physics (https://www.sciencedirect.com/journal/journal-of-computational-physics). In fact, such a journal might also be a logical option.*

   We believe that the introduction of the article in its current form is appropriate to introduce ice-sheet modellers - who are often unaware of the complexity of the GIA response and represent our main target audience - to the importance of including laterally-variable Earth structures in their simulations. We motivate this by invoking the GIA feedbacks exerted on marine-terminating glaciers, as depicted in Fig. 1, since their inclusion was shown to differentially impact 1D and 3D GIA models

coupled to ice-sheet models over glacial cycles (e.g., Gomez et al. 2018, van Calcar et al. 2023). Furthermore, we emphasise the present work allows for modelling on time scales relevant for Marine Ice-Sheet Instability over low-viscosity zones, as we discuss below. We are certainly happy to provide more information on the FastIsostasy-Seakon comparison, and our revision will do so.

The present work does not solve a generally time-dependent 3D viscoelastic flow problem but rather approximates such a problem on a 2D grid. We hope that this is clarified, in part, by the change in the title and we will include an additional sentence to highlight this at the very beginning of the manuscript.

2. *Fluid behavior, and elasticity for that matter, are poorly described, both from the standpoint of physics and basic mathematics. The approximation for the lithosphere is that of planar elasticity, an approximation in which a reduced stress tensorial balance is assumed. This is never mentioned, and the severity of the approximation also goes without explanation. Equally, for the fluid part of the physical space the assumption of the layering is that it can be 'lumped' and furthermore each layer acts in a manner like the shallow water approximation in physical oceanography. This puts severe restrictions on the type of interactions a laterally heterogeneous sub-lithosphere mantle may possess (i.e., lateral flow may occur but not like in a 3-D Stokes flow that we are familiar with in ice sheet models and/or the nominal 1-D structure / 3-D GIA simulation of the poloidal flow that employs the full equation system governing the gravitational-viscoelastic response to surface forcing. This could be accomplished by writing out the full equations, then dropping out the terms that yield the theory that is then developed in Section 2.1.*

The approximations and assumptions in our methodology which concern the reviewer are well documented in fundamental citations that are included in our manuscript. For instance, thin plate theory is a widely-used scientific term, and the assumptions behind it, including the reduced structure of the stress tensor, are clearly stated in Ventsel & Krauthammer (2001).

It is true that there are restrictions in the type of interactions ("lateral flow may occur but not like in a 3D Stokes flow") that may occur in FastIsostasy. We agree that this limitation should be explicitly stated and will do so in a revised version of the paper. However, in none of the benchmark calculations we summarise, and, we believe, none of applications to ice-sheet modelling we envisage, will this limitation be disqualifying.

We emphasise on l. 232-235 that we did not succeed in deriving Eq. 9 from first principles and can therefore not fulfil the reviewer's request expressed in the last sentence of this paragraph. Alternatively, we will add an appendix that emphasises which approximations are made in related work (Cathles 1975, Bueler et al. 2007, Coulon et al. 2021) and how Eq. 9 is inspired from these studies.

3. *The referencing is poor. This is usually a problem that can easily be sorted out: referee asks for a new reference, or to delete one, and the author response is simply*

*to agree or argue the contrary. This is not what I mean. The referencing as it stands in the submitted manuscript is quizzical, to the point at which it is clear, in my opinion, that the authors might not have read the papers that they so haphazardly refer to. For example, Section 2.4 discusses the sea-level computations. There is reference to the nonlinearity in the system that is required to be solved but has nothing to do with the traditional problem of nonlinearity treated, for example, by Spada and Stocchi (Computers & Geosciences 33, 2007, 538–562) and that is treated by Mitrovica and Milne (2003) and Kendall et al. (2005). The latter authors also treat, in an elegant way, the problem of migrating boundaries, which, to my reckoning, is NOT a nonlinearity but non-perturbative part of the global GIA problem, i.e., the ocean boundaries need to be updated, and this is simply a trial-and-error process. Mixing the reference to the method of Goezler (2020) and the references in the last three sentences on Page 12 is both confusing and wrong.*

As always, we are open to including references that we may have missed in our discussion, as we describe below. However the suggestion that we are "haphazardly" citing references, some of which were written by co-authors of this paper (Mitrovica et al. 2003, Kendall et al. 2005), is, in our view, unfair.

The reviewer questions our use of the term nonlinear in describing the sea-level theory and prefers the term "non-perturbative". Both terms are accurate. Migrating shorelines on a realistic topography introduces a nonlinearity with respect to the ice load in sea-level physics. We therefore see this concern as one of semantics and will make clearer that the nonlinearity is introduced by the use of a realistic topography.

Finally, as was pointed out by the second reviewer, Section 2.4. needs to include more details about the sea-level treatment. We believe that this addition will resolve the reviewer's concern about references to Goelzer et al. (2020).

*Another glaring example of confusion, is the fact that Table 2 shows values for Young's modulus and Poisson's ratio, implying that we are dealing with a compressible model. But then the Appendix A discusses necessary corrections for the assumption of incompressiblity and claims that the Maxwell time needs to be adjusted to account for this.*

We understand the confusion of the reviewer and will make clearer in the revision that Eq. 9 assumes incompressibility but that, as described in Appendix A, an adjustment of the Maxwell time is made to match the more realistic case of a compressible model.

*The Maxwell time (incorrectly written with Young's modulus) is not influenced by compressibility, since it is a material parameter of the constitutive equation. It is difficult to understand what the authors are up to, but as I interpret what is written in Appendix A, there is confusion between Maxwell time and relaxation time(s), the*

*latter of which can only be derived by solving the eigenvalue problem in the full inverse Laplace transform (though there are alternative approaches to obtaining the decay spectra). I am afraid Appendix A is simply wrong. Some justification is needed as to why this is implemented in the author-touted Julia code.*

The Maxwell time depends on the viscosity and Young's modulus. In fact, the latter depends on the shear modulus and the Poisson ratio. Hence compressibility influences the Maxwell time.

*Some additional remarks*

*Some parts of the paper are well-organized, and I encourage the authors to strengthen these in a revision. The section "FastIsostasy in the model hierarchy" (which seems not to have a number ??) is well written and informative. While the benchmark cases and comparisons in Section 4 are also good, but they are also incomplete. I think there is an underlying contradiction, and I allude to this in the opening sentence of item 1 above. The revised manuscript needs a table to define all acronyms (BSL, etc.).*

We thank the reviewer for pointing out the missing numbering and will add it. We believe a table of acronyms is not necessary, nevertheless the revised version of the manuscript will introduce each of them with more clarity. In our comments below, we argue that the benchmark cases cover the main issues of relevance but we are always open to further suggestions.

*The authors, it appears, are striving to have an efficient way to compute the coupled ice-sheet solid Earth problems. It appears they may have made progress toward this goal. I mention a 'contradiction'. It has to do with the time scales involved. The opening discussion gives a description of an ice sheet retreating on a retrograde slope (Figure 1, which I recommend be deleted). The time scale relevant to such analysis is comparable to the time scale of melt-water pulse 1A, or about 50-500 years, yet the benchmarks of the analysis involve time scales that are more than 1-2 orders of magnitude larger than this. So, this needs to be clarified. In my detailed remarks and conclusion to this review I suggest a way to better address this. As it stands the new code is tested against an FE model on time scales of a full glacial cycle. It's not clear that any of the faster time-scale phenomenon has been addressed. Again, my detailed remarks are designed to help the authors better address this.*

The reviewer's assertion on time scales is incorrect. The time scales of the forcing applied in Test 1-3 (Heaviside in time at t = 0, order of 1 year) and Test 4 (glacial cycle including a rapid retreat in West-Antarctica at t = -14kyr, order of 100-1000 years) correspond to those of meltwater-pulse 1A and more generally MISI. The long simulation times are achieved using short time scales of forcing. Finally, we emphasise that Test 3 as well as Test 4 include low-viscosity zones that produce fast deformational responses, as is evident in Fig. 7c, or in

an animation of the glacial cycle that can be found at https://github.com/JanJereczek/FastIsostasy.jl.

**Detailed Comments**

**Abstract**

*I find the phrase "solve its underlying partial differential equation" odd because Section 2.1 just jumps immediately to the approximations.*

This concern will be addressed by the additional section in the appendix mentioned above.

**Introduction**

*It appears from the 1st paragraph that the authors are interested in ice-solid Earth coupling. In the next three paragraphs the authors are interested in Antarctica, vertical deflection of the surface, lateral heterogeneity and run time per simulation. It's not clear that the title fits the goals. Secondly, it is odd that the authors choose not to reference Sasgen (2017), Kachuck (2020) or Weerdsteijn (2022), since they have goals that align tightly with goals of this paper, especially the former since he attacks lateral heterogeneity in a very simple way that is analogous to what is set up in FI3D.*

We point out that Weerdsteijn et al. (2023) is cited in the paper and that it offers a more complete approximation than the work in Weerdsteijn et al. (2022). We will add the two other references suggested by the reviewer. We must emphasise, however, that none of these approaches offers a large benchmark suite against a 3D GIA model, nor an open-source code that can be used by ice-sheet modellers to account for the lateral variability of the solid-Earth parameters.

*In any revision, there needs to be an explanation of the disparity of modeled time scales (instability vs glacial cycle).*

*Figure 1 needs to be removed since this geometry and time scale does not exist in the remainder of the paper. The physics is an important background to the motivation for the study. This manuscript is about developing an efficient code strategy and finding how much errors exists in the simplifications needed to speed up the run times.*

*This section is well organized, but here the authors need to provide a table of acronyms employed. In describing ELRA it is an oversight not to mention that using the relaxation time to estimate topography change is a method assumed by ice sheet models for a long time now (George Denton, for example).*

These points are answered above and below.

*Line 82. 'as e.g.' -> 'for example as in the case of'*

Agreed. The change will be performed.

*Line 88. I am confused by the 'neighboring points'. It's a continuous medium.*

This text will be revised to "vicinity".

*Line 93. Cite a few ice sheet modelers that use ELRA.*

References will be added.

*Line 106. How is tau (x, y) determined? As a relaxation time it will generally always longer than the local Maxwell time, and at any x , y position there are multiple relaxation times, each with their own wave number dependencies.*

Text on tau(x,y) will be added. We will also mention that in Coulon et al. (2021), one value is used for East Antarctica and another for West-Antarctica (within a range of possible relaxation time scales), with a Gaussian smoothing applied.

*Line 110 'vertical ..' -> 'gravity, vertical …'*

The text will be revised accordingly.

*Line 110 The phrasing 'by means of spherical harmonics' is comical. Better to say: 'by solving the partial differential equations of the time dependent boundary value problem after spherical harmonic expansion of the dependent variables.'*

This text will be modified.

*Lines 119-121. There is no mention of van der Wal's 3-d Finite Volume method. Also, perturbation methods are not really used in practice, though they can provide insight.*

We will include a reference to van der Wal in the revised version of the manuscript.

*Line 125. No mention is made of the ease with which a 1-D model can be structured in the context of a radially stratified earth model (just as in a seismic model) versus the great difficulty of parameterizing a Newtonian viscosity from a 3-D seismic model.*

We will add this material to the revised manuscript.

*Line 141. ' … against analytical, 1D …'. I don't know of any analytical 3-D solutions. Is this just a typo?*

The reviewer might have missed the important comma here. To prevent this, we suggest rephrasing the text as "against analytical solutions, as well as 1D and 3D numerical solutions".

**Section 2.1**

*Equations (2) and (3) need to be derived from the full governing equations in their vector and tensor forms.*

Eq. 2 and 3 express finite difference rules and numerical differentiation via a Fourier transform. There is therefore no governing equation associated with them.

**Section 2.2**

*Line 170. If eta sub l (x,y) is confined to a layer, then the model really is not 3-D. In the real earth we should imagine that flow from a weak zone centered at x1, y1, z1 can flow into/out of another neighboring (weaker/stronger) zone cantered at x2, y2, z2. This is kind of fundamental to a laterally heterogenous mantle under conditions of gravitational disequilibrium.*

The manuscript repeatedly states that FastIsostasy is not a 3D model. We believe that the use of FI3D as an acronym may have led the reviewer to think otherwise, although "3D" refers to the viscosity field used for benchmarking and not the dimensionality of FastIsostasy's domain. To prevent any confusion, we will change the title of the manuscript as mentioned above, and will respectively rename FI1D and FI3D to FI (ELVA) and FI (LV-ELVA).

*Line 180. It is unclear how this equation applies to layers at different depths sine the wave number dependencies should vary as a function of depth.*

As stated in l. 472, shallow depths (about 300 km) are typically used to compute the effective viscosity. Over shallow depths, the wave number does not change significantly and we believe that the approximation is therefore valid.

*Line 196. As stated above I do not understand this scaling. I don't think this is rationalized correctly.*

This is answered above and below.

*Line 215. The reference to Farrell is incomplete. What Green's function? A spherical model? Agreement with Test 3? That hasn't been introduced to the reader at this point in the manuscript.*

The text "Agreement with tests below" will be used instead and the reference will be made with more specificity.

*Line 221. A Stokes flow will always have a dynamic pressure term. The logic presented is flawed.*

We do not solve Stokes flow equations (which is in any case impossible to do on a 2D grid). Using LV-ELVA to approximate 3D solid-Earth deformation resembles the shallow ice approximation of Stokes flow in ice-sheet dynamics in the sense that a 3D problem is approximated by a 2D one. As stated above, we plan to clarify this point in the revised manuscript

*Section 2.3*

*Line 225. We don't normally see a partial u / partial t term in a momentum balance equation of GIA. Please derive this equation and Eq. (14). This is my first time to read a reference to a 'place holder matrix'. This perhaps this illuminates the problem with this paper for me as a reviewer. Perhaps it needs to be submitted to a journal wherein this choice of vernacular is familiar. J. Comp. Phys, or Physical Review, or others might be options.*

The partial u / partial t notation is standard in continuum physics and is used in the main references cited in or manuscript (Bueler et al. 2007, Coulon et al. 2021). A placeholder matrix is simply a variable used to illustrate an operation, as in Eq. 2. It can be replaced by any arbitrary matrix, as its name suggests. The rationale for the reviewer's suggestion to submit the present work to J. Comp. Phys, or Physical Review based on such arguments is difficult to understand.

*Section 2.4*

*Comments on Figure 4 and discussion of Sea Level. See my remarks above.*

*Section 3.*

*Rationalize the use of Julia beyond flowery language like 'vast ecosystem' and 'good performance'. There must be something at the heart of this language that has advantages over C or C++.*

The vast ecosystem is specifically discussed in the bullet points l. 302-330. In any case, it is not necessary for Julia to be superior to C or C++ to justify its use.

*Section 4.1*

*Equation (23) has some assumption of a time history. Please state this assumption.*

This equation assumes a Heaviside function in time for the load (which implies very fast time scales of the load, as discussed above). This point will be stated explicitly in the text.

*Should remind the reader that wave number kappa is a cylindrical wave number that only at high value corresponds to the wave number of a Cartesian system.*

We believe that this reminder is unnecessary in the manuscript.

*Section 4.2*

*Figure 6. It seems 'sea surface' is the depth of water column which changes as a function of time. This needs to be explained much earlier in the paper, maybe even in the Introduction.*

The exact term used in the caption is "sea-surface height", defined in Gregory et al. (2019). This does not correspond to the depth of the water column and the revised version of Section 2.4 will ensure that any such confusion will be avoided.

*Section 4.3*

*Figure 7. The results are really mixed here. Sometimes FI can receive a 'pass' and other times a 'fail'.*

We think the reviewer has misinterpreted these results. FI3D, as defined in l. 428-431, always passes the test by a large margin. FI1D yields an error that is typically made by using

ELVA (Bueler et al. 2007) and is only plotted for comparison. It is our intention in these results to demonstrate quantitatively how the comparison with Seakon improved in the implementation of FI3D relative to FI1D. This is the only criticism the reviewer raises that is related to the accuracy of the code and it appears to originate from misreading the figure and the text describing it.

*Line 457. This is a good point about the final quasi-equilibrium snapshot being misleading.*

We are grateful for this comment.

*Line 462. Concerning shorter time steps: this is common knowledge among those of us who do these computations and formulate models.*

We believe that not all ice-sheet modellers are aware of this and that it is advantageous to inform them when to expect shorter time steps.

**Section 4.4**

*Line 486. I am confused about how to interpret 'mean'. "Spread around unity" leaves me in the cold.*

The mean error is the average value of *e* (defined in Eq. 22) over the masked region. The text reads "spread around identity" (not "unity"), which is labelled in the legend of panel (a) that is referred to in the text. We believe these formulations are not ambiguous and ask the reviewer to be more specific about how we could make the point clearer.

*Figure 8 is generally good. The differences in the 3D approaches might be interpreted by some as being unacceptably large. But this depends on the specific geological or glaciological question being asked of the geoscientific modeler.*

As pointed out by the reviewer, it is up to the ice-sheet modeller to decide whether the error values are acceptable for their application. We emphasise that FastIsostasy is a computationally inexpensive regional model incorporating a treatment of sea-level and that our benchmarking against a fully 3D GIA model demonstrates that it will be useful for most ice-sheet modellers since it greatly reduces the error in computing vertical displacements compared to ELVA (FI1D) and ELRA (the latter comparison will be included in the revised version of the manuscript).

*Again, there is no reference to the important time scale for instability: 50 – 500 years.*

As stated above, the deglaciation includes a rapid retreat in West-Antarctica which is fully captured in the modelling, and that its great utility is established by the relatively small difference between the FastIsostasy and Seakon simulations, even for these such short time steps (around -14kyr).

*Appendix A*

*See above comments.*

*Appendix B*

*If material in Appendix A has any logical rationale (and I doubt it), then it needs folding into the main text. Appendix B can be added as a Figure companion to Figure 8 in the main text. I recommend removing the Appendices altogether.*

As answered above, the reviewer seems to have missed the influence of compressibility on the Maxwell time. The correction factor we introduce in Eq. A2 is justified by physical equations and is important to explore given that Eq. 9 relies on the assumption of compressibility.

In contrast, the correction factor introduced in Eq. A3, is of a mathematical nature, albeit one that relies on physical arguments. We believe that it is also an important parametrization that the reader should be aware of.

Finally, we believe that Appendix B has value since it presents the structure of the parametric heterogeneities of Test 3, two additional tests and a comparison of FI1D and SK1D. This additional information is not central to the paper but will be of interest for readers.

**Suggestion for an additional test**

To capture the physics of solid earth LV and stability of an ice sheet margin, I suggest the following test that samples shorter scales in both time and space.

Allow the x, y system at its various layers to have a set heterogeneity, as for example in one direction across the space. Then consider linear (in time) surface load as a half-torus (ring) that has negative changes in mass over 500 model years, testing the model prediction of vertical deflection at 50-year intervals at the surface across the entire space. This would excite both high and low values of kappa as the surface load would have both short and long spatial scales of loading.

As mentioned above, the glacial cycle presented in Test 4 includes all wavenumbers that are relevant to ice-sheet modelling. We therefore see this additional test as unnecessary.

---

## Author Comment (AC3)

We would like to thank the reviewer for their positive comments, as well as constructive suggestions for improving the manuscript. In particular, the reviewer points to section 2.4 and the need for a more explicit comparison to Coulon et al. (2021). As we outline below, we will address all their concerns in the revised version of the manuscript.

We now give a detailed answer to the reviewer's comment.

In this study, the authors present FastIsostasy, a regional GIA model capturing lateral variations in lithosphere thickness and mantle viscosity, as well as gravitationally consistent sea-surface changes. The key advantage of FastIsostasy is its ability to bypass the computational expense of 3-D full self-gravitating earth models. This makes it a very promising tool, particularly for ice-sheet modellers, as it facilitates the consideration of lateral variations in solid Earth properties often overlooked in sea-level projections due to computational constraints. FastIsostasy hence holds promise for significantly enhancing the accuracy of sea-level predictions, particularly for the Antarctic ice sheet. Spatial variations, especially beneath the West Antarctic ice sheet with its low mantle viscosity, have been identified as crucial in Antarctica. The potential for triggering negative feedbacks that limit and delay mass loss adds to the interest of this novel model.

Overall, I find the manuscript to be well-written and effectively presented. FastIsostasy appears to be a compelling tool, advancing beyond previous models that addressed a similar exercise (such as the Elementary GIA model). The benchmarking in section 4 against 1D and 3G GIA solutions is very interesting and provides valuable insights into the model's behaviour. The open-source and well-documented nature of the model adds immense value.

We thank the reviewer for acknowledging the development effort, including technical aspects such as the code documentation.

However, the manuscript could benefit from a few clarifications:

- If my understanding is correct, FastIsostasy is essentially a 2D model due to the lumping of the depth dimension. I think this needs to be emphasised more clearly in the manuscript. This aspect is sometimes presented in a misleading way, and a more explicit acknowledgement of this approximation and its implications is needed.

The reviewer is correct. FastIsostasy is a 2D model. Other reviewers shared this confusion and the issue will be clarified in the revised version of the manuscript.

- FastIsostasy allows to include gravitationally consistent sea-surface changes in a regional domain. Again, this allows for a significant improvement in ice-sheet projections, given that most ice-sheet models consider the sea surface to be uniform, missing important feedback influencing the stability of grounding lines. Unfortunately, the section introducing the sea-level model

and its improvements compared to Coulon et al. (to account for time-varying ocean area) lacks clarity. A restructuring and clarification of this section, with a focus on explaining how the improvements offer a key advantage, would enhance the manuscript (see my suggestions in the specific comments below).

We agree that the implementation of the sea level theory was not sufficiently documented. A more precise description will be provided in the revised version of the manuscript following the suggestions made in the reviewer's specific comments.

● Table 1 provides valuable insights for comparing FastIsostasy with other existing approaches. However, the current manuscript could benefit from a more explicit discussion in the main text regarding the similarities and differences between FastIsostasy and the Elementary GIA model. This is particularly important since both models aim to address similar challenges, namely, offering a computationally-efficient approach to incorporate LV solid Earth properties and sea-surface changes. While it is evident that FastIsostasy is more complex, there are noteworthy similarities between the two approaches. One notable advantage of FastIsostasy, briefly highlighted in the conclusion section, is its ability to avoid translating viscosities into relaxation times. Expanding on this point in the main text, in addition to the information presented in Table 1, would contribute to a more comprehensive understanding of the strengths and distinctions of FastIsostasy compared to the Elementary GIA model.

We agree with the reviewer's comment and will ensure that the revised manuscript provides a more thorough comparison between the two approaches.

Once these clarifications are addressed, along with the specific comments provided below, I believe this paper is suitable for publication and would make a very valuable contribution to the field.

**Specific comments :**

● Introduction, l.24-26: I would suggest reformulating this, as it implies that enhanced melting is the driver of bedrock uplift and sea-level drop, not the grounding-line retreat itself.

Agreed. This point will be clarified.

● Introduction, l.25: Maybe it is worth clarifying what is meant by 'sea-level' here, as a panel of definitions exist. I suspect you mean the relative sea level, i.e., the difference between the bedrock and the sea surface.

Yes, we do mean the sea surface elevation relative to crustal elevation. We will make this explicit.

- Introduction, l.31: Maybe clarify here that it is the creation of pinning points that explains the influence of GIA on the buttressing ice shelves?

Agreed. This point will be clarified.

- Introduction, l.34: Replace 'parameters' with 'properties'? To check throughout the manuscript.

This change will be made throughout the manuscript.

- Introduction, l.56: I would remove 'of the ice-sheet and GIA models' from the sentence as you do not discuss uncertainties in ice-sheet models themselves. For example, 'Such uncertainty thus needs to be propagated…

We will remove this text.

- Introduction, l.59-60: Coulon et al. (2021) addressed this parametric uncertainty by running large ensembles of simulations with a computationally efficient GIA model. This is worth mentioning here.

This will be indicated in the revised text.

- Introduction, l.62-64: This is not entirely true, as Coulon et al. (2021) proposed a way to account for the lateral variability in Antarctic rheological properties at a low computational cost. I know that it is presented later, but I think that, even though their model is of lower complexity than FastIsostasy, it is worth mentioning their work here, and more generally in this section, especially as their motivation was very similar to the one presented here.

This will be indicated in the revised text.

- Introduction, l. 65-69: Some of these concepts have not yet been introduced or only very briefly, e.g., the depth dependence of the mantle viscosity (only shown in Fig.2 but not mentioned in the text), or the dependence of the response time scale on the spatial scale of the load. Either introduce these before or simplify the description here and give these details later.

The concepts described on these lines will be more thoroughly introduced in a revised Introduction.

● Introduction, l.78-79: I am not sure what is meant by 'sufficiently constrained in literature'.

Although uncertainties remain in estimates of upper-mantle viscosity, the value adopted in published articles is always prescribed in the case - as in our article - of adopting a Maxwell rheology, The situation is more complicated in the case of a Burgers rheology, which requires specification of two viscosity fields. We will clarify this point in the revised manuscript.

● Introduction, ELRA description: I think this section lacks a few references, for example, to illustrate studies that use such a model.

These references will be added to the manuscript.

● Table 1: Why does the Elementary GIA model have a '~' symbol in LV? This would be worth explaining in the main text or caption. Same for the 'sea-level' for LV-ELRA and FastIsostasy, clarification in the caption might be useful.

In deriving a field of relaxation time, Coulon et al. (2021) apply a Gaussian smoothing over a binary field. This is an approximation of the structure of LV in comparison to adopting a field such as that depicted in Fig. 2. Deriving *a-priori* estimates of the relaxation time from Fig. 2 and using these in LV-ELRA is possible but tedious and does not consider the dependence of the adjustment time scale on the load wavelength. We will add text to justify our use of the "~" symbol at this point in the manuscript.

● Section 2.2, 198: While I find the idea of lumping the depth-varying viscosity profile at a location into an effective viscosity interesting and understand its value for the models' computational efficiency, I think that it remains important to discuss the limitations in more detail. In particular, it is important to acknowledge that (similar to previous LV-ELRA attempts) because you end up with a unique effective viscosity value, it does not allow to capture the full multi-normal mode response of the Earth to surface loading which is accounted for in 1-D and 3-D GIA models due to their viscoelastic layering. In fact, the larger the load, the deeper the deformation reaches into the mantle. The ease with which the mantle relaxes thus depends on the radial viscosity profile, with, e.g., the shallower layers being the more relevant at the local spatial scale.

This reviewer is correct on all these points. The connection between load size and depth sensitivity is well described in the literature. For example, the response to the ocean load or a much larger ice sheet (e.g., the Laurentide Ice Sheet) will depend on deeper mantle structure than we have considered. However, the near-field

displacement in the vicinity of the West Antarctic, for example, will be dominated by viscosity within the depth region we account for, which is reflected in the accuracy we've established for FastIsostasy relative to a fully 3D GIA simulation. In any case, we will discuss the issues the reviewer raised in the revised manuscript.

- Section 2.3, l.209-210: Is this similar to what is performed in Bueler et al. (2007)? If yes, this should be acknowledged. If not, maybe explain how it differs.

Yes, it is. This will be made explicit in the revised manuscript.

- Section 2.3, l.215: refer to section 4.3 here for clarity.

This will be done.

- Section 2.4: I find section 2.4 along with Figure 4 a little confusing. In particular, I find the motivation for the extension to Goelzer et al. (2020) unclear. Overall, your regional sea-level model is largely based on Coulon et al. (2021), except that you propose an improvement to account for the time-varying ocean surface. The motivation and significance of this improvement are not clearly expressed. I would suggest that you start this section by providing more detail on what influences sea-surface changes, i.e., explaining that you calculate the regional sea-level field using equation (20). This will give the reader more context. I would then introduce the gravitationally consistent sea-surface changes, i.e., the sea-surface height perturbation, which is what has been emphasised so far in the manuscript and especially in the introduction, and which will dominate the sea-level signal. Unless my understanding is wrong, equations (17-18) are required to improve the estimation of s(t). The section would also benefit from a better introduction to s(t) and how it is defined.

- Section 2.4, l. 276, and Figure 4: I believe that SLC has not been introduced so far.

The concern raised by the reviewer is similar to comments by one of the additional reviewers. We agree that our treatment of gravitationally self-consistent sea level should be discussed in more detail and we will provide a more substantive discussion in the revised manuscript.

- Section 4, l.356-362: Is it really necessary to define 'acceptable' error bounds a priori? Unless you can infer them from actual studies using the same 3D GIA model with different viscosity fields (if so, please provide a reference), the

values proposed here seem rather arbitrary. I think it is sufficient to say that larger errors comparable to parametric uncertainties are acceptable.

This is an opinion shared by other reviewers, which we agree with. We will modify the manuscript accordingly.

● Section 4.2, I.412-414: What do you think is the influence of the regional versus global domain? Could it influence the larger differences in N towards the edge of the load? It might be worth mentioning it.

The use of a regional domain introduces errors associated with large scale gravitational effects that can only be captured in a global geometry. We will include additional explanation of the limitations of a regional domain in the revised manuscript.

● Section 4.2, I.415: I think I would use 'comparable' instead of 'excellent'.

This revision will be made in the revised manuscript.

● Section 4.3, I.429: I find the name FI3D misleading. If my understanding is correct, the parameter fields in FastIsostasy are 2D and not 3D, given that the depth dimension is lumped. Please clarify.

We agree and will revise the label to avoid confusion (for more details, see the summary of answers to the reviews).

● Section 4.3, I.440: To what do you attribute this underestimation of the forebulge?

Our hypothesis: Analyses of the resolving power of data related to glacial isostatic adjustment have shown that forebulge dynamics may be sensitive to relatively deep mantle viscosity (e.g., Mitrovica et al., JGR, 1993). We suggest that the inaccuracy arises because our mapping of 3D viscosity structure into 2D does not capture this sensitivity.

Section 4.3, I.453: I find the reference to ELRA misleading given that it also has a computationally-efficient LV version (Coulon et al., 2021).

We will resolve this confusion in the revised version of the manuscript.

- Section 4.3, l.455-459: This is an interesting point, but I do not understand the reference to Le Meur and Huybrechts here since (i) they do not only look at the final uplift map but also at the transient evolution (for the mean bedrock evolution) and (ii) they do not consider heterogeneous solid Earth configurations.

The reviewer is correct to point out that LeMeur and Huybrecht (1996) present a time series of the mean bedrock elevation across different models. Our point is the fact that the spatial comparison is only performed for the present-day uplift map. We believe that this is not sufficient and that the spatial pattern of the error should be analysed for various time steps, or alternatively for the time step of maximal error, as we do in Fig. 8. LeMeur and Huybrecht do not consider heterogeneous Earth structures but a heterogeneous response will nevertheless arise because the spatial scale associated with retreat in the Ross and Ronne regions is different from mass flux associated with the comparatively small retreat in (for instance) the Amery region. Since none of the regional GIA models included in LeMeur and Huybrechts incorporate a dependence on the load wavelength, there may be substantial differences in the spatial pattern of displacement that are "smoothed out" by simply studying the time series of the mean displacement. Our objection would be milder if the time series showed the spatial mean and maximum error over time, since these are stricter metrics.

- Section 4.4, l.473: observed where? Please provide context or a reference.

The revised manuscript will provide more details by discussing results obtained by running simulations with various maximal depths.

- Figure A3: the colorbar is missing.

Thank you for noticing this error, which will be corrected.

- Section 4.4, l.477-479 (Figure 8): I find this choice of the masking corresponding to the LGM extent questionable, especially as the area that matters for marine ice sheets is the area around the grounding line. I would suggest taking the whole domain, or an area larger than the ice sheet extend to include the vicinity of the grounding line. Typically, you have shown in the previous tests that FI underestimates the peripheral forebulge. The masking applied here may therefore ignore this signal.

Since the LGM only represents a few kiloyears of the full glacial cycle, the peripheral forebulge and grounding-line vicinity are included in the mask for the vast majority of the simulation, although admittedly not for the LGM. Furthermore, the error in the peripheral forebulge is relatively small (about 10 metres for all cases of Test 3), albeit

systematic, and is therefore likely to have only marginal significance in a glacial cycle that exhibits displacements of more than 550 metres. This issue will be resolved by extending the mask, as we indicated in our summary of the answers to the reviews.

- Figure 8: The panel subscripts are arranged in a confusing way.
- Section 4.4, l.501: I believe that t=-14kyr is not shown on the figure.

Thank you for pointing out both issues, which originate from a late modification of Fig. 8 shortly before submission and will be corrected.

- Section 4.4, l.502: I am not sure where to look at. From Figures 8e and 8h, it seems to me that FI3D underestimates around West Antarctica and overestimates around East Antarctica.

The revised version of the manuscript will include an additional sentence to make clear that panel (h) shows that over most of the mask the vertical displacement is underestimated, with the exception of the Eastern margin. This is likely to be due to the rotational feedback and the associated increase in ocean load in this region, which is captured by Seakon but not by FastIsostasy.

- Section 4.4, l.507: But does FastIsostasy take into account the loading influence of sea-surface changes? I don't think this is mentioned in the manuscript?

It does. This inclusion is evident in Eq. 1. However, we concede that section 2.4. misses this point, which will be included in the revised version of the manuscript.

- Conclusion, l.525-532: I am not sure whether these limitations have their place in the conclusion, as I do not think they (or at least for some of them) have been discussed before. Perhaps instead in a 'discussion' or 'limitations' section, or in section 2, which presents the model?

This is a sound point. The revised version of the manuscript will follow this advice.

**Minor comments/typos:**

- Introduction, l.44-48: This long sentence is a bit hard to follow, maybe try to split it for clarity?
- Introduction, l.56: 'ensemble simulations' -> 'ensemble of simulations'.

- Figure 1, caption: 'from (Whitehouse et al., 2019)' -> from Whitehouse et al. (2019). I spotted this issue at other locations in the manuscript.
- Figure 1, caption: 'enhanced melting at the grounding line, leading to larger thickness…' -> 'enhanced sub-shelf melting, leading to grounding-line retreat, and therefore larger thickness and increased outflow at the grounding line'.
- Section 2.2, l.179: 'the following scaling factor'?
- Section 2.3, l.232: 'a an ad-hoc'
- Section 4, l.346: 'ice loading history', or even instead 'by an ice loading history over a full glacial cycle'?
- Section 4.1, l.393: 'present the following differences'
- Section 4.3, l.421: 'in (Gomez et al., 2018)'

Thank you for pointing out these errors. They will all be corrected in the revised manuscript.

---

## Author Comment (AC4)

We thank the reviewer for their positive comments in regard to the manuscript, and their constructive suggestions for improving the discussion of our methodology and results. We detail below our response to each of the reviewer's comments.

**Summary:**

Swierczek-Jereczek and colleagues present a regional Earth model that improves previous regional approaches by adding capabilities to account for the heterogeneous structure of the Earth as well as for barystatic and gravitational feedbacks on relative sea level. FastIsostacy is mainly build on the fast Fourier transform (FFT) formulation (Bueler et al., 2007) of the Lingle-Clark bed deformation model (Lingle and Clark, 1985), where two linear models for purely elastic and viscous displacement were superimposed. This results in a single time-dependent differential equation for the combined vertical displacement (Cathles 1975). Numerically, this equation is discretized in time using a finite difference scheme, while spatial derivatives are computed in Fourier space (Fourier spectral collocation method), allowing for using explicit integration methods instead of solving for large systems of linear equations.

This is correct, excepting that some spatial derivatives arising in Eq. 9 are evaluated by a finite difference method. The hybrid nature of our solving technique arises from this choice rather than from the time integration.

The proposed Earth model also applies a regional sea level model introduced by Coulon et al., (2021), which here additionally accounts for barystatic sea-level dependent ocean area (shoreline migration), which is required in the conversion of ice volume changes into sea-level changes. The authors show that on glacial time scales this correction can explain about 4% of the sea level change.

The presented Earth model is of intermediate complexity between the simplified models used in current ice-sheet simulations and the full spherical GIA models, nicely categorized in Table 1 by the authors. FastIsostacy (or LV-ELVA in their notation) has not been coupled to an ice sheet model (at least not shown here) but from an ice modeling perspective it potentially fills a gap. The authors raise the problem, that GIA models are rarely open source and computational expensive due to the numerical integration against the global load history. Parameter ensemble simulations, as often used to estimate uncertainties in sea-level projections, require computational costs of the GIA model comparably to the ice sheet model or lower. As standalone model, FastIsostacy makes use of Julia programing language allowing for GPU parallelization for the FFT-related operations and additional functionalities. The main speed-up of FastIsostacy compared to full spherical GIA models, however, is due to the Fourier spectral collocation method, as already used in ice sheet model coupling (e.g. PISM, https://research-software-directory.org/software/pism).

**General assessment:**

FastIsostacy combines the previous ELVA approach (Bueler et al., 2007) with a regional sea level approach (Coulon et al., 2021) and extends those by adding a correction for ocean area and for projection (distortion factor K) and by including hydrostatic force induced by elastic displacement. The authors suggest to account for ocean and sediment loads, which has not been treated in the previous ELVA, as this may influence the boundary conditions (load perturbations should vanish at the boundaries assuming spatial periodicity) and therefore the FFT solution, even for an extended computational domain. For ice load changes in the domain center with distance to the lateral boundaries this assumption is more likely fulfilled.

Most importantly, laterally-variable solid-Earth structures can be treated without introducing major errors compared to a fully 3D GIA model. As mentioned at l. 295 of the manuscript, accounting for changes in sediment and liquid water column is only performed within a masked region to fulfil the vanishing boundary conditions mentioned by the reviewer.

The authors claim to account for a higher vertical resolution in Earth structure using a "lumped" weighted average defining an effective viscosity, a concept commonly used in terms of 2D Shallow Approximations in ice sheet modeling. I am not sure if the channelized (lateral) viscous flow of mantle material (e.g. in a low viscosity channel) can be really represented this way, as the systematic underestimation of peripheral forebulges compared to the 3D GIA model suggests. Also self-gravitational effects are ignored in the approach. The viscosity (Maxwell time) tuning, which is motovated by representing compressibility and shear modulus variations, may compensate for the effects of reduced complexity. The comparison to the 3D GIA effects also shows the effect of the missing rotational sea level feedback in FastIsostacy, which can be of the order of 10s of meters for glacial cycles simulations. This can be significant in terms of marine ice sheet instability and the onset of deglaciation in coupled ice sheet – GIA models.

Nevertheless, the modeling work done by the authors is remarkable, and add another piece to bridging between ice sheet model and GIA model community. The performed standalone test simulations cover different aspects of the implementation (verification, 1D benchmark, 3D and glacial cycle application for Antarctica), which makes sense to me. The presented misfits are below 20%, which can still be of the order of 100m. The authors argue that this is comparable to parametric uncertainties, but those misfits can be amplified in coupled simulations, especially for a more realistic deglacial ice sheet grounding line retreat (with faster time scales).

We thank the reviewer for their positive comments which acknowledge that FastIsostasy represents a major step forward compared to current simplified GIA approaches that are standard in ice-sheet modelling, and that it provides a computationally highly efficient alternative to fully 3D GIA models. The limitations of FastIsostasy mentioned by the reviewer are systematically described in the manuscript.

We have coupled FastIsostasy to the Yelmo ice-sheet model but the evaluation of its performance requires a detailed benchmark comparison against a coupled ice-sheet–3D

GIA model. This effort will be the focus of future work. A detailed response to the reviewer's comment on the effect of lumping the depth dimension is given below.

This model approach may still fill a gap, but ice sheet – GIA coupling is just improving significantly with reduced computational costs and comparably small coupling time steps (e.g. up to 10yr with a GIA model based on spherical harmonics as in Albrecht et al., preprint), while providing a more comprehensive GIA response covering all implied GRD (gravitational, rotational, deformational) feedbacks, with a globally conserved water budget, considering different sea level contributions and their interaction (e.g. Greenland) and global relative sea level fingerprints (geoid), also accounting for horizontal displacements (for validation against GNSS data). Hence, sea level projections and uncertainties could be directly linked to potential impacts along the global coastlines. This is what decision makers want to know in the end, and maybe it is time for ice sheet modelers to do this step forward, to think their models more globally.

We fully agree with this point and believe that 3D GIA models and FastIsostasy have complementary goals. 3D GIA models are the appropriate tool for studying the global, space-time variation in sea level across glacial cycles and into the modern world. On the other hand, the development of FastIsostasy was motivated by the application to coupled ice-sheet/sea-level modelling since it avoids the prohibitive computational costs associated with 3D GIA models while dramatically improving accuracy relative to simplified GIA models currently in use. It can therefore be a very useful tool for ensemble simulations exploring model sensitivities. We will include a few sentences in the revised manuscript to highlight this complementarity.

**Detailed comments:**

l. 58 I think that SELEN (Spada & Melini, 2019) come with a BSD3 open source software license, which allows commercial usage. This is comparable to the permissive MIT license used for FastIsostacy, both without copyleft.

This is indeed the case, however SELEN is a 1D GIA model. Our statement refers to 3D models. We nonetheless agree that mentioning Spada & Melini (2019) here is a good point, since it is the first open-source effort in GIA modelling to our knowledge.

l. 162 What would be the induced effect of neglecting the distortion factor, just a rough estimate? In terms of estimating sea-level contributions, the same weighting terms should be used (Goelzer et al., 2020) in the regional sea-level model (Eq. 16). I encourage the authors to double-check for double-accounting?

The distortion of the projection is, in fact, accounted for in Goelzer et al. (2020) but only in the computation of the sea-level contribution. Our approach extends its meaning to the derivatives of the displacement field arising in Eq. 9. Neglecting the distortion factor has an

only marginal effect for the simulations performed in the present work. However, we believe that the factor can play an important role in simulations of the past evolution of larger ice sheets, e.g. Laurentide + Greenland on one domain.

l. 251 Does u_i,j in Eq. 15 also depend on time t?

Yes! This dependence will be made explicit.

l. 252 To my knowledge this requirement in the current PISM implementation has been improved by using an extended computational domain for the Earth model, at least 4 times the size of the ice domain. The authors also mention an extended domain in l. 292, but without any details.

We will include this additional information in the revised manuscript.

l. 258 Maybe refer to Gregory et al., (2019) for definition of 'barystatic sea level' change. I think that sea level change induced by vertical land movement (V_pov) is treated separately.

We will incorporate this suggestion in the revised manuscript.

l. 262 For gradual sea level changes the slightly different approach by Adhikari et al., (2020) may be a valid alternative.

Thank you for pointing this out. We will include a reference to Adhikari et al. (2020) in the revised manuscript.

l. 263 the sea level depends on the ocean area as function of the sea level, which is implicit, but not necessarily nonlinear.

It becomes nonlinear when using a realistic topography, as shown in Figure 4. This will be made explicit in the revised text.

l. 289 Eq. 20 is an approximation of gravitationally consistent geoid changes, allowing to approximate near-field relative sea-level changes, but the complete sea-level equation (Farrell & Clark, 1976) is not solved here, as the deformation of the whole Earth surface is not considered.

This is correct. We will emphasize this point in the revised manuscript.

l. 295 This predefined mask should be shown somewhere. Does this also apply to the ocean mass changes in Eq. 1? Peripheral forebulge effects at the edge of the LGM ice sheet extent can provide important feedbacks and should be represented (see Albrecht et al., 2023). Does "extended domain" then mean, without the mask?

We agree and will show the mask in the supplementary material. This addition will make clear that the peripheral forebulge and relevant ocean mass changes are included in it. The mask is only applied to the pressure term of Eq. 9 and is introduced to prevent a nonzero load close to the boundary, which allows one to solve Eq. 9 with a Fourier collocation. Note that this issue is only relevant for Test 4, since the other tests present a load that is concentrated at the centre of the domain (⅓ of the total length).

The term "Extended domain" refers to a domain with a doubling of the dimensions, which naturally arises when computing a matrix convolution numerically. The mask does not apply to any of these computations and is only included on the right-hand side of Eq. 9.

l. 306 What are FFT plans?

FFT plans include all precomputable operations of an FFT and can therefore save a substantial amount of time. We will include this explanation in the revised manuscript.

Fig. 5: It would be helpful to indicate in the figure where the load is located (vertical dashed line, or illustrated at top of panel a) at 1000km. Similar in Fig. 6 for the margin (10°) or shape and for Fig. 7 for the anomaly extent.

We agree. These additions will appear in the revised version of the manuscript.

l. 440 Albrecht et al., (2023) highlight a possible forebulge feedback for ice sheet growth.

This is a relevant citation and we will include it in the revised version of the manuscript.

Fig. 7: It would be nice if the experiment names (150→50km, 150→250km, 21→20, 21→22) were put into a subtitle, legend or figure caption. Also the forebulge region could be highlighted (arrow?). I like the detailed error statistics.

This is a very good that will make the figure easier to read and interpret.

l. 462 Doesn't 'stiffer' mean a larger Maxwell or relaxation time?

Stiff here refers to the mathematical vocabulary of ODEs. We will resolve this confusion in the revised version of the manuscript.

l. 466 How is the forcing applied, the model uses the ICE6G ice history ΔH_ice in Antarctica and the barystatic sea level change from the northern hemisphere (or outside of the computational domain) from ICE6G as s(t)?

This point was not sufficiently described in the manuscript and Section 2.4 will be extended to add the required discussion. We note that the barystatic sea level used here is obtained by regionally averaging the sea-surface height obtained in Seakon, which differs significantly from the global mean. We believe that using external barystatic sea level is the best option for simulations of the past, since real-world data is available. Updating the barystatic sea level as described in Section 2.4 will only be used for projections. This point will be highlighted in the revised version of the manuscript.

l. 472 This overestimation suggest that the weighted mean (Eq. 4) may not be valid for n layers.

This overestimation can be understood to be a consequence of the lumping process, which requires setting a characteristic wavelength, as mentioned on l. 193. The wavelength chosen here relies on the characteristic extent of the load in ice-sheet modelling, which is sensitive to structure in  the top-most layers of the sub-lithospheric mantle - at least for load sized considered in the manuscript. We will include an explanation of this point in the revised version of the manuscript.

Furthermore, we want to emphasise that the bulk of the error budget in the FastIsostasy simulations  arises from lumping the third dimension of a Cartesian domain (Fig. 3). As shown in Fig. A2, relative errors that are comparable to those displayed in Fig. 8 arise even when only including 3 layers (a lithosphere, a channel and a viscous half-space - similar to Lingle & Clark (1985) and Bueler et al. (2007)).

Fig. 8: panel labels may be twisted. Panel e) and f) clearly show the order of magnitude of the rotational sea level feedback of the order of 10s of meters (as discussed in l.506ff).

The inverted labels will be corrected.

l. 500 80m maximum error sounds quite a lot, even though it is below the 20% acceptance limit.

This is correct. The main achievement here is the level of error reduction (at almost no computational cost) compared to a configuration with homogenous solid-Earth parameters.

l. 501 mentions peak maximal error of FI3D at -14kyr, while the figure subtitle and panel b) says -16kyr.

Yes, this will be corrected in the revised manuscript.

l. 515 The authors may confuse model time steps with coupling time steps or coupling intervals. The more comprehensive 3D GIA models likely use time steps lower than 10 years as well.

The studies cited in l. 515 uses model time steps of more than 100 years.

l. 518 The factor 70000 for Seakon vs. FastIsostacy might be a bit exaggerated, as it is only valid if you compare CPUh with GPUh, while GPU also use internal parallelization. If only wall clock time or energy consumption was compared (GPU: 175W, 128 CPU core node: 280W), it would be still a factor of around 1000. So more importantly here is the fact, that computational costs for the GIA model should be comparable to the ice sheet model (or lower). And this is already the case (see Albrecht et al., 2023, Table 1, which can be associated with a factor of still around 100 for PISM-VILMA compared to FI3D).

We partially agree with this comment and will simply mention an order of magnitude for the difference in run time in the revised manuscript. We nonetheless draw attention to the fact that the GPU used in the study has a significantly lower power consumption (80 W) and the total energy required for the computation is 80W * 10 mins vs. 280W * 4.5 days, which is substantial, to say the least. Most importantly, the acquisition price of the GPU (<1000 euro) is much lower than that of a 128 CPU cluster. We include these additional remarks in the revised manuscript.

Fig. A3 colorbar is missing

Yes, this will be corrected.

The affiliation of Konstantin Latychev states 'Seakon', or should it rather be the Physics Department of the University of Toronto?

The affiliation of Konstantin Latychev changed recently and is correct as it is in the current version of the manuscript.

Typos are covered by the other reviewers already.

**References not mentioned in manuscript:**

Adhikari, S., Ivins, E. R., Larour, E., Caron, L., and Seroussi, H.: A kinematic formalism for tracking ice–ocean mass exchange on the Earth's surface and estimating sea-level change, The Cryosphere, 14, 2819–2833, https://doi.org/10.5194/tc-14-2819-2020, 2020.

Albrecht, T., Bagge, M., and Klemann, V.: Feedback mechanisms controlling Antarctic glacial cycle dynamics simulated with a coupled ice sheet–solid Earth model, EGUsphere [preprint], https://doi.org/10.5194/egusphere-2023-2990, 2023.

Gregory, J. M., Griffies, S. M., Hughes, C. W., Lowe, J. A., Church, J. A., Fukimori, I., Gomez, N., Kopp, R. E., Landerer, F., Cozannet, 610 G. L., Ponte, R. M., Stammer, D., Tamisiea, M. E., and van de Wal, R. S. W.: Concepts and Terminology for Sea Level: Mean, Variability and Change, Both Local and Global, Surveys in Geophysics, 40, 1251–1289, https://doi.org/10.1007/s10712-019-09525-z, 2019.

Spada, G. and Melini, D.: SELEN4 (SELEN version 4.0): a Fortran program for solving the gravitationally and topographically self-consistent sea-level equation in glacial isostatic adjustment modeling, Geosci. Model Dev., 12, 5055–5075, https://doi.org/10.5194/gmd-12-5055-2019, 2019.

We thank the reviewer for suggesting these additional references and we will include them in the revised version of the manuscript. Spada and Melini (2019) is actually already included, but it will be referred to in the discussion about open-source GIA code.

---

## Author Comment (AC5)

We thank the reviewer for their positive assessment of our manuscript and for their constructive suggestions for improvement. We provide below a detailed response to their comments.

*Swierczek-Jereczek et al. present a new GIA model, FastIsostasy, based on the Fourier collocation method that can include lateral viscosity and lithospheric thickness variations in the mantle. The new model is compared to the existing 3D self-gravitating visco-elastic Earth model Seakon. They show a maximum error of 0.2 over a glacial cycle when FastIsostasy is used compared to Seakon. Although the error is not negligible, FastIsostasy has the potential to be coupled to ice sheet models because the runtime is very short and the model is open source.*

*The manuscript is well written. The introduction provides a great overview of current existing methods used in GIA models. The second part of the introduction, "FastIsostasy in the model hierarchy", provides a clear overview of the different GIA models. Table 1 also includes which numerical scheme is used for each type of GIA model, but the numerical scheme are not explained in the text of the section "FastIsostasy in the model hierarchy". For this manuscript, the FDM/FCM and the numerical scheme of Seakon are particularly important and should be described in more detail.*

The numerical scheme of Seakon is extensively described in Latychev et al. (2005), whereas the combination of FDM/FCM adopted in FastIsostasy is described in Section 2.3. We will nevertheless extend the discussions around l. 240 to add details about both methods.

*The method section explains quite well the details of the new method developed but should elaborate more on the effect of simplifications in FastIsostasy in order to reduce computational cost compared to Seakon. Line 503 states that tuning can be done easily but doesn't explain what tuning would exactly be required.*

Tuning the density or the effective viscosity field could be performed by minimising the mean square difference of the predictions to the Seakon-derived displacement and would only require the use of an optimisation technique. See for instance Kalman inversion in the documentation of the code. Simpler techniques such as hand tuning can also be adopted.

*Also, line 537-539 states that the difficulty of creating meaningful ensembles is decreased but even though different viscosity fields are available, it is not shown how FastIsostasy performs using different viscosity fields. It is necessary to discuss the implication of the use of effective viscosity, the characteristic wavelength, and alfa. Equation (6) suggests that the alpha is selected to improve the fit to a 3D model, which means the results depend on a particular 3D GIA model. Are alpha and W fixed for all comparisons? That results in conclusions that are dependent on this choice, which should be emphasized. Tests for different alpha and W are required to have robust conclusions on the accuracy of FI3D.*

All tests use the same alpha and W choices, which demonstrates that this aspect of the methodology need not to be tuned by the user (although alternate choices may be appropriate if the characteristic wavelength of an application is significantly different than the

cases treated in our manuscript). We will add to the manuscript a discussion regarding the choice of the characteristic wavelength, which influences the effective viscosity, and a comparison between the response time scale of an incompressible vs. compressible 1D GIA model, which is relevant to the parameter alpha.

*FastIsostasy is presented in line 533-534 as a model that can greatly reduce the error of bedrock displacement compared to ELRA and ELVA and that is useful coupled to an ice sheet model due to the short computation time. However, there are no results presented in this manuscript that compare the performance of FastIsostasy with the most widely used GIA model in ice sheet modelling, the ELRA model and the laterally varying ELRA model so there could also be no conclusion about the reduction of the eror of FastIsostasy compared to ELRA. It is therefore not shown whether FastIsostasy is an improvement on what already exists. The focus of the paper should therefore be changed and the introduction rewritten with less focus on coupled ice sheet – GIA models, or a comparison with ELRA should be shown.*

We agree with this suggestion and will include ELRA in the comparisons treated in the manuscript. We will also explore the possible inclusion of LV-ELRA - however, this is a more complex undertaking since viscosity fields in this case need to be converted into fields of relaxation time.

*In line 361 the authors justify an error of 0.2. However, the error should ideally be smaller than the error introduced by different parametrizations. In some cases this error could be in the order of a hundred meters, which is significant and could have a large effect on ice dynamic models. In comparison, 1D GIA model benchmark study show much lower errors.*

Most reviewers rightfully indicate that our choice of the threshold in relative error is arbitrary. We will therefore remove it from the manuscript and simply mention the different values obtained in the tests. We emphasise that the appeal of FastIsostasy is an error reduction compared to ELVA (and ELRA as we will show in the revised version of the manuscript), without any significant increase in the computation time. It will be up to the user to decide whether the error level is acceptable for their application. This point will be made more explicit in the manuscript.

*Furthermore, from figure 8b it can be seen that the error between FastIsostasy 3D and Seakon 3D after 8 kyear is larger than the effect of 3D rheology itself.*

This is correct, although the errors incurred by adopting FastIsostasy are overall quite small (only a few percent). We emphasise that adopting a 3D GIA model is substantially more computationally expensive than FastIsostasy, which makes the latter appealing for many ice-sheet modellers.

*In that case, differences between FastIsostasy and global GIA models are outside of the range of parametric uncertainties, which contradicts the conclusion in line 534-535. The*

*conclusions on the performance of FastIsostasy should therefore be more considerate of the large error rather than accept them compared to an arbitrary standard, especially since the values hold for certain choices of the resolution.*

We agree with the comment. We will emphasise that the effect of LV (and therefore the parametric uncertainties) is only important for large transients displayed during the (last) deglaciation.

*Specific comments*

*Line 3-4: The impact of 3D GIA on ice sheet dynamics has only been shown for glacial cycles and not yet for projections. This sentence is suggesting that it has been shown. Please include in this sentence that the impact has only been shown over a glacial cycle.*

We agree with this comment, although we are aware of a manuscript by Natalya Gomez and colleagues that is in review that involves projections. We will cite this work if it has appeared online before this review process continues, but will revise the sentence if it has not.

We believe that it is still appropriate, later in the text, to point out that accelerated mass loss in Antarctica has been observed in the Amundsen region, where lower viscosities of the upper mantle have been robustly inferreded. Additionally, most studies showing a collapse of the WAIS display an onset of the instability in this very region. Therefore, 3D viscoelasticity can be expected to play an important role in future ice loss of the AIS.

*Line 6: An iterative coupling scheme is required when simulating a glacial cycle but it hasn't been studied yet whether iterations are required to simulate projections. Projections have been performed without an iterative coupling scheme using a 1D GIA model and there are no published projections using a 3D GIA model. The need for an iterative coupling scheme is therefore not an argument why 3D GIA models are not used in ice sheet models. I would suggest to, instead, include that 3D GIA models are not used in ice sheet modelling because the effect of 3D GIA is not known, 3D GIA models are computationally expensive and the coupling scheme is complex to apply.*

We strongly believe that 3D GIA models are not used in ice-sheet models because of their prohibitive computational requirements, not because there is a presumption that 3D variations in Earth structure are not important. Gomez's work, for example, amply demonstrates that such variability alters ice sheet evolution and so it is reasonable to assume that it impacts projections - an issue that is the purpose of the new Gomez article mentioned above. Nevertheless, we will revise the manuscript to avoid speculation.

*Line 12-13: Please include the value of error here instead of mentioning that the agreement is very good.*

We agree. The value will be cited in the revised manuscript.

*Line 15: The Fortran version is not provided yet, according to the data availability section. I suggest to include in this sentence that it will be provided.*

Since the submission of the paper, significant work has been invested in developing the Fortran version of the code, which is now ready to be used.

*Line 44-47: The impact of 3D GIA over a glacial cycle doesn't necessarily mean that the impact is also large over projections of a much shorter time scale. In multiple places in the introduction, the distinction between what has been studied over glacial cycles and what has been studied over projections is not clear. There are studies that show a significant effect of using 3D GIA compared to 1D GIA over a glacial cycle using a coupled ice sheet – GIA model (for example Gomez et al., 2018 & van Calcar et al., 2023) and from recent history till present day using a GIA model with a prescribed ice history (for example Blank et al., 2021). However, in this manuscript, results from Gomez et al. and van Calcar et al. are presented as if they show the impact of 3D GIA in projections as well, which is not the case. There are studies that show the importance of 3D GIA in projections using uncoupled models, such as Yousefi et al. (2022) but this study is not referenced in the manuscript. Currently, there are no publications on coupled ice sheet – 3D GIA models used for projections. This distinction should be made more clear throughout the introduction and the references to Yousefi et al. and Blank et al. should be added.*

Again, we are aware of new work that does demonstrate the importance of incorporating 3D structure in coupled GIA/ice-sheet projections. As mentioned above, if this work appears in print then we will cite it, but if it does not appear we will revise the text accordingly.

*Line 47: Include that sea level contributions from the basins are 19.2 and 3.4 m at present day.*

The values are mentioned in the manuscript and we understand that the reviewer wants to emphasise that they refer to the present-day state of Antarctica. We believe this is already unambiguous in the text.

*Line 58: Include references of the 3D GIA models that you are referring to.*

These references will be included in the revised manuscript.

*Line 62-64: Whether or not the ice-sheet modelling community is well aware of the how important 3D GIA is, is subjective. There are only a few studies showing the importance of 3D GIA over a glacial cycle and in projections and there are no published studies simulating projections using coupled ice dynamic – 3D GIA models. It can therefore be argued how well informed the ice sheet modelling community is up to this point and how well aware they could be without so many studies. I suggest to only mention that 3D GIA models are computationally expensive and complex to couple to an ice sheet model.*

We will revise the manuscript accordingly.

*Line 81: It is worth mentioning that there might be no asthenosphere at certain locations in Antarctica, and that ELRA includes that mantle, but that does not weigh against the confusion that it could cause to change a name that has been used in numerous papers since 1996. I suggest to leave the name as it is.*

*Line 86: Add the constant "flexural rigidity" and "lithospheric thickness" in the text, as these are other important parameters in the ELRA model.*

The lithospheric thickness is already mentioned and the flexural rigidity is derived from it, as pointed out later in the text.

*Line 101-102: To improve the readability, provide a short explanation about the difference between a viscous channel and a viscous half space.*

We agree. This explanation will be included in the revised manuscript.

*Line 109-112: Include that 1D GIA models also include the buoyancy effect of the core on the mantle and the mantle on the lithosphere.*

The impact of mantle buoyancy on the lithosphere (which is, by definition part of the mantle) is also captured by other models. Buoyancy effects of the core on mantle is not included in most 1D GIA models.

*Line 118: Include reference A et al. (2012), and Huang et al. (2023) for the finite element method.*

These references will be included in the revised manuscript.

*Line 119-121: The referenced models in this sentence (Gomez et al. and van Calcar et al.) are coupled ice sheet – GIA models, which require a much longer simulation time than 3D GIA models by itself. Since this section is solely about 3D GIA models, a simulation time of weeks is not applicable.*

Correct. Instead, we will refer to computation times of Seakon (4.5 days for glacial cycle as pointed out later).

*Line 123-124: The 3D GIA model in van Calcar et al. uses timesteps varying from approximately 1 to 1000 years, depending on the ice loading and the deformation rate, so the lower limit of the timestep of 3D GIA models is not accurate. Furthermore, it is not clear*

*why it is relevant in this context that GIA models sometimes have a larger timestep than ice sheet models.*

We agree.

*Line 126-139: The manuscript mentions two regional models specifically (Coulon et al., 2021 & Weerdesteijn et al., 2023). However, it is not clear why these two are picked out, since there are multiple other 1D and 3D GIA regional models (Nield et al., 2018; Book et al., 2022). I suggest to move line 126-129 to the section about LV-ELRA, and line 129-135 to the section about 1D GIA models. Also explain why Coulon et al. and Weerdesteijn et al. are mentioned specifically, and not other regional models. Some other important references are missing, such as Book et al. (2022), who used a similar method as this manuscript for a regional model focused on Thwaites glacier, and Kachuck et al. (2020).*

We believe that l.126-129 and 129-135 should not be moved since they build upon some of the listed models in the context of regional modelling, and particularly relevant for this publication. We will, however, include references to the papers by Nield, Book and Kachuck.

*Line 136-137: The available GIA models have a runtime acceptable for modelling ice sheets over glacial cycles, the runtime is only not acceptable to perform ensemble studies with a wide parameter space. Please include this nuance in the manuscript.*

We agree. This nuance will be included in the revised manuscript.

*Line 137-138: Could you define what is meant by "complexity gap" since there are regional 3D GIA models.*

We will define this more explicitly in the revised manuscript.

*Line 155-159: To improve readability, include a sentence to explain why a placeholder field is used and what the pseudo-differential operator represents.*

A placeholder field is cited here to illustrate the operation and it can be replaced by any other field. The pseudo-differential operator has no straightforward explanation in intuitive terms and we prefer here to refer to Bueler et al. (2007) on this issue.

*Line 168-169: Add whether the viscous half space have a variable or constant thickness.*

A half-space has no thickness since it is infinite. We believe that this is sketched in Fig. 3 and already included in the name "half-space".

*Line 180: Clarify in the text whether R is computed at each time step.*

Any computation related to the parameters, including the computation of the effective viscosity, are performed only once at initialisation. We will include this explanation in the revised manuscript.

*Line 248: Please include why it is required that the far-field displacement should be zero.*

This will be discussed in the revised manuscript.

*Line 307: It is not clear what "in-place" means.*

In-place refers to the fact that computations are performed without any memory allocation. This is of importance when programming in julia and is commonly used in other GMD publications.

*Line 340: To improve readability, include reference to Spada et al. (2011).*

This reference will be included in the revised manuscript.

*Line 415: Given the negligible maximal difference in displacement between the 1D GIA models of Spada et al. (2011), a maximal difference of 0.16 between FastIsostasy and the 1D GIA models of Spada et al. is relatively large. Also, purely based on this idealized test, it cannot be stated that FastIsostasy can replace 1D GIA models. This is also shown by figure 8, showing a maximal error of about 0.8 around -4000 years between the SK1D and FI1D, which is relatively large for a benchmark test.*

We emphasise that the mean error is very low (mean(e) < 0.05) and that the metric used in the manuscript (maximum over time and space) is the strictest possible metric. In particular the error is much lower for the cap load (max(e) < 0.07, mean(e) < 0.03), which, from an ice-dynamics perspective, is more coherent in space and therefore far more plausible than the disk load.

For the comparison between SK1D and FI1D, see Fig. A3, which shows a maximal error of about 0.16 (and not 0.8). We believe that this justifies replacing a 1D GIA model by FastIsostasy for many applications. We will state this only later in the text (around l. 475) since an earlier appearance of this statement (l. 415) misses complementary information from Test 4, as pointed out by the reviewer.

*Line 421-422: Please include the reference to the chosen viscosity.*

The viscosity fields are theoretically derived and were defined within the present manuscript. The fields are shown in the supplementary material, which is referred to in the main text.

*Line 438-439: The error of FastIsostasy compared to Seakon can be different when a realistic ice load with a realistic Earth rheology is used. Whether FastIsostasy can be used in regional ice sheet models should therefore be concluded based on test 4 as well and should not be stated based on only test 3.*

We agree with this and we will revise the manuscript accordingly to emphasise this point.

*Line 460-461: Could you quantify the error tolerance and adaptive time stepping.*

We believe this goes beyond the scope of the manuscript, since it depends on the specific application being considered.

*Line 472: Please include the results of this test in the manuscript and quantify what is meant with "better results".*

We agree. This will be included in the revised manuscript.

*Line 479: Quantify the offset in the forebulge and the implication of leaving the forebulge out of the presentation of the results.*

As pointed out in the answer to the third reviewer, the mask used for the error metric includes the forebulge for most of the simulation, only excluding it at LGM. This point should be resolved in the revised manuscript, since the mask adopted in the revision will be larger.

*Line 502: Define what is meant by "worst case".*

This text will be changed to "time step of maximal error" in the revised manuscript.

*Line 502: Define which region is meant by "near field".*

By near field we here refer to the displacement field below the ice sheet. This definition will be included in the revised manuscript.

*Line 505: The pattern of the rotational feedback is described as "a subtle dipole separated by a great circle" but when one doesn't know what the pattern of rotation looks like, the description is not so clear. It could be described as a gradient from east to west outside of the grounding line.*

We will revise the manuscript to make the geometry of rotational feedback clearer.

*Line 512: It would be useful to include what the runtime would be when a higher resolution is used.*

We believe that Fig. 5 already provides a measure of how the run time scales with the dimension. In particular, we emphasise that the same equations are solved regardless of the test, meaning that the run times are comparable for the same problem size. The only difference in run time originates from the stiffness of the right hand side of the formulation, as exemplified in test 4 between the laterally constant and laterally variable setup of FastIsostasy.

*Line 515: The timestep of the GIA model used in van Calcar et al. (2023) is dynamic and is about 1 year when the deformation rate is high. The convergence of the ice-sheet and GIA histories are there to reach a present day bedrock topography when simulation a glacial cycle. Those iterations would be needed by any ice sheet model coupled to FastIsostasy as well when a glacial cycle is simulated. The iterations are therefore not related to the time step of the GIA model. This should be corrected in the text.*

We agree with this point and we will revise the manuscript accordingly.

*Line 517: Include the resolution of Seakon.*

The resolution will be included in the revised manuscript.

*Line 520: It is unclear what is meant by "much richer".*

This terminology is explained in the text after "much richer".

*line 533-534: It should be stated more explicitly when FastIsostasy performs better than 1D GIA models, namely between -22 and -10 kyr of the glacial cycle.*

We agree and will revise the manuscript accordingly.

*Line 541-542: Clarify what is meant by "it minimizes the misrepresentation of the GIA feedback".*

We agree that this can be clarified by mentioning the deformational and gravitational response.

*Line 560: A discussion should be included that compares the conclusion of this paragraph with literature that does compare incompressible and compressible models, such as Huang et al. (2023). Is this increase in viscosity consistent with literature? Describe the limitations of increasing viscosity instead of including compressibility.*

See comment above. We will include a comparison between compressible and incompressible versions of the same 1D GIA code.

*Technical corrections*

We thank the reviewer for suggesting the corrections below. We agree with them and will address them, unless specified otherwise.

*Line 20: impacting > altering*
*Line 169/Footnote 1: API is not defined.*
*Line 225: Define F.*

F is defined by Eq. (9). It is simply a shorthand notation for the right-hand side of the equation.

*Line 242: Define ODE*

It is defined in the text (l. 200).

*Line 268: A0 is not defined.*
*Line 276: SLC is not defined.*
*Line 365: Define parameters.*
*Line 501: t = -14 kyr > t = -16 kyr*
*Figure and table comments*

*- Table 1: includes a description of the rheology, but it is not clear what is meant by Maxwell-like.*

Yes, this should be replaced by "relaxed"

*- Table 2: Please include a short description of the parameters in the caption.*
*- Figure 4: To improve readability, include the definition of variables in the caption.*

---

## Author Comment (AC6)

We are particularly grateful to have received extensive and constructive feedback from 4 reviewers. Since the reviewers' comments sum to more than 20 pages, we summarise, in the present document, the main changes that we envision for the revised manuscript. We believe that the vast majority of the reviewers' concerns will herewith be addressed.

Main changes:

1. Some reviewers were confused by the dimension of FastIsostasy's domain. This is partly due to the acronym "FI3D" used in Test 3 and Test 4, where "3D" refers to the dimension of the viscosity field. We understand this concern and will replace "FI1D" and "FI3D" with "FI (ELVA)" and "FI (LV-ELVA)" respectively. In addition, we will make clear, from the beginning, that FastIsostasy is a 2D model by slightly modifying the title to *FastIsostasy v1.0 – An accelerated, regional 2D GIA model accounting for the lateral variability of the solid Earth.*

2. The 4th reviewer suggests performing comparisons to ELRA in some of the tests. We will include such tests in the revised manuscript. We believe this new material will complement the existing analysis and make clearer that FastIsostasy represents a significant methodological improvement relative to commonly used regional approximations of the GIA response in ice-sheet modelling.

3. The original version of the manuscript defines a maximum error tolerance of 20% for all tests. Several reviewers suggested that this choice was arbitrary. We will therefore remove this threshold in the revised manuscript and replace it with additional comparisons of the error metrics across models.

4. The 2nd reviewer argued that Section 2.4. is incomplete. In the revised version of the manuscript, we will address this by describing more extensively how sea level is modelled and coupled to the deformational response. This is made possible by introducing the mask of active region; this is mentioned in the manuscript but material will be added to the supplement in the revised manuscript. We will explore the impact of the sea-level treatment in Test 4 by including a simulation with fixed sea level.

5. The 2nd, 3rd and 4th reviewer expressed concerns about the mask used in Test 4 to quantify the error, given the importance of the peripheral forebulge. This mask includes the forebulge during the vast majority (~90%) of the glacial cycle since it is based on the maximal extent (LGM) of the AIS. To avoid any concern in this regard, we will use a mask defined by the active region instead. We emphasise that this new mask will not affect the error metrics substantially.

6. The 3rd reviewer recommended indicating the location of load and forebulge in several of the figures appearing in the results section. We agree that this revision will ease the interpretation of the plots.

7. The 1st reviewer expressed concerns about the derivation of the governing equation of LV-ELVA. The submitted version of the manuscript clearly indicated that our approach stems from an ad-hoc combination of the equations used in Bueler et al. (2007) and Coulon et al. (2021). However, we will add a new section in the supplementary material to describe our approach in more detail.

8. The 1st and 3rd reviewers expressed concerns about the correction factor applied to the effective viscosity. To address this, we will show a comparison between relaxation times computed using a compressible and an incompressible 1D GIA model and will describe the limitations of lumping the depth dimension more extensively. In particular, we will discuss the choice of the characteristic wavelength in greater detail.

9. The 3rd reviewer suggests extending the comparison between Seakon and FastIsostasy by including a discussion of energy consumption (see detailed answer to 3rd reviewer for more details). We will do so in the revised manuscript.

10. The 3rd and 4th reviewer pointed out that the run times of 3D GIA models mentioned in the submitted manuscript are somewhat overestimated. We will revise this discussion by including the timing statistics cited in Albrecht et al. (in revision).

11. The reviewers suggested additional references, some of which we were not aware of. We will include all of these in the revised manuscript. In addition, we will improve the referencing to Coulon et al. (2021), Gomez et al. (2018), Van Calcar et al. (2023) and Albrecht et al. (2024).

12. In agreement with a comment of the 2nd reviewer, we will move the sections regarding the limitations of FastIsostasy to Section 2.

In addition to these revisions, we will address all the individual comments raised in the reviews, including the correction of small errors and confusing phrases.

We believe that FastIsostasy presents significant advantages relative to regional GIA models (by significantly increasing accuracy without an increase in computational cost) and to global 3D GIA models (by being simpler to implement and reducing time and energy consumption by at least three orders of magnitude, whilst introducing errors that are acceptable for most applications). FastIsostasy is not a replacement for a 3D GIA model in studies of global sea level variations across ice age cycles and into the modern world. However, FastIsostasy can be an important tool in ice sheet modelling by improving the representation of the deformational and gravitational feedbacks associated with laterally-variable Earth structures. We will emphasise this motivation in the revised manuscript.

To our knowledge, FastIsostasy is the only GIA model with a dynamically built documentation, a suite of automated tests and a fully transparent development process. Additionally, the Fortran version of the software is now ready to use, thus allowing the user to choose between the advantages of a modern programming language and those of a statically compiled code that is compatible with any language. These aspects are important, though admittedly of a more technical rather than scientific nature.

---

## Author Comment (AC7)

We are particularly grateful to have received extensive and constructive feedback from 4 reviewers. Since the reviewers' comments sum to more than 20 pages, we summarise, in the present document, the main changes that we envision for the revised manuscript. We believe that the vast majority of the reviewers' concerns will herewith be addressed.

Main changes:

1. Some reviewers were confused by the dimension of FastIsostasy's domain. This is partly due to the acronym "FI3D" used in Test 3 and Test 4, where "3D" refers to the dimension of the viscosity field. We understand this concern and will replace "FI1D" and "FI3D" with "FI (ELVA)" and "FI (LV-ELVA)" respectively. In addition, we will make clear, from the beginning, that FastIsostasy is a 2D model by slightly modifying the title to *FastIsostasy v1.0 – An accelerated, regional 2D GIA model accounting for the lateral variability of the solid Earth.*

   The title of the article and the naming of the experiments was changed accordingly. In particular see l. 552-554. and l. 604-605.

2. The 4th reviewer suggests performing comparisons to ELRA in some of the tests. We will include such tests in the revised manuscript. We believe this new material will complement the existing analysis and make clearer that FastIsostasy represents a significant methodological improvement relative to commonly used regional approximations of the GIA response in ice-sheet modelling.

   This was included (see Fig. 8 and 9).

3. The original version of the manuscript defines a maximum error tolerance of 20% for all tests. Several reviewers suggested that this choice was arbitrary. We will therefore remove this threshold in the revised manuscript and replace it with additional comparisons of the error metrics across models.

   This was changed (see Section 4).

4. The 2nd reviewer argued that Section 2.4. is incomplete. In the revised version of the manuscript, we will address this by describing more extensively how sea level is modelled and coupled to the deformational response. This is made possible by introducing the mask of active region; this is mentioned in the manuscript but material will be added to the supplement in the revised manuscript. We will explore the impact of the sea-level treatment in Test 4 by including a simulation with fixed sea level.

   Section 2.4 has been extended to include a more thorough description of the se-level treatment.

5. The 2nd, 3rd and 4th reviewer expressed concerns about the mask used in Test 4 to quantify the error, given the importance of the peripheral forebulge. This mask includes the forebulge during the vast majority (~90%) of the glacial cycle since it is based on the maximal extent (LGM) of the AIS. To avoid any concern in this regard, we will use a mask defined by the active region instead. We emphasise that this new mask will not affect the error metrics substantially.

The active mask is mentioned there and shown in Fig. 9, pointing out that it is larger than the LGM extent of the AIS. This did not impact the error plot (panel a). The transient errors in Fig. 9 (panel b) are computed for the whole domain and not only for the mask. This was not mentioned in the previous version of the manuscript and has now been included at l. 627.

6. The 3rd reviewer recommended indicating the location of load and forebulge in several of the figures appearing in the results section. We agree that this revision will ease the interpretation of the plots.

   This was included in Fig. 6, 7 & 8.

7. The 1st reviewer expressed concerns about the derivation of the governing equation of LV-ELVA. The submitted version of the manuscript clearly indicated that our approach stems from an ad-hoc combination of the equations used in Bueler et al. (2007) and Coulon et al. (2021). However, we will add a new section in the supplementary material to describe our approach in more detail.

   This was included in Appendix A.

8. The 1st and 3rd reviewers expressed concerns about the correction factor applied to the effective viscosity. To address this, we will show a comparison between relaxation times computed using a compressible and an incompressible 1D GIA model and will describe the limitations of lumping the depth dimension more extensively. In particular, we will discuss the choice of the characteristic wavelength in greater detail.

   This was included (see Fig. C1 and l. 265-280).

9. The 3rd reviewer suggests extending the comparison between Seakon and FastIsostasy by including a discussion of energy consumption (see detailed answer to 3rd reviewer for more details). We will do so in the revised manuscript.

   This was included at l. 666-670.

10. The 3rd and 4th reviewer pointed out that the run times of 3D GIA models mentioned in the submitted manuscript are somewhat overestimated. We will revise this discussion by including the timing statistics cited in Albrecht et al. (in revision).

    This was included, among others at l. 664-665.

11. The reviewers suggested additional references, some of which we were not aware of. We will include all of these in the revised manuscript. In addition, we will improve the referencing to Coulon et al. (2021), Gomez et al. (2018), Van Calcar et al. (2023) and Albrecht et al. (2024).

    This was improved (in particular see Introduction).

12. In agreement with a comment of the 2nd reviewer, we will move the sections regarding the limitations of FastIsostasy to Section 2.

    Done. See Section 2.5.

In addition to these revisions, we will address all the individual comments raised in the reviews, including the correction of small errors and confusing phrases.

We believe that FastIsostasy presents significant advantages relative to regional GIA models (by significantly increasing accuracy without an increase in computational cost) and to global 3D GIA models (by being simpler to implement and reducing time and energy consumption by at least three orders of magnitude, whilst introducing errors that are acceptable for most applications). FastIsostasy is not a replacement for a 3D GIA model in studies of global sea level variations across ice age cycles and into the modern world. However, FastIsostasy can be an important tool in ice sheet modelling by improving the representation of the deformational and gravitational feedbacks associated with laterally-variable Earth structures. We will emphasise this motivation in the revised manuscript.

To our knowledge, FastIsostasy is the only GIA model with a dynamically built documentation, a suite of automated tests and a fully transparent development process. Additionally, the Fortran version of the software is now ready to use, thus allowing the user to choose between the advantages of a modern programming language and those of a statically compiled code that is compatible with any language. These aspects are important, though admittedly of a more technical rather than scientific nature.

---

## Author Response (AR1)

The main changes that were performed are listed in the overview that we posted as an answer to the first round of comments by the reviewers.